# Concerted SUMO-targeted ubiquitin ligase activities of TOPORS and RNF4 are essential for stress management and cell proliferation

Julio C. Y. Liu [1], Leena Ackermann[1], Saskia Hoffmann[1], Zita Gál[1], Ivo A. Hendriks [2], Charu Jain[1], Louise Morlot[1], Michael H. Tatham [3], Gian-Luca McLelland [4], Ronald T. Hay [3], Michael Lund Nielsen [2], Thijn Brummelkamp [4,5], Peter Haahr [4,6] ✉ & Niels Mailand [1,7] ✉

Protein SUMOylation provides a principal driving force for cellular stress responses, including DNA–protein crosslink (DPC) repair and arsenic-induced PML body degradation. In this study, using genome-scale screens, we identified the human E3 ligase TOPORS as a key effector of SUMO-dependent DPC resolution. We demonstrate that TOPORS promotes DPC repair by functioning as a SUMO-targeted ubiquitin ligase (STUbL), combining ubiquitin ligase activity through its RING domain with poly-SUMO binding via SUMO-interacting motifs, analogous to the STUbL RNF4. Mechanistically, TOPORS is a SUMO1-selective STUbL that complements RNF4 in generating complex ubiquitin landscapes on SUMOylated targets, including DPCs and PML, stimulating efficient p97/VCP unfoldase recruitment and proteasomal degradation. Combined loss of TOPORS and RNF4 is synthetic lethal even in unstressed cells, involving defective clearance of SUMOylated proteins from chromatin accompanied by cell cycle arrest and apoptosis. Our findings establish TOPORS as a STUbL whose parallel action with RNF4 defines a general mechanistic principle in crucial cellular processes governed by direct SUMO–ubiquitin crosstalk.

Protein modification by small ubiquitin-like modifier (SUMO) polypeptides impacts thousands of cellular targets and constitutes an important regulatory signaling mechanism in numerous aspects of biology, especially cellular stress responses[1,2]. SUMOylation proceeds via a three-step enzymatic cascade involving E1, E2 and E3 enzymes, similar to the machinery underlying protein ubiquitylation. Although SUMOylation per se is a non-proteolytic modification, extensive crosstalk between SUMO and ubiquitin signals occurs in cells, positioning SUMOylation to effectively instruct the proteasomal degradation of many SUMO targets, brought about by their SUMO-dependent ubiquitylation. This is mediated by SUMO-targeted ubiquitin ligases (STUbLs), which combine E3 ubiquitin ligase activity with SUMO-interacting motifs (SIMs), enabling them to specifically recognize and ubiquitylate SUMOylated proteins[1,3]. STUbLs typically contain multiple SIMs,

[1]Protein Signaling Program, Novo Nordisk Foundation Center for Protein Research, University of Copenhagen, Copenhagen, Denmark. [2]Proteomics Program, Novo Nordisk Foundation Center for Protein Research, University of Copenhagen, Copenhagen, Denmark. [3]Centre for Gene Regulation and Expression, School of Life Sciences, University of Dundee, Dundee, UK. [4]Division of Biochemistry, The Netherlands Cancer Institute, Amsterdam, The Netherlands. [5]Oncode Institute, Utrecht, The Netherlands. [6]Department of Cellular and Molecular Medicine, Center for Gene Expression, University of Copenhagen, Copenhagen, Denmark. [7]Department of Cellular and Molecular Medicine, Center for Chromosome Stability, University of Copenhagen, Copenhagen, Denmark. ✉e-mail: peter.haahr@sund.ku.dk; niels.mailand@cpr.ku.dk

imparting selectivity for proteins modified by poly-SUMOylation, which predominantly involves SUMO2/3 that, unlike SUMO1, undergoes efficient chain formation, similar to ubiquitin[1]. Reminiscent of STUbLs, the deubiquitinase USP7 has been reported to function as a SUMO-targeted ubiquitin protease counteracting SUMO-dependent protein ubiquitylation and degradation[4].

Two mammalian STUbLs, RNF4 and RNF111, which differ with respect to their preference for recognizing different SUMO chain configurations, are currently known[5–9]. These enzymes have key roles in maintaining genome integrity, exerted in part by ubiquitylating proteins that undergo SUMOylation upon their association with chromatin, driving their displacement[1,3,10]. An extreme case of this regulatory principle is seen for DNA–protein crosslinks (DPCs), where the protein adduct becomes extensively SUMOylated due to its covalent trapping on DNA[11,12]. DPCs, generated by many chemotherapeutic agents, are bulky cytotoxic DNA lesions that undermine genome integrity by obstructing essential DNA-associated transactions[13,14]. We and others showed that, outside the context of DNA replication, DPC resolution critically relies on SUMOylation of the protein adduct, triggering its subsequent ubiquitylation by RNF4 and proteolytic processing via the p97/VCP unfoldase and the proteasome[11,12,15,16]. This is prominently observed for post-replicative DPCs involving the DNA methyltransferase DNMT1, generated by the cytosine analog 5-Aza-2′-deoxycytidine (5-AzadC), which, after its incorporation into genomic DNA during chromosome replication, forms a DPC with the DNMT1 catalytic cysteine upon DNA methylation re-establishment[11,15,17]. Importantly, 5-AzadC-induced DPCs not only provide a model for studying SUMO-dependent resolution of defined DPCs, as 5-AzadC (also known as decitabine) and related DNA methyltransferase inhibitors are also used clinically for treatment of acute myeloid leukemia and myelodysplastic syndrome[18]. Indeed, combining 5-AzadC and SUMO inhibitor treatment synergizes in reducing lymphoma tumor cell growth in vitro and in vivo[19], consistent with the requirement for SUMOylation in promoting DNMT1 DPC resolution. Direct SUMO-ubiquitin crosstalk also has a critical role in promoting proteasomal degradation of PML bodies in response to arsenic treatment, which induces SUMOylation and ensuing RNF4-dependent ubiquitylation of the PML protein[8,9]. This has important ramifications for treatment of acute promyelocytic leukemias caused by a chromosomal translocation fusing the PML and RARA (retinoic acid receptor alpha) genes, and most patients can be cured by combination therapy involving arsenic and retinoic acid[20]. These and other studies show that STUbLs are crucial effectors of SUMO-regulated stress responses, whose pharmacological intervention have a demonstrated potential in cancer therapeutics. Whether mammalian genomes encode STUbLs in addition to RNF4 and RNF111 is unclear.

In the present study, we discovered that the E3 ligase protein TOPORS defines a STUbL in human cells, acting in parallel with RNF4 to promote critical SUMO-driven responses by promoting efficient poly-ubiquitylation of SUMOylated substrates to facilitate their processing via the p97-proteasome pathway, with TOPORS preferentially targeting SUMO1-modified proteins. Moreover, we reveal a strong synthetic lethal interaction between RNF4 and TOPORS or the deubiquitinase USP7, whose activity is essential for sustaining TOPORS stability, characterized by accumulation of hyper-SUMOylated proteins on chromatin accompanied by defective cell cycle progression and apoptosis. Collectively, our findings establish concerted TOPORS and RNF4 STUbL activities as a general mechanistic principle in critical cellular processes governed by SUMO–ubiquitin crosstalk.

## Results

### TOPORS is required for SUMO-dependent DPC repair

Utilizing the notion that the cellular DNMT1 pool is quantitatively depleted after 5-AzadC-induced formation and resolution of DNMT1 DPCs[11,15], we performed genome-wide insertional gene-trap mutagenesis screens with ultra-deep resolution in human HAP1 cells to identify regulators of SUMO-dependent DNMT1 DPC resolution. Mutagenized cells subjected to 5-AzadC or mock treatment were fixed and stained with a specific DNMT1 antibody, and cell populations with relatively high or low antibody staining were isolated by fluorescence-activated cell sorting (FACS) and analyzed by deep sequencing (Fig. 1a and Extended Data Fig. 1a). The screens revealed a range of genes enriched for disruptive gene-trap mutations in 5-AzadC-treated, but not mock-treated, cells with high DNMT1 signal, indicative of a potential DNMT1 DPC repair defect (Fig. 1b, Extended Data Fig. 1b and Supplementary Data 1). Both RNF4 and SUMO2 scored as specific negative regulators of DNMT1 abundance in 5-AzadC-treated cells, consistent with their key roles in DNMT1 DPC degradation[15], whereas the antibody target encoded by DNMT1 was the strongest positive regulator, as expected (Fig. 1b). DCK, encoding deoxycytidine kinase that is required for 5-AzadC incorporation into genomic DNA (and, thus, DNMT1 DPC formation) by converting it to 5-AzadCMP[21], also scored as a prominent negative regulator (Fig. 1b), further supporting the validity of our screen. Interestingly, insertions in the gene encoding TOPORS, a RING-type E3 ligase, were strongly enriched among cells showing defective DNMT1 degradation upon 5-AzadC treatment (Fig. 1b). Using previously established chromatin fractionation–based and quantitative imaging–based assays to monitor DNMT1 DPC repair kinetics[15], we found that TOPORS-knockout (KO) in independent HAP1 clones or siRNA-mediated knockdown in U2OS osteosarcoma cells strongly delayed DNMT1 DPC resolution after 5-AzadC treatment (Fig. 1c–e and Extended Data Fig. 1c–e). TOPORS deficiency also impaired the timely clearance of SUMO2/3 foci induced by treatment with formaldehyde, another inducer of DPCs that are processed via SUMOylation[11] (Extended Data Fig. 1f,g). Supporting a direct involvement of TOPORS in processing SUMOylated DPCs, TOPORS interacted with DNMT1 in a 5-AzadC-dependent and SUMO-dependent

**Fig. 1 | Genome-wide screens reveal an essential role of TOPORS in SUMO-dependent DPC repair. a**, Workflow of FACS-based haploid genetic screens for DNMT1 abundance (created with BioRender). **b**, Screen for DNMT1 abundance in EdU-positive cells after co-treatment with 5-AzadC and EdU (n = 1). Positive and negative regulators of DNMT1 abundance are labeled blue and yellow, respectively (two-sided Fisher's exact test, FDR-corrected P ≤ 0.05; non-significant genes are shown in gray). Genes scoring as significant DNMT1 regulators in a mock screen (Extended Data Fig. 1b) were filtered out, except for DNMT1. **c**, HAP1 cell lines released from single-round thymidine synchronization in early S phase were treated with 5-AzadC for 30 min and collected at indicated times. Chromatin-enriched fractions were immunoblotted with indicated antibodies. **d**, Representative images of U2OS cells transfected with indicated siRNAs, treated as in **c**, pre-extracted and immunostained with DNMT1 antibody. Scale bar, 10 μm. **e**, DNMT1 foci formation in **d** was analyzed by quantitative image-based cytometry (QIBC) (red bars, mean; >8,300 cells analyzed per condition). **f**, HeLa cells stably expressing GFP–DNMT1 were synchronized in early S phase by double thymidine block and release, treated with 5-AzadC and/or SUMOi for 30 min and subjected to GFP IP under denaturing conditions. Immobilized GFP–DNMT1 complexes were incubated with whole-cell lysate from parental HeLa cells and immunoblotted with indicated antibodies. **g**, Clonogenic survival of 5-AzadC-treated U2OS cells transfected with indicated siRNAs (mean ± s.e.m.; n = 3 independent experiments). **h**, Immunoblot analysis of U2OS cells treated with USP7i for the indicated times in presence or absence of the proteasome inhibitor MG132. Arrow indicates band corresponding to endogenous TOPORS. **i**, U2OS cells released from single thymidine synchronization in early S phase were treated with SUMOi and 5-AzadC for 30 min, released into medium containing USP7i or not and collected at indicated times. DNMT1 foci formation was analyzed as in **d** and **e** (red bars, mean; >20,000 cells analyzed per condition). **j**, Clonogenic survival of 5-AzadC-treated U2OS cells in presence or absence of USP7i (mean ± s.e.m.; n = 3 independent experiments). Data information: data are representative of five (**d**), three (**e,f,i**) and two (**c,h**) independent experiments with similar outcomes.

manner, mirroring the behavior of RNF4 (Fig. 1f). Moreover, cells lacking TOPORS displayed hypersensitivity to treatment with 5-AzadC or formaldehyde (Fig. 1g and Extended Data Fig. 1h), similar to our previous findings for RNF4 (ref. 15).

Additional ubiquitin network components including *USP7* and *UBE2K* were also enriched as significant negative regulators of

DNMT1 abundance upon 5-AzadC treatment (Fig. 1b and Supplementary Data 1). Recent proteomic studies suggested acute and highly selective loss of TOPORS expression upon treatment of cells with inhibitors of the deubiquitinase USP7 (refs. 22,23), and we confirmed that a specific USP7 inhibitor, FT671 (ref. 24) (USP7i), caused a rapid proteasome-dependent decline in TOPORS abundance, whereas RNF4

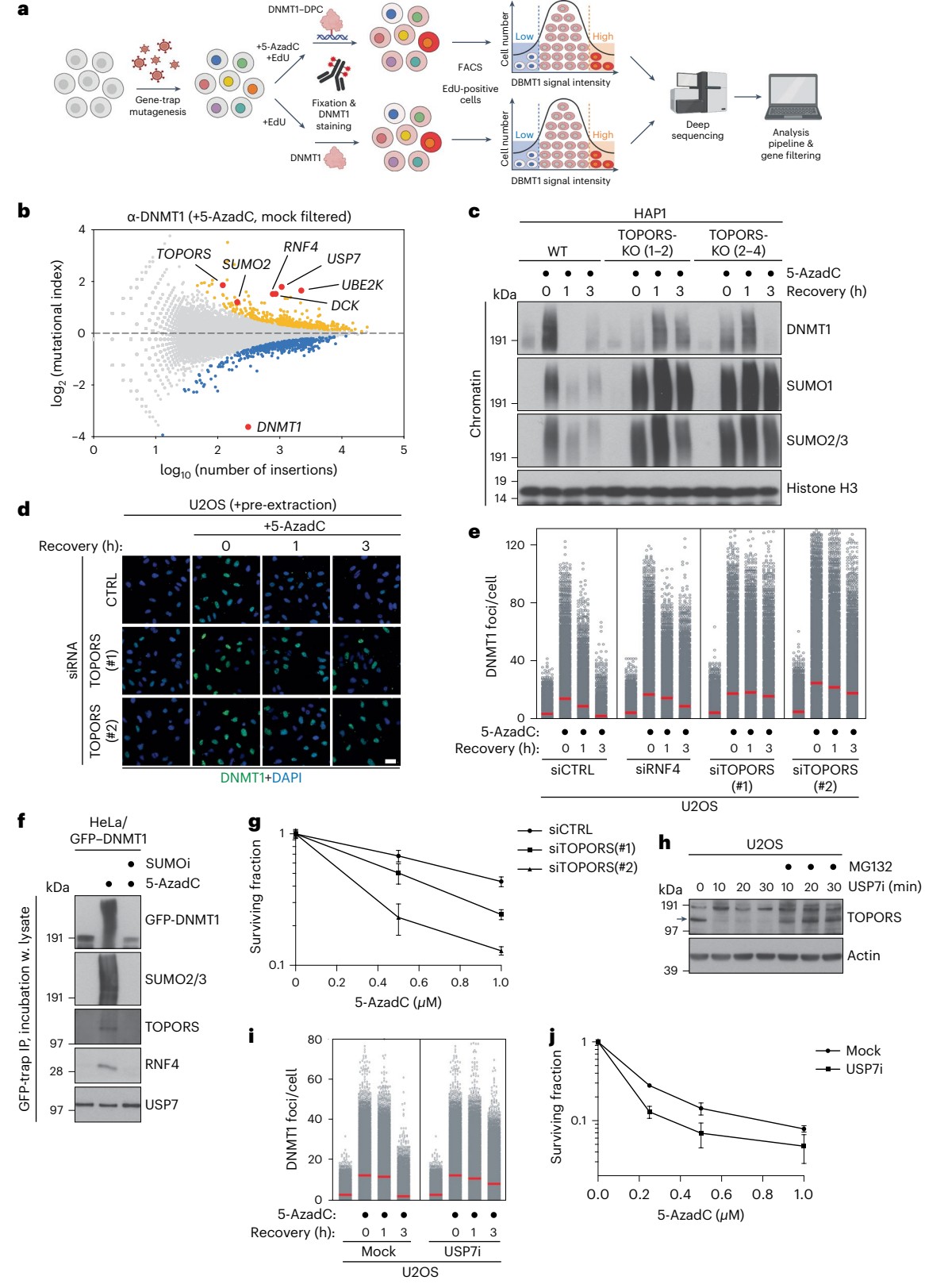

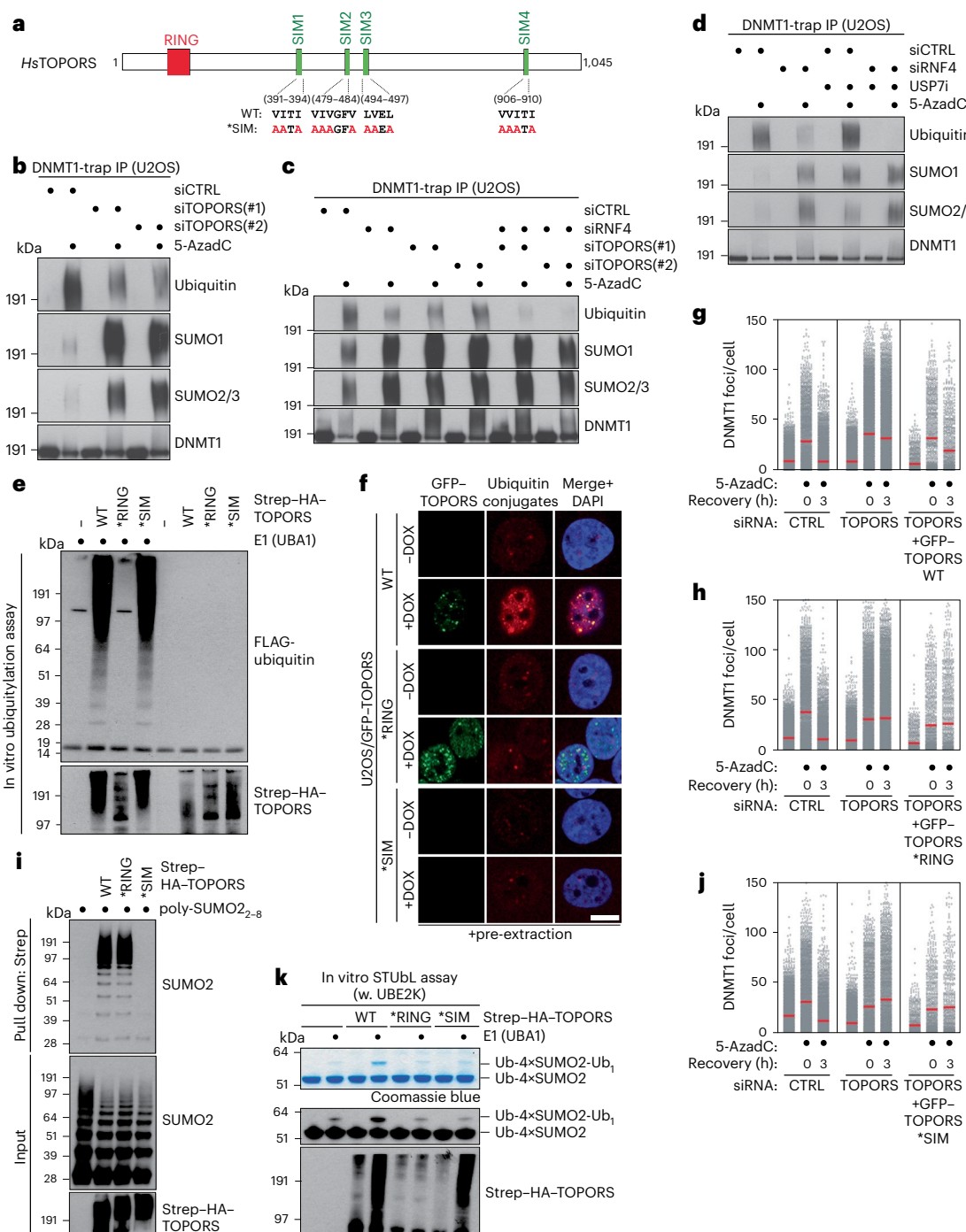

**Fig. 2 | TOPORS functions as a SUMO-targeted ubiquitin ligase in DPC repair.**
**a**, Domain organization of human TOPORS and mutations introduced to generate the TOPORS *SIM mutant. **b**, Immunoblot analysis of U2OS cells transfected with siRNAs for a total of 72 h that were released from synchronization in early S phase, treated or not with 5-AzadC for 30 min and subjected to DNMT1 IP under denaturing conditions. **c**, As in **b**, except that cells were treated with 5-AzadC 40 h after siRNA transfection. **d**, As in **b**, except that USP7i was added where indicated. **e**, Immunoblot analysis of in vitro ubiquitylation reactions containing recombinant Strep–HA–TOPORS proteins (Extended Data Fig. 2c), E1 (UBA1) and E2 (UBE2D1) enzymes, FLAG–ubiquitin, ATP and Ub–VS and SUMO2–VS. **f**, Representative images of stable U2OS/GFP–TOPORS cell lines that were induced or not to express GFP–TOPORS proteins with doxycycline (DOX), pre-extracted and fixed and immunostained with ubiquitin conjugate-specific antibody (FK2). Scale bar, 10 μm. See Extended Data Fig. 2e for images of non-pre-extracted U2OS/GFP–TOPORS cell lines where expression of GFP–TOPORS *SIM can be seen. **g**, U2OS cells were sequentially transfected with siRNAs and expression plasmids (GFP–TOPORS WT and mCherry–H2B). Cells were then subjected to single-round thymidine synchronization in early S phase and, upon release from the block, treated with 5-AzadC for 30 min and pre-extracted and fixed at indicated times. After DNMT1 immunostaining, DNMT1 foci formation in transfected (mCherry-positive) cells was analyzed by quantitative image-based cytometry (QIBC) (red bars, mean; >1,030 cells analyzed per condition). **h**. As in **g**, except that cells were transfected with GFP–TOPORS *RING and mCherry–H2B plasmids (red bars, mean; >680 cells analyzed per condition). **i**, Immunoblot analysis of recombinant Strep–HA–TOPORS proteins incubated with poly-SUMO2 chains and subjected to Strep-Tactin pulldown. **j**, As in **g**, except that cells were transfected with GFP–TOPORS *SIM and mCherry–H2B plasmids (red bars, mean; >910 cells analyzed per condition). **k**, Coomassie staining (top) and immunoblot analysis using SUMO2 and HA antibodies (bottom) of in vitro STUbL reactions containing recombinant Strep–HA–TOPORS proteins, E1 (UBA1) and E2 (UBE2K) enzymes, FLAG–ubiquitin, 4×SUMO2 STUbL, Ub–VS and SUMO2–VS. Data information: data are representative of four (**c**) and three (**b**,**d**–**k**) independent experiments with similar outcomes.

was not affected (Fig. 1h and Extended Data Fig. 1i). Consistently, like TOPORS depletion, USP7i delayed DNMT1 DPC turnover and hypersensitized cells to 5-AzadC (Fig. 1i,j and Extended Data Fig. 1j). However, unlike TOPORS and RNF4, USP7 did not display 5-AzadC-dependent and SUMO-dependent interaction with DNMT1 (Fig. 1f), suggesting that it functions indirectly in SUMO-dependent DPC resolution by underpinning TOPORS stability. Indeed, TOPORS, but not RNF4, interacted with USP7 and was modified by auto-ubiquitylation that was antagonized by recombinant USP7 (Extended Data Fig. 1k–m). Collectively, these data identify TOPORS as a critical factor in SUMO-dependent DPC resolution and suggest that USP7 deubiquitinase activity is required for this response by counteracting proteasome-mediated TOPORS degradation.

### TOPORS functions as an E3 ubiquitin ligase in DPC repair

We next asked how TOPORS promotes SUMO-dependent DPC resolution. Although TOPORS contains a conserved N-terminal RING domain (Fig. 2a), its precise enzymatic function remains unclear, as TOPORS has been reported to function as an E3 ligase for both ubiquitin and SUMO[25–30]. We, therefore, investigated how TOPORS impacts DNMT1 DPC SUMOylation and ubiquitylation, using stringent isolation of endogenous DNMT1 under denaturing conditions. TOPORS knockdown decreased DNMT1 DPC ubiquitylation, whereas DNMT1 DPC SUMOylation was greatly enhanced (Fig. 2b and Extended Data Fig. 2a), reminiscent of our findings for RNF4 (ref. 15). Co-depletion of RNF4 and TOPORS, whose impact could be assessed only at early timepoints after siRNA transfection due to synthetic lethality as detailed below, reduced DNMT1 DPC ubiquitylation, but not SUMOylation, to background levels, unlike the modest effect of individually depleting either protein for a short, 40-h period (Fig. 2c). We observed similar effects of TOPORS and/or RNF4 depletion on SUMO-dependent ubiquitylation of camptothecin-induced topoisomerase 1 (TOP1) DPCs (Extended Data Fig. 2b,c), consistent with the known role of RNF4 in ubiquitylating these DPCs and the association between TOPORS and TOP1 (refs. 12,31). These findings suggest that RNF4 and TOPORS both contribute to ubiquitylation of SUMOylated DPCs and together are responsible for the bulk of these modifications. Indeed, co-depletion of TOPORS and RNF4 abrogated DNMT1 DPC resolution (Extended Data Fig. 2d), similar to the effect of inhibiting SUMOylation[15]. Quantitative proteomic analysis further indicated an inverse impact of TOPORS or RNF4 knockdown on 5-AzadC-induced DNMT1 ubiquitylation and SUMOylation levels (Extended Data Fig. 2e and Supplementary Data 2), suggesting an uncoupling of these modifications. Like TOPORS knockdown, treatment with USP7i that rapidly reduces TOPORS abundance exacerbated the DNMT1 DPC ubiquitylation defect in cells lacking RNF4, even though USP7i alone moderately increased 5-AzadC-induced DNMT1 ubiquitylation, possibly reflecting its reported role in deubiquitylating SUMO and/or stabilizing DNMT1 itself[4,32–34] (Figs. 1h and 2d).

These observations suggest that TOPORS might catalyze ubiquitylation, but not SUMOylation, of DNMT1 DPCs in parallel with RNF4. Consistently, purified wild-type (WT) TOPORS displayed strong E3 ubiquitin ligase in vitro, and this activity was abrogated by introducing point mutations predicted to disrupt the E2-binding capacity of its RING domain (*RING) (Fig. 2e and Extended Data Fig. 2f). By contrast, little SUMO E3 ligase activity of TOPORS was apparent in vitro, and no difference between TOPORS WT and *RING was observed (Extended Data Fig. 2g). Moreover, using cell lines conditionally expressing GFP–TOPORS proteins, we noted that the detergent-resistant, chromatin-associated nuclear GFP–TOPORS foci displayed strong enrichment of ubiquitin conjugates in a RING-dependent manner, whereas both GFP–TOPORS WT and *RING foci co-localized with, and showed a similar level of, SUMO2/3 (Fig. 2f and Extended Data Fig. 2h,i). Ectopically expressed GFP–TOPORS WT but not *RING alleviated the DNMT1 DPC resolution defect resulting from depletion of endogenous TOPORS (Fig. 2g,h), indicating that TOPORS E3 ligase activity is

required for DNMT1 DPC resolution. In fact, overexpression of TOPORS *RING not only failed to rescue the DNMT1 DPC repair defect in cells lacking endogenous TOPORS but also had a dominant-negative impact on DNMT1 DPC resolution when endogenous TOPORS was present (Extended Data Fig. 2j). Collectively, these data suggest that TOPORS acts as an E3 ubiquitin ligase for SUMOylated DPCs along with RNF4.

### TOPORS is a STUbL

Given the similar behavior and requirements of RNF4 and TOPORS in SUMO-dependent DPC resolution, we surmised that TOPORS might function as a STUbL. Supporting this idea, TOPORS readily interacted with poly-SUMO2 chains in vitro (Fig. 2i), as also indicated by proteomics-based profiling of SUMO-binding proteins in human cells[35]. Sequence analysis revealed four prospective conserved SIMs in the middle and C-terminal portions of TOPORS (Fig. 2a). Simultaneous functional disruption of these motifs by alanine substitutions in their hydrophobic core (*SIM) had no impact on TOPORS E3 ubiquitin ligase activity in vitro but abrogated TOPORS interaction with poly-SUMO2 chains and accumulation in detergent-resistant nuclear SUMO2/3-positive foci (Fig. 2a,e,f,i and Extended Data Fig. 2i). Consistent with the presence of multiple SIMs, TOPORS preferentially bound long poly-SUMO2 chains and did not interact with monomeric SUMO2 (Fig. 2i and Extended Data Fig. 3a). Despite maintaining full E3 ubiquitin ligase activity in vitro, GFP–TOPORS *SIM did not promote increased nuclear ubiquitylation, unlike its WT counterpart (Fig. 2f), suggesting that TOPORS might function as a STUbL. Moreover, the TOPORS *SIM mutant failed to rescue the DNMT1 DPC repair defect of cells depleted for endogenous TOPORS and did not interact with DNMT1 DPC sites (Fig. 2g,j and Extended Data Fig. 3b).

To directly assay for TOPORS STUbL activity, further anticipated based on the combination of ubiquitin E3 ligase activity and SIMs, we performed in vitro ubiquitylation assays with purified TOPORS proteins and linearly fused 4×SUMO2 proteins mimicking a poly-SUMO2 chain[36]. Notably, TOPORS stimulated ubiquitylation of 4×SUMO2 in a RING-dependent and SIM-dependent manner, establishing it as a bona fide STUbL (Fig. 2k). However, compared to bacterially purified RNF4, TOPORS was inefficient in promoting 4×SUMO2 ubiquitylation in the presence of the generic E2 ubiquitin-conjugating enzyme UBE2D1 (Extended Data Fig. 3c). TOPORS displayed higher STUbL activity in the presence of UBE2K, an E2 enzyme that was among the hits in our genetic screen for DNMT1 DPC repair factors (Fig. 1a,b and Supplementary Data 1), similar to RNF4 whose 4×SUMO2-directed E3 ligase activity was supported to a similar extent by UBE2D1 and UBE2K (Extended Data Fig. 3c). However, TOPORS and RNF4 only stimulated UBE2K-dependent ubiquitylation of a ubiquitin-fused 4×SUMO2 protein (Ub–4×SUMO2) but not 4×SUMO2 (Fig. 2k and Extended Data Fig. 3c), consistent with UBE2K functioning as an E2 that elongates ubiquitin modifications by catalyzing their K48 linkage-specific ubiquitylation[37–39]. Thus, as elaborated below, TOPORS appears to be a relatively inefficient STUbL toward SUMO2/3-modified proteins that is mainly able to extend pre-existing ubiquitylation marks on these substrates. Consistent with a potential role of UBE2K in this process and our screen data, UBE2K knockdown delayed the resolution of DNMT1 DPCs (Extended Data Fig. 3d,e). We conclude that TOPORS is a novel STUbL that promotes SUMO-dependent DPC repair via this activity.

### TOPORS and RNF4 cooperatively drive multiple STUbL pathways

The joint action of RNF4-dependent and TOPORS-dependent STUbL activities in promoting SUMO-dependent DPC resolution raised the question of whether this partnership extends to other STUbL-driven processes. We, therefore, used an established PML KO cell line stably reconstituted with YFP-tagged PML[40] to assess whether TOPORS is involved in arsenic-induced PML degradation, which is also mediated by SUMOylation and RNF4 STUbL activity[8,9]. Interestingly, paralleling

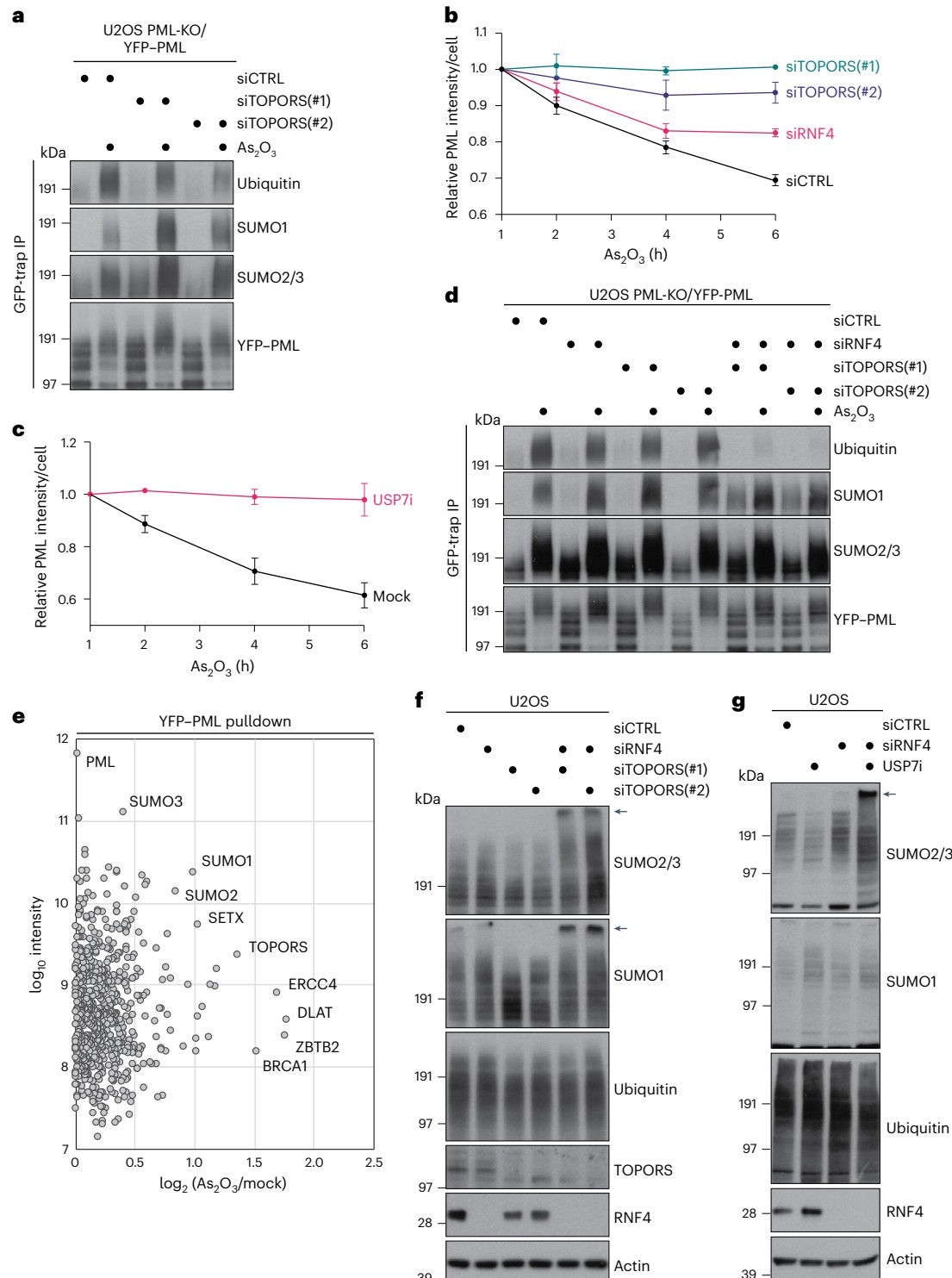

**Fig. 3 | TOPORS and RNF4 cooperatively drive multiple STUbL-driven responses. a**, U2OS PML-KO cells stably reconstituted with YFP-tagged PML-V (U2OS PML-KO/YFP–PML) were transfected with indicated siRNAs for 72 h and exposed to arsenic ($As_2O_3$) for 1 h. Cells were subjected to GFP IP under denaturing conditions followed by immunoblotting with indicated antibodies. **b**, U2OS PML-KO/YFP–PML cells transfected with indicated siRNAs were treated with arsenic and fixed at indicated timepoints. YFP–PML intensity was analyzed by quantitative image-based cytometry (QIBC) (mean ± s.e.m.; $n = 3$ independent experiments). **c**, U2OS PML-KO/YFP–PML cells were exposed to arsenic in the presence or absence of USP7i, fixed at the indicated times and analyzed as in **b** (mean ± s.e.m.; $n = 3$ independent experiments). **d**, As in **a**, except that cells were treated with arsenic 40 h after transfection with indicated siRNAs. **e**, U2OS PML-KO/YFP–PML cells were grown in SILAC medium and exposed to

1 µM arsenic for 2 h (heavy condition) or left untreated (light condition). PML bodies were enriched by affinity purification using GFP nanobody crosslinked to magnetic beads, and associated proteins were identified and quantified by mass spectrometry. Selected outliers with increased abundance and PML itself are indicated. A single experiment was performed. **f**, Immunoblot analysis of U2OS cells transfected with indicated siRNAs for 40 h. Slow-migrating, hyper-SUMOylated proteins ('well products') are indicated by arrows. See also Supplementary Fig. 1a. **g**, Immunoblot analysis of U2OS cells transfected with indicated siRNAs for 48 h and grown in the absence or presence of USP7i for an additional 12 h. Slow-migrating, hyper-SUMOylated proteins ('well products') are indicated by arrows. See also Supplementary Fig. 1b. Data information: data are representative of three (**a,d,f,g**) independent experiments with similar outcomes.

the impact of RNF4 depletion (Extended Data Fig. 4a), TOPORS knockdown diminished arsenic-induced PML ubiquitylation, whereas SUMO-modified forms of PML accumulated (Fig. 3a). This effect appeared particularly prominent for SUMO1, a trend also seen for DNMT1 DPCs (Figs. 2b and 3a and Extended Data Fig. 2a). Consistently, TOPORS knockdown strongly impaired arsenic-induced PML body degradation, to a greater extent than the defect seen in RNF4-depleted cells (Fig. 3b and Extended Data Fig. 4b). Inhibiting USP7 activity likewise impaired the timely clearance of PML bodies after arsenic treatment (Fig. 3c), consistent with previous observations[41]. Although individual TOPORS or RNF4 depletion for a short (40-h) period only mildly reduced arsenic-induced PML ubiquitylation, their combined knockdown largely abrogated this modification (Fig. 3d). This suggests that RNF4 and TOPORS cooperatively promote arsenic-induced PML ubiquitylation and degradation, mirroring their joint involvement in DPC resolution. Further linking TOPORS to PML body degradation, proteomic analysis of YFP–PML pulldowns showed increased TOPORS association with PML upon arsenic treatment (Fig. 3e, Extended Data Fig. 4c and Supplementary Data 3)[40]. Moreover, overexpression of TOPORS *RING, but not WT, significantly stabilized PML bodies (Extended Data Fig. 4d), indicating that TOPORS E3 ligase activity is important for turning over these structures. These findings suggest that TOPORS and RNF4 jointly promote multiple STUbL-driven pathways. In line with this notion, we noted that co-depletion of RNF4 and TOPORS, but not individual knockdown of either ligase, led to a striking accumulation of high-molecular-weight SUMOylated proteins in total cell extracts manifesting as slow-migrating 'well products' in SDS-PAGE (Fig. 3f and Supplementary Fig. 1a), suggesting a severe defect in processing SUMO-modified proteins when both RNF4 and TOPORS are absent. Again, USP7i treatment recapitulated the impact of TOPORS depletion in causing accumulation of hyper-SUMOylated proteins in cells lacking RNF4 but not in control cells (Fig. 3g and Supplementary Fig. 1b). These findings suggest that the coupling of TOPORS and RNF4 E3 ubiquitin ligase activities is a general principle in STUbL-driven pathways, exemplified by their joint action in SUMO-dependent processing of DPCs and PML bodies.

### TOPORS and RNF4 jointly recruit p97 to STUbL substrates

We next asked why both RNF4 and TOPORS are necessary for efficient STUbL substrate turnover. Recent work, which we confirmed (Extended Data Fig. 5a,b), demonstrated that p97 activity is essential for SUMO-dependent degradation of DNMT1 DPCs and PML bodies[16,40]. Indicative of a role for TOPORS STUbL activity in promoting p97 recruitment to SUMOylated proteins, the nuclear foci formed by stably expressed GFP–TOPORS not only accumulated high levels of ubiquitin conjugates but also displayed striking p97 enrichment, in a manner dependent on TOPORS E3 ligase activity and SUMO binding (Figs. 2f and 4a). Consistently, TOPORS knockdown markedly reduced p97 recruitment to DNMT1 DPC sites, to a similar extent as RNF4 depletion (Fig. 4b). Moreover, unlike WT TOPORS, cells expressing TOPORS *RING or *SIM failed to support p97 recruitment to DNMT1 DPC sites (Extended Data Fig. 5c). Depletion of TOPORS or RNF4 likewise diminished p97 occupancy in PML bodies upon arsenic treatment (Fig. 4c). Furthermore, p97 inhibition phenocopied the accumulation of hyper-SUMOylated proteins seen upon combined RNF4 and TOPORS loss (Extended Data Fig. 5d), suggesting that these STUbLs together ensure efficient processing of SUMOylated proteins by promoting p97 recruitment. The role of RNF4 and TOPORS in promoting p97-dependent processing of ubiquitylated substrates may be confined to STUbL targets, as neither RNF4 nor TOPORS depletion affected the levels of a ubiquitylated model substrate Ub(G76V)-GFP whose proteasomal turnover requires p97 activity[42] but not SUMOylation (Extended Data Fig. 5e).

These findings suggested that TOPORS and RNF4 both make important contributions to ubiquitin signals underlying p97 recruitment to STUbL substrates. To test this, we quantified the abundance

of individual ubiquitin linkages in our proteomic analysis of DNMT1 DPCs isolated under stringent conditions (Fig. 2b). This showed that DNMT1 DPCs are mainly modified by K48-linked ubiquitin chains (Extended Data Fig. 5f and Supplementary Data 2), a major signal for p97-dependent and proteasome-dependent turnover[43], and that both TOPORS and RNF4 depletion leads to a sizeable decrease in these modifications (Fig. 4d and Supplementary Data 2), in accordance with our findings above. By contrast, RNF4, but not TOPORS, was required for the comparatively lower level of 5-AzadC-induced K63-linked and K11-linked ubiquitylation of DNMT1 (Fig. 4d, Extended Data Fig. 5f and Supplementary Data 2). Similarly, knockdown of RNF4, but not TOPORS, reduced arsenic-induced K63-linked ubiquitylation of PML (Extended Data Fig. 5g). Thus, TOPORS and RNF4 both contribute to recruiting p97 to SUMOylated targets by promoting their K48-linked ubiquitylation, and RNF4 may further diversify the ubiquitin linkage landscape on these substrates to augment p97 accumulation[44].

### TOPORS is a SUMO1-selective STUbL

STUbL substrates including DPCs and PML are extensively modified by both SUMO1 and SUMO2/3 (Figs. 2b and 3a). Because SUMO1 only shares around 50% sequence identity with SUMO2/3 (which are nearly identical)[10], we speculated that the requirement for both TOPORS and RNF4 in STUbL pathways could involve a differential preference for targeting SUMO1 and SUMO2/3 substrates. Interestingly, probing total ubiquitylated cellular proteins isolated by the MultiDsk affinity reagent[45] under denaturing conditions to prevent co-purification of associated factors revealed that loss of TOPORS, but not RNF4, led to a notable decline in proteins co-modified by ubiquitin and SUMO1, whereas levels of ubiquitylated proteins modified by SUMO2/3 were moderately increased (Fig. 4e and Supplementary Fig. 2). This suggests that TOPORS may preferentially act on and provide a principal source of STUbL activity toward SUMO1-modified proteins, in keeping with the modest STUbL activity of TOPORS toward 4×SUMO2 substrates in vitro (Fig. 2k and Extended Data Fig. 3c). To test this, we employed previously described COMP–SUMO model substrates fusing SUMO to a COMP pentamerization domain[46], resembling multi-mono-SUMO1-modified or multi-mono-SUMO2/3-modified proteins (Fig. 4f). Because SUMO1 engages in homotypic chain formation to a much lesser extent than SUMO2/3 (ref. 1), COMP–SUMO1 may more faithfully recapitulate cellular SUMO1-modified substrates than linear poly-SUMO1 fusions. Using these substrates, we found that purified TOPORS displays markedly higher ubiquitin ligase activity toward COMP–SUMO1 than a corresponding COMP–SUMO2 substrate and monomeric SUMO1 in vitro (Fig. 4f,g). Moreover, although TOPORS efficiently binds poly-SUMO2 chains in vitro, we observed using agarose-conjugated SUMO1 and SUMO2 that it displays some preference for interacting with SUMO1, whereas RNF4 showed an inverse selectivity, in line with previous findings (Figs. 2i and 4h,i)[35,46]. In cells, RNF4 and SUMO1 depletion had additive impacts on reducing DNMT1 DPC ubiquitylation and resolution, mimicking the effect of combined RNF4 and TOPORS loss (Fig. 2c and Extended Data Figs. 2d and 5h,i). By contrast, such a relationship was not apparent for RNF4 and SUMO2/3 co-depletion (Extended Data Fig. 5h). Together, these data suggest that the key contribution of TOPORS to STUbL pathways is exerted via its preference for targeting SUMO1-modified proteins. The combined TOPORS and RNF4 STUbL activities may, thus, generate complex ubiquitin landscapes on both SUMO1-modified and SUMO2/3-modified proteins to facilitate optimal processing by the p97-proteasome pathway.

### Combined loss of TOPORS and RNF4 is synthetic lethal

To further interrogate the functions of RNF4 and TOPORS and their interrelationship, we profiled genetic interactions impacting fitness in HAP1 cell lines lacking TOPORS or RNF4 (Extended Data Figs. 1c and 6a,b). Because genetic screens in unchallenged HAP1 WT cells indicated a long-term fitness defect associated with KO of RNF4 but not TOPORS

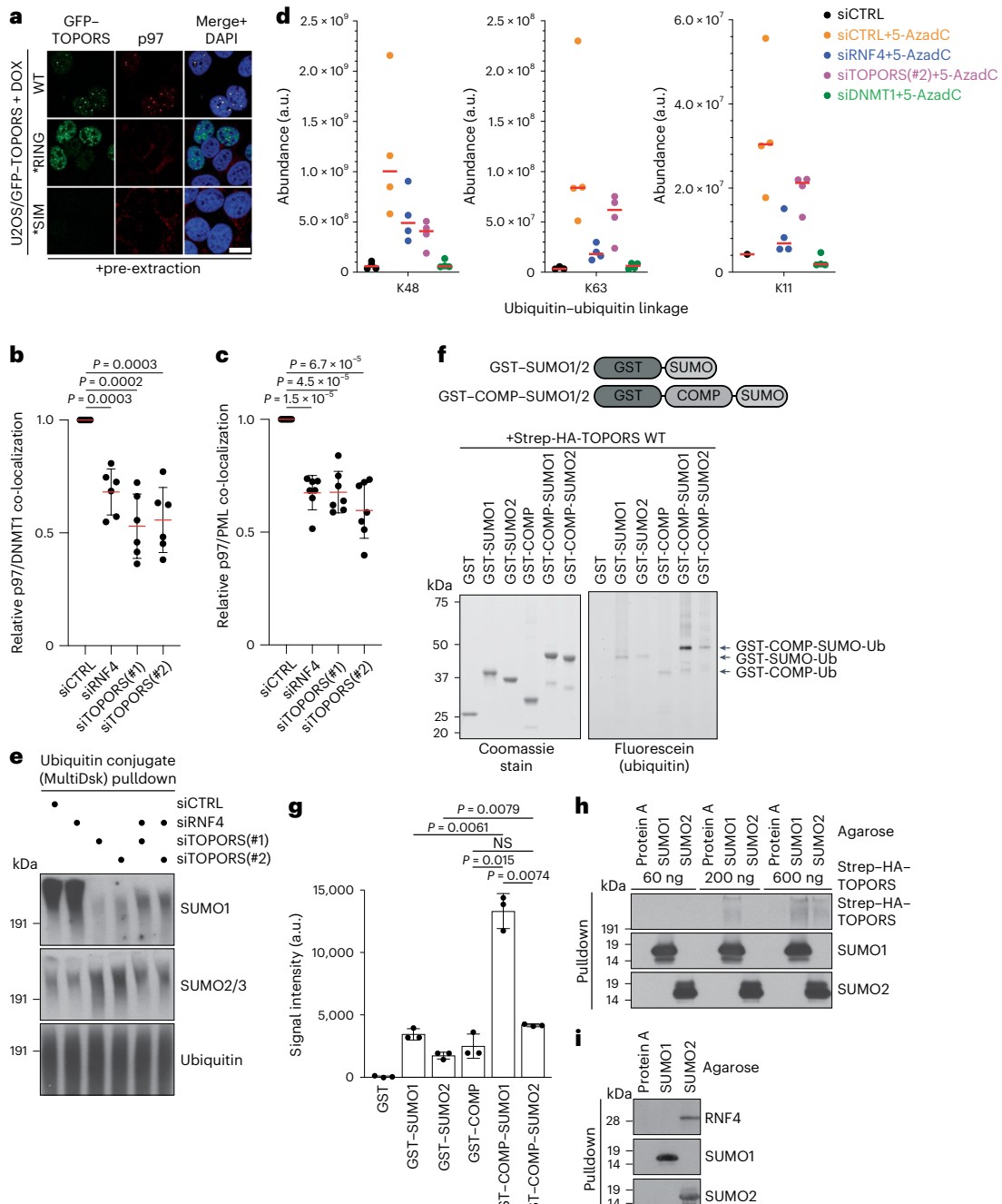

**Fig. 4 | TOPORS and RNF4 generate complex ubiquitin topologies on STUbL substrates to promote p97 recruitment. a**, Representative images of doxycycline (DOX)-treated U2OS/GFP–TOPORS cell lines immunostained with p97 antibody after pre-extraction and fixation. Scale bar, 10 μm. **b**. U2OS cells transfected with indicated siRNAs were released from single thymidine synchronization in early S phase, treated with 5-AzadC for 30 min, pre-extracted and immunostained with p97 and DNMT1 antibodies. p97 and DNMT1 intensity in DNMT1 DPC foci was analyzed by quantitative image-based cytometry (QIBC) (mean ± s.e.m.; *n* = 6 independent experiments; one-tailed paired *t*-test). **c**. U2OS PML-KO/YFP–PML cells transfected with indicated siRNAs for 72 h and treated with arsenic for 1 h were pre-extracted and immunostained with p97 antibody. p97 and YFP intensity in YFP-PML bodies was analyzed by quantitative image-based cytometry (QIBC) (mean ± SEM; *n* = 7 independent experiments; one-tailed paired *t*-test). **d**, U2OS cells transfected with indicated siRNAs were subjected to DNMT1 IP under denaturing conditions to isolate DNMT1 but not associated proteins. Samples were digested with trypsin, and di-glycine remnants on ubiquitin were identified by mass spectrometry (*n* = 4 independent experiments). **e**, Immunoblot analysis of U2OS cells transfected with indicated

siRNAs for 40 h and subjected to MultiDsk pulldown under denaturing conditions to isolate total cellular ubiquitin conjugates but not associated, non-covalently bound proteins. Input blots are shown in Fig. 3f. See also Supplementary Fig. 2. **f**, Coomassie staining (left) and fluorescein-ubiquitin visualization (right) of in vitro STUbL reactions containing purified Strep–HA–TOPORS and indicated GST–SUMO protein substrates (top), supplemented with E1 (UBA1) and E2 (UBE2D1) enzymes, ubiquitin (10% labeled with 5-IAF) and ATP. **g**, Quantification of data in **f** (mean ± s.d.; *n* = 3; NS, not significant; two-tailed paired *t*-test). **h**, Indicated amounts of recombinant Strep–HA–TOPORS *RING protein was incubated with SUMO1-conjugated, SUMO2-conjugated or Protein A–conjugated agarose beads. Beads were washed extensively, and bound material was immunoblotted with HA, SUMO1 and SUMO2/3 antibodies. **i**, Whole-cell extract from U2OS cells was incubated with SUMO1-conjugated, SUMO2-conjugated or Protein A–conjugated agarose beads. Beads were washed extensively, and bound material was immunoblotted with RNF4, SUMO1 and SUMO2/3 antibodies. Data information: data are representative of three (**a,e,f,h,i**) independent experiments with similar outcomes.

(Fig. 5a)[47], we endogenously tagged RNF4 with a C-terminal degron (dTAG-HA) enabling its efficient conditional depletion upon addition of the dTAG-13 degrader (Extended Data Fig. 6b). TOPORS-KO and dTAG-13-dependent RNF4 depletion in these cell lines impaired DNMT1 DPC resolution in S phase, as expected (Fig. 1c and Extended Data Fig. 6c). Intriguingly, fitness-based gene-trap mutagenesis screens in TOPORS-KO cells identified *RNF4* as a major synthetic lethal hit, and disrupting *TOPORS* gave rise to a strong synthetic lethality effect in RNF4-deficient cells (Fig. 5a–d and Supplementary Data 4). Underscoring the critical role of USP7 activity in sustaining TOPORS expression, *USP7* was also a strong synthetic lethality hit in cells lacking RNF4 but not TOPORS (Fig. 5a–d). We validated a profound loss of proliferative potential resulting from co-depleting TOPORS and RNF4 in U2OS cells (Fig. 5e and Extended Data Fig. 6d). Likewise, USP7i strongly suppressed proliferation of RNF4-deficient, but not TOPORS-KO, cells (Extended Data Fig. 6e–g). Co-depleting TOPORS and RNF4, but not either individual ligase, caused prominent induction of PARP1 cleavage, an apoptosis marker (Fig. 5f), suggesting that joint loss of TOPORS and RNF4 is incompatible with cell survival. USP7i addition to RNF4-depleted cells produced a similar effect (Extended Data Fig. 6h). These findings demonstrate that the combined actions of TOPORS and RNF4 are essential for cell viability and proliferation, corroborating the critical importance of concerted TOPORS and RNF4 STUbL activities in promoting the turnover of SUMOylated proteins.

To understand how simultaneous loss of TOPORS and RNF4 undermines cell proliferation, we performed live-cell imaging experiments using a Fucci cell line to track individual cell cycle phases via fluorescent reporters[48]. Co-depletion of TOPORS and RNF4 induced extensive cell death, as expected, and this predominantly occurred around early S phase (Extended Data Fig. 7a). In line with this, a strong accumulation of cells at the G1/S transition and a concomitant loss of S/G2 phase cells was apparent when both TOPORS and RNF4, but not either ligase alone, were depleted (Fig. 5g). Again, this effect was phenocopied by USP7i treatment of RNF4-depleted cells (Extended Data Fig. 7b). These observations suggest that combined RNF4 and TOPORS loss gives rise to a severe DNA replication defect. To test this prediction, we performed 5-ethynyl-2-deoxyuridine (EdU) labeling experiments with cells released from G1/S transition arrest by double thymidine block. Although individual depletion of TOPORS or RNF4 had little impact on EdU incorporation efficiency, cells lacking both TOPORS and RNF4 displayed markedly reduced DNA synthesis rates in early S phase and failed to sustain replication further into S phase (Extended Data Fig. 7c). RNF4-depleted cells exposed to USP7i likewise exhibited diminished EdU incorporation rates (Extended Data Fig. 7d). The DNA replication defect arising from combined TOPORS and RNF4 knockdown coincided with DNA damage accumulation as evidenced by increased γH2AX formation in S phase cells (Fig. 5h), suggesting that simultaneous loss of TOPORS and RNF4 leads to extensive replication fork stalling and/or collapse. We reasoned that this DNA replication defect might result from obstacles to replisome progression conferred by SUMOylated proteins that fail to get extracted from chromatin, in keeping with the known role of STUbL activity in this process[3]. Indeed, combined knockdown of TOPORS and RNF4 caused strong accumulation of SUMOylated proteins on chromatin and a concomitant decrease in ubiquitylation, with TOPORS depletion alone triggering a substantial increase in SUMO1-modified chromatin-bound proteins (Fig. 5i). Collectively, these data suggest that combined TOPORS and RNF4 STUbL activities are not only instrumental for SUMO-dependent stress responses but also exert an essential role during unperturbed cell proliferation by ubiquitylating chromatin-bound SUMOylated proteins, driving their displacement to allow unhindered advancement of the replication machinery.

## Discussion

By catalyzing selective ubiquitylation of SUMOylated proteins, STUbLs enable direct crosstalk between ubiquitin- and SUMO-mediated signaling in cells, acting as key effectors of stress responses whose pharmacological modulation has shown strong potential in cancer treatment. A detailed understanding of the mechanisms and functions of STUbLs in cell biology is, therefore, of considerable importance. In the present study, we discovered that human TOPORS is a STUbL by virtue of its RING domain and poly-SUMO-binding SIMs, a configuration analogous to that of other STUbLs, such as RNF4 and RNF111. The precise enzymatic function of TOPORS has been uncertain, as previous studies provided evidence for TOPORS functioning as an E3 ligase for both ubiquitin and SUMO[25–29]. Although we do not exclude that TOPORS may serve as a SUMO E3 ligase in some contexts, our collective data show that TOPORS is more active as a ubiquitin ligase and argue that this function provides the critical activity underlying its role in STUbL-dependent processes.

We found that TOPORS makes key contributions to established RNF4-driven pathways, including SUMO-dependent DPC resolution and PML body degradation. Rather than simply serving as a backup to RNF4, we demonstrated that TOPORS and RNF4 are both required for the efficiency of STUbL reactions. Notably, the dedicated partnership between TOPORS and RNF4 not only underpins the efficacy of SUMO-driven stress responses but is also essential for cell proliferation and viability under normal conditions. Accordingly, in the absence of both TOPORS and RNF4, cells accrue high levels of hyper-SUMOylated proteins, presumably due to failure in processing these species via the p97-proteasome pathway caused by their lack of ubiquitylation. This is accompanied by a strong DNA replication defect that may result from the inability to properly displace SUMOylated proteins from chromatin, obstructing progression of the replication machinery. Indeed, we found that combined TOPORS and RNF4 loss disrupts the normal balance of SUMO and ubiquitin modifications on chromatin, which has been proposed to be critical for maintaining the activity of replication forks[49]. Given that STUbL activity is important for removing both DPCs and non-covalently bound proteins from DNA[3,12,15], we consider it likely that SUMOylated forms of both classes of proteins

---

**Fig. 5 | Combined loss of TOPORS and RNF4 is synthetic lethal. a**, Haploid genetic fitness screen in HAP1 WT cells, presented as a fishtail plot in which genes are plotted according to ratio of sense/anti-sense orientation of gene-trap insertions (*y* axis) and the total number of insertions in a particular gene (*x* axis) (*n* = 4 biological replicates (independent clones)). Blue dots represent genes with significant sense bias (binomial *P* < 0.05, FDR corrected, across all replicates). A single representative replicate is shown. **b**, As in **a** but for HAP1 TOPORS-KO cells (*n* = 2 biological replicates (independent clones)). **c**, As in **a** but for HAP1 RNF4−dTAG−HA cells treated with 0.25 μM dTAG-13 for 10 d (*n* = 2 biological replicates (independent clones)). **d**, Gene rank plot for significant synthetic lethal genes for HAP1 TOPORS-KO (left) and RNF4−dTAG−HA (right) compared to HAP1 WT (Fisher's exact test; *P* < 0.05; odds ratio < 0.8). The size of the dot represents the difference (Δ) in sense ratio bias. **e**, Normalized logarithmic proliferation quantification for U2OS cells transfected with indicated siRNAs for the indicated times, as determined by IncuCyte image-based confluence analysis (mean ± s.e.m.; *n* = 3 independent experiments). Imaging was started at 24 h after siRNA transfection. **f**, Immunoblot analysis of U2OS cells transfected with indicated siRNAs for 48 h. **g**, U2OS Fucci cells were transfected with indicated siRNAs for 36 h. To determine cell cycle position, CDT1 (mKO2-hCdt1⁺) and Geminin (mAG-hGem⁺) intensities were analyzed by quantitative image-based cytometry (QIBC). Quantification of cells in different cell cycle phases was done using the indicated gates. **h**, U2OS cells were transfected with indicated siRNAs, treated with thymidine for 18 h and released into S phase for 6 h. Cells were immunostained with γH2AX antibody and analyzed by QIBC (red bars, mean; >3,900 cells analyzed per condition). **i**, Immunoblot analysis of chromatin-enriched fractions of U2OS cells transfected with indicated siRNAs for 40 h. Data information: data are representative of three (**f**,**i**) and two (**g**,**h**) independent experiments with similar outcomes.

accumulate on chromatin and contribute to impeding replication when both TOPORS and RNF4 are absent. Our findings thus reveal an unexpected mechanistic complexity of major STUbL-driven processes, suggesting that the concerted action of TOPORS and RNF4 is a general modus operandi of these pathways with indispensable roles in cell proliferation.

A key question emanating from our discoveries is why both TOPORS and RNF4 are needed for productive execution of

STUbL-mediated responses. Together with other findings, the notion that combined RNF4 and TOPORS loss is lethal, whereas their individual depletion is not, suggests that these STUbLs have both overlapping and non-redundant roles in ubiquitylating SUMOylated proteins. According to this scenario, STUbL-driven pathways become inefficient but retain basic functionality in the absence of either ligase, whereas loss of both TOPORS and RNF4 leads to profound defects in ubiquitylation and proteolytic processing of SUMOylated proteins. Our collective

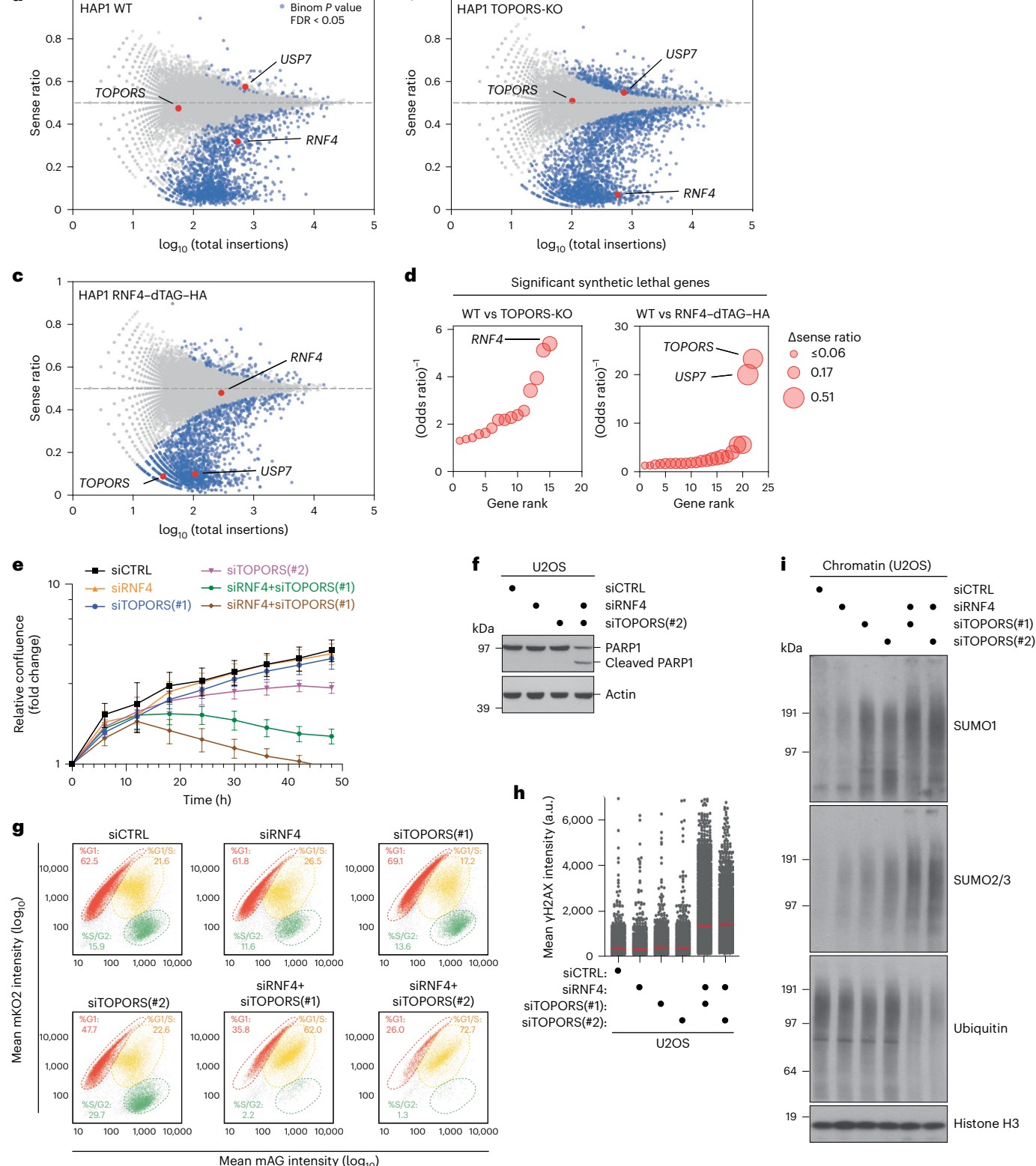

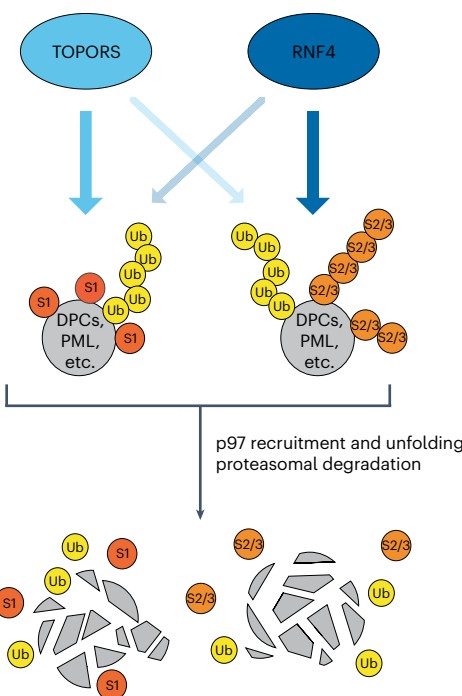

**Fig. 6 | Concerted action of TOPORS and RNF4 drives major STUbL-mediated processes—model.** Parallel E3 ubiquitin ligase activities of TOPORS and RNF4 provide a key driving force for major STUbL-mediated cellular processes, including SUMO-dependent DPC repair and PML body degradation. This involves a selectivity of TOPORS for targeting SUMO1-modified proteins, whereas RNF4 may have some preference for SUMO2/3-modified proteins. However, these preferences are not absolute. Accordingly, in the absence of either ligase, STUbL-driven pathways become inefficient but retain basic functionality, whereas combined loss of TOPORS and RNF4 leads to a profound defect in the ubiquitylation and proteolytic processing of SUMOylated proteins, accompanied by synthetic lethality.

evidence supports a division of labor between TOPORS and RNF4, enabling efficient generation of a complex ubiquitin landscape on SUMO substrates to optimally support p97 recruitment (Fig. 6). Indeed, both TOPORS and RNF4 make sizeable contributions to the modification of SUMOylated DNMT1 DPCs by K48-linked ubiquitin chains, a primary recognition signal for p97 cofactor complexes[43], and RNF4 appears to be the main effector of non-K48-linked ubiquitylation of STUbL substrates that may further enhance p97 recruitment[44], consistent with its ability to catalyze K63-linked ubiquitin chain formation[50]. Perhaps more importantly, we found that TOPORS is crucial for ubiquitylating SUMO1-modified proteins in cells and displays a clear preference for these substrates in vitro. Correspondingly, RNF4 may be particularly efficient in targeting SUMO2/3-modified proteins, based on our observation and previous work indicating a preference for binding to SUMO2/3 relative to SUMO1 (refs. [35],[46]). Indeed, we found that RNF4 loss has little impact on levels of cellular proteins co-modified by SUMO1 and ubiquitin. Thus, a major underlying reason for the critical importance of combined TOPORS and RNF4 activities for the integrity of STUbL pathways may be their selectivity for targeting SUMO1-modified and SUMO2/3-modified proteins, respectively (Fig. 6). Such complementary preferences of TOPORS and RNF4 in recognizing and/or modifying particular SUMO modifications on substrates could at least partially reflect the distinct configuration of their multiple SIMs. Poly-SUMO2/3 chain binding by the tandem SIMs in RNF4 converts inactive monomeric RNF4 into an active dimer[6], and it is conceivable that this could be difficult to achieve by multiple mono-SUMO1 modifications, unless they are in close proximity. With

the more scattered distribution of its SIMs, TOPORS may be better configured to ubiquitylate proteins modified by multi-mono-SUMO1 modifications. Addressing the precise relationships between individual STUbLs and different SUMO modification configurations will be important but is challenged by limited current insights into these topologies and a shortage of tools and methods for analyzing specific SUMO polymer architectures[1]. This notwithstanding, it seems likely that the concerted actions of TOPORS and RNF4 enables flexibility in STUbL-driven processes targeting multitudinous cellular substrates and that the relative importance of TOPORS and RNF4 STUbL activities may depend on the extent to which individual SUMOylated targets are modified by SUMO1 and SUMO2/3.

Altogether, our findings uncover TOPORS as a novel STUbL with a preference for SUMO1-modified proteins and establish that direct SUMO-ubiquitin crosstalk mediated by parallel TOPORS and RNF4 activities is essential for viability and proliferation of human cells. These improved insights into the mechanistic basis of important SUMO-driven processes could have considerable bearings on current, promising treatment strategies targeting the SUMO system in cancers and could offer potential new opportunities for therapeutic intervention. Pharmacological modulation of STUbL-dependent processes, including arsenic-induced PML body degradation and resolution of 5-AzadC-induced DPCs, has documented clinical benefits[19,20]. Moreover, many cancers are addicted to a hyperactive SUMO system as a means to mitigate high levels of stress, and the specific SUMO E1 inhibitor TAK-981 has emerged as a promising anti-cancer agent, synergizing with immune checkpoint inhibitors in mice[51]. Targeting the critical role of TOPORS in STUbL-driven responses might provide a more specific, less toxic alternative to full inhibition of SUMOylation in some clinical contexts. In this regard, harnessing the notion that USP7 inhibitors, several of which display efficacy for cancer therapy in preclinical studies[52], potently deplete TOPORS and recapitulate the consequences of TOPORS loss due to a critical role of USP7 in antagonizing TOPORS auto-ubiquitylation and preventing its proteasomal degradation may be particularly useful. Further studies of TOPORS mechanisms of action and functions are, therefore, well warranted.

## Online content

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

## Methods

### Cell culture

Human HeLa (CCL-2) and U2OS (HTB-96) cells were obtained from the American Type Culture Collection. HEK293-EBNA1-6E cells were a kind gift from Yves Durocher (National Research Council Canada). HeLa, U2OS and HEK293-6E cells were propagated in DMEM supplemented with 10% (v/v) FBS and 1% penicillin–streptomycin. HAP1 cells[53] were cultured in IMDM (Gibco) supplemented with 10% heat-inactivated FCS (Thermo Fisher Scientific) and penicillin–streptomycin–glutamine solution (Gibco). HeLa cells stably expressing GFP–DNMT1 were described previously[11]. To generate inducible cell lines expressing WT and mutant GFP–TOPORS proteins, pcDNA5/FRT/TO/GFP/TOPORS plasmids were co-transfected with pOG44 into U2OS Flp-In T-Rex cells (Thermo Fisher Scientific) using Lipofectamine 3000 (Invitrogen). Clones were selected in medium containing hygromycin B (Thermo Fisher Scientific) and blasticidin (InvivoGen) and verified by microscopy. All cell lines were regularly tested negative for mycoplasma infection and were not authenticated.

Plasmid DNA transfections were performed using FuGENE 6 (Promega), Lipofectamine 2000 (Invitrogen) or Lipofectamine 3000 (Invitrogen) according to the manufacturers' protocols. Cell cycle synchronizations were performed as previously described[54], using single treatment with thymidine for 18 h. Unless otherwise indicated, the following doses of chemicals and genotoxic agents were used: 5-AzadC (10 μM, Sigma-Aldrich), EdU (20 μM, Sigma-Aldrich), arsenic trioxide (arsenic; 1 μM, Sigma-Aldrich), formaldehyde (500 μM, Thermo Fisher Scientific), camptothecin ((CPT); 10 μM; AH Diagnostics); FT-671 (USP7i; 2 μM, MedChemExpress), MG132 (20 μM, Sigma-Aldrich), ML-792 (SUMOi; 2 μM, MedKoo Biosciences), MLN-7243 (Ub-E1i; 5 μM, Active Biochem), NMS-873 (p97i; 5 μM, Sigma-Aldrich), dTAG-13 (0.25 μM, Tocris Biosciences) and thymidine (2 mM, Sigma-Aldrich).

### Plasmids

Full-length cDNAs encoding human TOPORS WT (codon optimized, Invitrogen GeneArt), *RING (I105A,L140A,K142A; codon optimized) and TOPORS *SIM (V391A,I392A,I394A,V479A,I480A,V481A,V484A,L494A,V495A,L497A,V906A,V907A,I908A,I910A; codon optimized) were PCR amplified and cloned into pcDNA5/FRT/TO/GFP via KpnI and NotI (for generation of stable cell lines) or pcDNA4/TO/Strep–HA via EcoRV and NotI (for protein expression).

### siRNAs

siRNA transfections were performed using Lipofectamine RNAiMAX (Invitrogen) according to the manufacturer's instructions. All siRNAs were used at a final concentration of 20 nM. The following siRNA oligonucleotides were used: non-targeting control (CTRL): 5′-GGGAUACCUAGACGUUCUA-3′; RNF4: 5′-GAAUGGACGUCUCAUCGUU-3′; TOPORS(#1): 5′-GUCCUAAGGCCUUCGUAUAAU-3′; TOPORS(#2): 5′-CCCUGCUCCUUCAUACGAA-3′; SUMO1 (5′-GGACAGGAUAGCAGUGAGA-3′); SUMO2/3 (5′-GUCAAUGAGGCAGAUCAGA-3′); UBE2K(#1): 5′-GAAUCAAGCGGGAGUUCAA-3′; UBE2K(#2): 5′-CCUAAGGUCCGGUUUAUCA-3′; UBE2K(#3): 5′-CCAGAAACAUACCCAUUUA-3′ and UBE2K(#4): 5′-GCAAAUCAGUACAAACAAA-3′ (a 1:1:1:1 mix of all four UBE2K siRNAs was used).

### Immunoblotting, immunoprecipitation and chromatin fractionation

Immunoblotting was performed as previously described[55]. To prepare cell extracts, cells were lysed in EBC buffer (50 mM Tris, pH 7.5; 150 mM NaCl; 1 mM EDTA; 0.5% NP40; 1 mM DTT) or MultiDsk lysis buffer (50 mM Tris, pH 8.0; 1 M NaCl; 5 mM EDTA; 1% NP40; 0.1% SDS) supplemented with protease and phosphatase inhibitors on ice for 20 min, and lysates were cleared by centrifugation (21,000g, 20 min). For immunoprecipitation (IP) experiments to analyze protein modification by SUMO and ubiquitin, cells were lysed in denaturing buffer

(20 mM Tris, pH 7.5; 50 mM NaCl; 1 mM EDTA; 0.5% NP40; 0.5% SDS; 0.5% sodium deoxycholate; 1 mM DTT) supplemented with protease and phosphatase inhibitors, followed by sonication. For co-IP experiments, cells were lysed in EBC buffer supplemented with protease and phosphatase inhibitors on ice for 20 min. Lysates were then cleared by centrifugation (21,000g, 20 min) and incubated with GFP-Trap or DNMT1-Trap agarose (ChromoTek) overnight at 4 °C. After washing, immobilized proteins were eluted from the beads by boiling in 2× Laemmli sample buffer for 5 min. For two-step IPs, GFP-tagged DNMT1 was purified on GFP-Trap agarose (ChromoTek) under denaturing conditions as described above and washed extensively in denaturing buffer. Beads were equilibrated in EBC buffer and incubated with whole-cell lysate prepared in EBC buffer at 4 °C for 4 h. Beads were then washed in EBC buffer, and proteins were eluted by boiling in 2× Laemmli sample buffer for 5 min.

For pulldowns of total cellular ubiquitin conjugates using the MultiDsk affinity reagent[45], purified Halo-tagged MultiDsk (15 μg per condition) was pre-incubated with HaloLink resin (Promega) for 1 h at room temperature in binding buffer (100 mM Tris, pH 7.5; 150 mM NaCl; 0.05% IGEPAL). Excess protein was removed by washing with binding buffer supplemented with 1 mg ml⁻¹ BSA. Cells were lysed in MultiDsk buffer supplemented with protease and phosphatase inhibitors on ice for 15 min, followed by sonication. Lysates were then cleared by centrifugation (21,000g, 20 min) and incubated with Halo-MultiDsk resin overnight at 4 °C. After multiple washes of beads, ubiquitylated proteins were eluted by boiling in 2× Laemmli sample buffer for 5 min.

For chromatin fractionation, cells were first lysed in buffer 1a (10 mM Tris, pH 8.0; 10 mM KCl; 1.5 mM MgCl₂; 0.34 M sucrose; 10% glycerol; 0.1% Triton X-100) supplemented with protease and phosphatase inhibitors on ice for 5 min, followed by centrifugation (2,000g, 5 min) to recover the soluble proteins. Pellets were then washed once in buffer 1b (buffer 1a supplemented with 500 mM NaCl) and once in buffer 1a, followed by resuspension in buffer 2 (50 mM Tris, pH 7.5; 150 mM NaCl; 1% NP40; 0.1% SDS; 1 mM MgCl₂; 125 U ml⁻¹ benzonase) supplemented with protease and phosphatase inhibitors. Lysates were incubated in a thermomixer (37 °C, 1,000 r.p.m., 15 min), and solubilized chromatin-bound proteins were obtained by centrifugation (21,000g, 10 min).

### Antibodies

Antibodies to human proteins used in this study included: actin (1:20,000 dilution (MAB1501, Millipore, RRID: AB_2223041)); DNMT1 (1:1,000, described in ref. [11]); FLAG (1:1,000 (A00187, GenScript, RRID: AB_1720813)); GFP (1:5,000, Abcam (ab6556, RRID: AB_305564)); HA (1:1,000 (11867423001, Roche, RRID: AB_390918)); histone H3 (1:20,000, Abcam (ab1791, RRID: AB_302613)); PARP1 (1:1,000, Santa Cruz Biotechnology (sc-8007, RRID: AB_628105)); PIAS1 (1:1,000 (ab77231, Abcam, RRID: AB_1524188)); RNF4 (1:5,000, described in ref. [54]); SUMO1 (1:1,000, Thermo Fisher Scientific (33-2400, RRID: AB_2533109)); SUMO2/3 (1:1,000, Abcam (ab3742, RRID: AB_304041); 1:1,000, Abcam (ab81371, RRID: AB_1658424)); TOP1 (1:500, Bethyl (A302-590A, RRID: AB_2034875)); TOPORS (1:250, sheep polyclonal raised against full-length human TOPORS); ubiquitin (1:1,000, Santa Cruz Biotechnology (sc-8017 AC, RRID: AB_2762364); 1:1,000, Millipore (04-263, RRID: AB_612093)); ubiquitin (K48-linked) (1:1,000, Millipore (05-1307, RRID: AB_1587578)); ubiquitin (K63-linked) (1:1,000, (05-1308, RRID: AB_1587580)); UBE2K (1:1,000, Cell Signaling Technology (3847, RRID: AB_2210768)); and USP7 (1:1,000, Bethyl (A300-033A, RRID: AB_203276)).

### Immunofluorescence and high-content image analysis

Cells were pre-extracted on ice in stringent pre-extraction buffer (10 mM Tris-HCl, pH 7.4; 2.5 mM MgCl₂; 0.5% NP-40; 1 mM PMSF) for 8 min and then in ice-cold PBS for 2 min before fixation with 4% formaldehyde for 15 min. If not pre-extracted, cells were subjected to

permeabilization with PBS containing 0.2% Triton X-100 for 5 min before blocking. Coverslips were blocked in 10% BSA and incubated with primary antibodies for 2 h at room temperature, followed by staining with secondary antibodies and DAPI (Alexa Fluor, Life Technologies) for 1 h at room temperature. Coverslips were mounted in Mowiol 4-88 (Sigma-Aldrich).

Manual image acquisition was performed with a Zeiss LSM 880 laser scanning confocal microscope, using a Plan-Apochromat ×40 1.30 oil DIC M27 objective and ZEN software (Zeiss). Coverslips were prepared as described above, except that they were mounted with VECTASHIELD Mounting Medium (Vector Laboratories) and sealed with nail polish. Raw images were exported as TIFF files and processed using Adobe Photoshop. If adjustments in image contrast and brightness were applied, identical settings were used on all images of a given experiment. For automated image acquisition, images were acquired with an Olympus Ixplore ScanR system equipped with an Olympus IX-83 wide-field microscope, a Yokogawa CSU-W1 confocal spinning disk unit, four 50/100-mW laser diodes, UPLSXAPO20× NA 0.80 WD 0.60 mm or UPLSXAPO40×2 NA 0.95 WD 0.18 mm and a Hamamatsu Orca Flash 4 sCMOS camera. Automated and unbiased image analysis was carried out with ScanR analysis software. Data were exported and processed using Spotfire (TIBCO Software).

### Live-cell imaging

U2OS cells stably expressing Fucci reporter constructs (mKO2-hCdt1(30-120) and mAG-hGem(1-110))[48] were transfected with siRNAs and seeded onto an ibidi dish 16 h before acquisition. Medium was changed to Leibovitz's L15 medium (Life Technologies) supplemented with 10% FBS before filming. Live-cell imaging was performed on a DeltaVision Elite system using a ×40 oil objective (GE Healthcare). Images were acquired every 10 min or 15 min for 48 h. Three z-stacks of 5 μm were imaged. SoftWork software (GE Healthcare) were used for data analysis.

### Proliferation and survival assays

Relative proliferation was measured using an IncuCyte (Sartorius) instrument. Cells ($6 \times 10^4$) were seeded into 24-well or 96-well plates 24 h after siRNA treatment and imaged every 2 h. Mean confluency was determined from four (Fig. 5e) or nine (Extended Data Fig. 6f,g) images. For survival assays, approximately 400 cells were plated per 60-mm plate and treated with 5-AzadC for 24 h or formaldehyde for 30 min. Cells were then washed with PBS twice, and fresh medium was added. Colonies were fixed and stained after approximately 2 weeks with cell staining solution (0.5% w/v crystal violet, 25% v/v methanol). The number of colonies was quantified using a GelCount (Oxford Optronix) colony counter.

### Purification of recombinant proteins

For purification of Strep–HA–TOPORS proteins, HEK293-6E suspension cells were cultured in FreeStyle F17 expression medium (Gibco) supplemented with 4 mM L-glutamine (Gibco), 1% FBS, 0.1% Pluronic F-68 (Gibco) and 50 μg ml$^{-1}$ G418 (Invivogen) in a 37 °C incubator on a shaker rotating at 140 r.p.m. HEK293-6E cells were transfected with Strep–HA–TOPORS WT, *SIM or *RING expression plasmids using PEI transfection reagent (PolyScience). After 24 h, cells were harvested and snap frozen in liquid nitrogen. The cell pellet was resuspended in lysis buffer (100 mM Tris, pH 8.0; 350 mM NaCl; 0.05% NP-40; protease inhibitors), treated with benzonase and sonicated. The lysate was then cleared by centrifugation at 25,000g at 4 °C for 30 min. The cleared lysate was incubated with Strep-Tactin Superflow resin (IBA Lifesciences) and incubated at 4 °C for 2 h. Beads were washed in a gravity column using wash buffer (100 mM Tris, pH 8.0; 600 mM NaCl; 0.05% NP-40), and TOPORS protein was eluted using elution buffer (100 mM Tris, pH 8.0; 350 mM NaCl; 2.5 mM desthiobiotin). The elute fractions were run on a 4–12% NuPAGE Bis-Tris protein gel (Invitrogen)

and stained with Instant Blue Coomassie Protein Stain (Expedeon). Fractions were concentrated on Microcon-30kDa Centrifugal Filters (Millipore), snap frozen in liquid nitrogen and stored at −70 °C. For in vitro ubiquitylation and SUMOylation reactions, 0.35 μg of Strep–HA–TOPORS prep was used; for Strep-Tactin pulldowns, 0.7 μg was used; and for STUbL assays, 2.8 μg was used.

Purification of GST–SUMO proteins was described previously[56]. Plasmids for recombinant expression of GST–COMP, GST–COMP–SUMO1 and GST–COMP–SUMO2 (ref. [46]) were a kind gift from Andrew Sharrocks (University of Manchester). The same procedure of expression and purification was followed as for GST–SUMO proteins, except that Rosetta (DE3) BL21 *Escherichia coli* was used and proteins were not eluted from the glutathione agarose resins before use in assays.

### In vitro ubiquitylation, SUMOylation and deubiquitylation reactions

All in vitro ubiquitylation and SUMOylation reactions were carried out in reaction buffer containing 50 mM Tris, pH 7.5; 150 mM NaCl; 0.1% NP-40; 5 mM MgCl₂; 0.5 mM TCEP. Unless otherwise stated, the following final concentrations were used: FLAG ubiquitin (R&D Systems, 20 μM), ATP (Sigma-Aldrich, 3 mM), UBA1 (100 nM), UBE2D1 (0.5 μM), UBE2K (R&D Systems, 0.28 μM), SAE1-UBA2 (R&D Systems, 100 nM), UBE2I (R&D Systems, 0.5 μM), SUMO2 (R&D Systems, 2 μM), poly-SUMO2$_{2-8}$ chains (R&D Systems, 0.3 μg per reaction), RNF4 (47 nM), GST–PIAS1 (Enzo Life Sciences), 4×SUMO2 and Ub-4×SUMO2 (1.8 μM)[36]. HA–SUMO1 vinyl sulfone (VS) and HA–ubiquitin VS (R&D Systems) were added to all reactions containing purified Strep–HA–TOPORS proteins to inhibit deubiquitinase and SUMO protease activities co-purifying with Strep–HA–TOPORS.

For GST–SUMO and GST–COMP–SUMO ubiquitylation assays, GST proteins immobilized on glutathione agarose were incubated at 22 °C with agitation for 1 h with Strep–HA–TOPORS (0.6 μM), His₆-UBE1 (0.1 μM), UBE2D1 (1 μM) and 40 μM ubiquitin (10% labeled with 5-IAF) in reaction buffer supplemented with 2 mM ATP. Beads were washed twice with reaction buffer and transferred to a new tube, and bound proteins were eluted with 30 μl of 1× Laemmli sample buffer. Fluorescein-ubiquitin species on gel were visualized by laser scanning (Typhoon, Cytiva Life Sciences) using the Cy2 setting before staining with Coomassie blue. Data for ubiquitin-modified GST substrates from triplicate reactions were quantified by densitometry (ImageJ) from non-saturated negative scans.

For in vitro deubiquitylation reactions with recombinant USP7, GFP–TOPORS expressed in U2OS cells was immunoprecipitated using GFP-Trap agarose under denaturing condition as above, followed by extensive washing. Beads containing bound GFP–TOPORS were equilibrated in deubiquitylation buffer (50 mM Tris, pH 7.5; 150 mM NaCl; 5 mM DTT) and incubated with 0.5 μg of recombinant USP7 protein (Ubiquigent) with shaking at 30 °C for 30 min. Beads were washed, eluted by boiling in 2× Laemmli sample buffer for 5 min and analyzed by immunoblotting.

### SUMO-binding assays

Human recombinant SUMO1 or SUMO2 coupled to agarose at 0.5 mg of protein per milliliter of settled resin (Enzo Life Sciences) was incubated with whole-cell extracts from U2OS cells or recombinant Strep–HA–TOPORS *RING protein in EBC buffer overnight with constant agitation at 4 °C. After multiple washes, bound proteins were eluted in 2× Laemmli sample buffer and analyzed by immunoblotting. To assay for TOPORS binding to poly-SUMO2 chains, purified Strep–HA–TOPORS proteins (0.7 μg per reaction) and recombinant poly-SUMO2$_{2-8}$ chains (R&D Systems, 0.3 mg per reaction) were incubated with Strep-Tactin Superflow resin (IBA Lifesciences) and incubated at 4 °C for 2 h in EBC buffer. The beads were then washed in EBC buffer, and bound proteins were eluted in 2× Laemmli sample buffer and analyzed by immunoblotting.

## Flow cytometry

Cells were collected by trypsinization, fixed in 4% paraformaldehyde in PBS for 15 min and permeabilized in 0.2% Triton X-100, 2% FBS in PBS for 20 min at room temperature. Permeabilized cells were washed in FACS buffer (PBS + 10% FBS) and incubated with sheep polyclonal anti-DNMT1 antibody[15] diluted in FACS buffer for 90 min at room temperature. After two washes in FACS buffer, cells were incubated with anti-sheep Alexa Fluor 488 (Invitrogen) diluted in FACS buffer for 1 h at room temperature. For EdU co-staining, washed cells were subsequently stained using the Click-iT Plus EdU Alexa Fluor 647 Kit (Invitrogen) according to the manufacturer's instructions. After two additional washes in FACS buffer, cells were resuspended in FACS buffer containing 1 µg ml$^{-1}$ DAPI (Thermo Fisher Scientific), strained to a single-cell solution (40-µm filter) and analyzed using an LSRFortessa flow cytometer (BD Biosciences). Analysis was performed and plots were generated using FlowJo software (version 10.8.1).

## Mutagenesis of HAP1 cells

Gene-trap mutagenesis of HAP1 cell lines was carried out as described[57]. In brief, a BFP-containing variant of gene-trap retrovirus was generated in low-passage HEK293T cells by co-transfection of gene-trap vector, retroviral packaging plasmids Gag-pol and VsVg and pAdvantage (Promega). Media containing retrovirus were collected 48 h and 72 h after transfection and concentrated using centrifugation filters (Amicon), and the pooled, concentrated retrovirus was used to transduce $40 \times 10^6$ HAP1 cells supplemented with protamine sulfate (8 µg ml$^{-1}$) for 24 h. After recovery, mutagenized HAP1 cells were expanded and used for genetic screens.

## FACS-based screens for DNMT1 abundance

To identify regulators of 5-AzadC-induced DNMT1 processing, mutagenized HAP1 WT cells were expanded to $3 \times 10^9$ cells per screen. Two screens were performed: a mock DNMT1 screen where cells were labeled for 2 h with 10 µM EdU and a perturbation DNMT1 screen with co-treatment of 10 µM 5-AzadC and EdU. Treated cells were then trypsinized, fixed, permeabilized and stained for DNMT1 and EdU as described in the 'Flow cytometry' subsection above. Cells were sorted on a BD FACSAria Fusion cell sorter gating for haploid EdU-incorporating cells, and the cells with the 5% highest and lowest DNMT1 signal, respectively, were collected. Genomic DNA was extracted from sorted cells ($1.3 \times 10^7$ cells per channel) using the QIAamp DNA Mini Kit (Qiagen). Gene-trap insertion site recovery and sequencing libraries were generated as previously described[58]. Amplified libraries were sequenced on a HiSeq 2500 (Illumina) with a read length of 65 bp (single-end read). Sequencing reads from each sample (low and high) were aligned to the human genome (hg38) allowing one mismatch and assigned to non-overlapping protein-coding gene regions (RefSeq). The number of unique gene-trap insertions in the sense direction of each gene was normalized to the total number of sense insertions of each sorted population. The mutational index (MI) was calculated for each gene by comparing the number of unique, normalized sense integrations of the high population to that of the low population using a two-sided Fisher's exact test (false discovery rate (FDR)-corrected $P < 0.05$). Every gene was then plotted in fishtail scatterplots comparing the combined number of unique insertions identified in the two populations of a given gene (log$_{10}$, $x$ axis) to its MI (log$_2$, $y$ axis). To identify genes selectively affecting DNMT1 abundance in response to 5-AzadC, but not in the mock screen, a comparative filtering of genes was performed. Significant positive and negative regulators that scored as such in both screens (except for the antigen target DNMT1) were removed from the 5-AzadC screen to highlight DNMT1 DPC-specific regulators and generate Fig. 1b. Significant regulators for each individual screen can be found in Supplementary Data 1. Fishtail scatterplots were generated using GraphPad Prism software.

## Fitness-based screens to identify synthetic lethal interactions

Haploid genetic fitness screens were carried out as described in ref. 47. In brief, mutagenized HAP1 cell lines (minimum coverage of $2.5 \times 10^8$ cells per screen) were passaged for 10 d, trypsinized and fixed using Fixation Buffer I (BD Biosciences) for 10 min at 37 °C. For RNF4−dTAG cells, cells were passaged in the presence of 0.25 µM dTAG-13 to induce loss of RNF4. Cells were then permeabilized with Perm Buffer III (BD Biosciences) and stained in FACS buffer containing 2.5 µg ml$^{-1}$ DAPI (Thermo Fisher Scientific). After washing in FACS buffer, cells were strained to a single-cell solution (40-µm filter), and a minimum of $3 \times 10^7$ haploid cells (based on DAPI content) were isolated using a BD FACSAria Fusion cell sorter. Isolation of genomic DNA, library preparation and insertion site mapping were done as described for the FACS-based screens above. Analysis of gene-trap insertion orientation bias (sense ratio) was performed as previously described[47], and the analysis pipeline can be found on GitHub (https://github.com/BrummelkampResearch/phenosaurus). For each replicate experiment, a two-sided binomial test was calculated, which gives a $P$ value for each gene. These $P$ values were then corrected for FDR using the Benjamini−Hochberg procedure, and the least significant $P$ value among the individual replicates of a genotype was used to determine if a gene was considered significant. Every replicate corresponds to an independent clonal cell line of the respective genotype. Four independent cultured WT control datasets published in ref. 47 were used as control (available at Sequence Read Archive SRP058962, accession numbers SRX1045464, SRX1045465, SRX1045466 and SRX1045467). To identify genes that affect cell viability selectively in TOPORS-deficient and RNF4-deficient cell lines, the number of disruptive sense integrations and non-disruptive antisense integrations for each gene was compared to that in the four control datasets using a two-sided Fisher's exact test. Genes with a significant orientation bias in screen replicates of TOPORS-deficient and RNF4-deficient cells, respectively, in addition to a significantly altered sense ratio ($P < 0.05$, odds ratio < 0.8) in relation to the control datasets, were considered as hits.

## Generation of HAP1 cell lines

To generate clonal TOPORS-KO cell lines, parental HAP1 cells were co-transfected with a blasticidin resistance cassette and the CRISPR−Cas9 vector pX330 containing sgRNA sequences targeting *TOPORS* (sgTOPORS1 (5′-AACAGTACTCCACTATCCGG-3′) and sgTOPORS2 (5′-GGTAGCGAAATCGTCGATCA-3′)). Cells were then briefly selected (48 h) during clonal outgrowth, and gene status was monitored using Sanger sequencing of genomic DNA and immunoblot analysis. C-terminal degron tagging of endogenous RNF4 was carried out by a generic CRISPR−Cas9 strategy as previously described[59] but using a modified pTIA donor vector containing HA-tagged FKBP12(F36V) (dTAG) and a P2A sequence followed by a blasticidin cassette for integration selection (pTIA dTAG−HA P2A Blast). To generate clonal cell lines, parental HAP1 cells were co-transfected (1:1) with pTIA dTAG donor vector and a CRISPR−Cas9 vector containing an sgRNA targeting the last exon (exon 8) of RNF4 (pX330 sgRNF4 (5′-TACTTCATATATAAATGGGG-3′)) and subjected to blasticidin selection during clonal outgrowth. HAP1 RNF4−dTAG clones were validated by Sanger sequencing of the genomic locus and immunoblot analysis. The two RNF4−dTAG clones used contained an in-frame insertion of the dTAG donor vector at the C-terminus of RNF4, after amino acid His186 (hg38 genome coordinate chr4:2,513,804).

## Reporting summary

Further information on research design is available in the Nature Portfolio Reporting Summary linked to this article.

## Data availability

The mass spectrometry proteomics data have been deposited to the ProteomeXchange Consortium[60] via the Proteomics Identification

(PRIDE) partner repository (http://www.ebi.ac.uk/pride) under dataset ID PXD041717 (https://proteomecentral.proteomexchange.org/cgi/GetDataset?ID=PXD041717) (Supplementary Data 2) and dataset ID PXD041718 (https://proteomecentral.proteomexchange.org/cgi/GetDataset?ID=PXD041718) (Supplementary Data 3). Raw sequencing data from genetic screens have been deposited to the National Center for Biotechnology Information's Sequence Read Archive (https://www.ncbi.nlm.nih.gov/sra) under dataset ID PRJNA975887 (https://dataview.ncbi.nlm.nih.gov/object/PRJNA975887?reviewer=af0jv5dqrcq2pd277kfdb5lk7e). All other data supporting the findings of this study are available within the article. Source data are provided with this paper.

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

## Acknowledgements

We thank A. Mazouzi and members of the Mailand laboratory for helpful discussions; A. Sharrocks (University of Manchester) and Y. Durocher (National Research Council Canada) for providing reagents; and G. H. Prince for bioinformatic support. This work was supported by grants from the Novo Nordisk Foundation (NNF14CC0001 and NNF18OC0030752 (N.M.)), the Lundbeck Foundation (R223-2016-281 (N.M.)), the Independent Research Fund Denmark (0134-00048B (N.M.)) and the Danish National Research Foundation (DNRF-115 (N.M.) and DNRF-166 (P.H.)). R.T.H. was supported by an Investigator Award from the Wellcome Trust (217196/Z/19/Z) and, with M.H.T., a program grant from Cancer Research UK (C434/A21747). J.C.Y.L. was supported by the Croucher Foundation. P.H. was supported by the Novo Nordisk Foundation.

## Author contributions

Conceptualization: J.C.Y.L., P.H. and N.M. Methodology: J.C.Y.L., S.H., I.A.H., M.H.T., G.-L.M., R.T.H., M.L.N., T.B., P.H. and N.M. Investigation: J.C.Y.L., L.A., S.H., Z.G., I.A.H., C.J., L.M., M.H.T. and P.H. Writing—original draft: N.M. Writing—review and editing: all authors. Supervision: R.T.H., M.L.N., T.B. and N.M. Project administration: P.H. and N.M. Funding acquisition: R.T.H., P.H. and N.M.

## Competing interests

The authors declare no competing interests.

## Additional information

**Extended data** is available for this paper at https://doi.org/10.1038/s41594-024-01294-7.

**Correspondence and requests for materials** should be addressed to Peter Haahr or Niels Mailand.

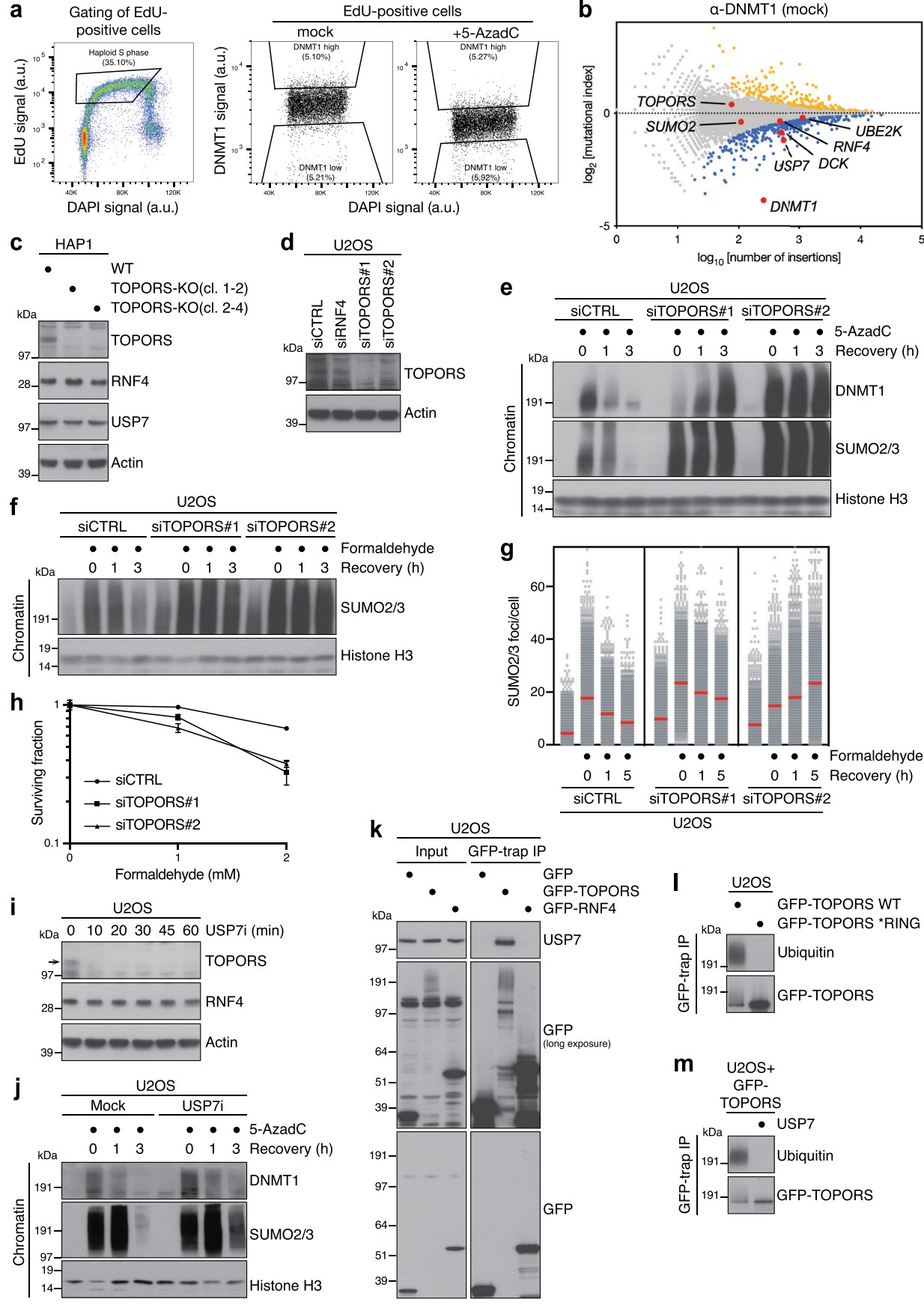

**Extended Data Fig. 1 | See next page for caption.**

**Extended Data Fig. 1 | TOPORS is required for SUMO-dependent DPC resolution and is stabilized by USP7. a**. Flow cytometry plots of gating strategy for FACS-based DNMT1 haploid genetic screens (Fig. 1b; Extended Data Fig. 1b). **b**. Mock screen for DNMT1 abundance in EdU-positive cells ($n = 1$). Positive and negative regulators are labeled in blue and yellow, respectively (two-sided Fisher's exact test, FDR corrected p ≤ 0.05; non-significant genes are shown in grey). **c**. Immunoblot analysis of HAP1 WT and TOPORS-KO cells. **d**. Immunoblot analysis of U2OS cells transfected with non-targeting control (CTRL), RNF4, or TOPORS siRNAs. **e**. U2OS cells transfected with indicated siRNAs were released from thymidine synchronization, exposed to 5-AzadC for 30 min and collected at indicated times. Chromatin-enriched fractions were immunoblotted with indicated antibodies. **f**. Immunoblot analysis of chromatin-enriched fractions of siRNA-transfected U2OS cells exposed to formaldehyde for 1 h and collected at indicated times. **g**. Cells treated as in (f) were pre-extracted and stained with SUMO2/3 antibody. SUMO2/3 foci counts were analyzed by QIBC (red bars, mean; >4,400 cells analyzed per condition). **h**. Clonogenic survival of U2OS cells transfected with indicated siRNAs and exposed to formaldehyde for 30 min before replating (mean ± SEM; $n = 3$ independent experiments). **i**. Immunoblot analysis of U2OS cells treated with USP7i. Arrow indicates the band corresponding to endogenous TOPORS. **j**. As in (e), except USP7i was administered together with 5-AzadC where indicated. **k**. Immunoblot analysis of U2OS cells transfected with GFP expression plasmids and subjected to GFP IP. **l**. Immunoblot analysis of U2OS cells transfected with indicated GFP-TOPORS expression constructs and subjected to GFP IP under denaturing conditions. RING domain mutations (*RING) inactivating TOPORS E3 ubiquitin ligase activity (Fig. 2e) abolish its ubiquitylation, suggesting this reflects auto-ubiquitylation. **m**. U2OS cells transfected with GFP-TOPORS WT construct were subjected to GFP IP under denaturing conditions. Immobilized proteins were incubated with or without recombinant USP7 protein and immunoblotted with ubiquitin and GFP antibodies. Data information: Data are representative of four (d), three (f,g,i,k) and two (c,e,j,l,m) independent experiments with similar outcome.

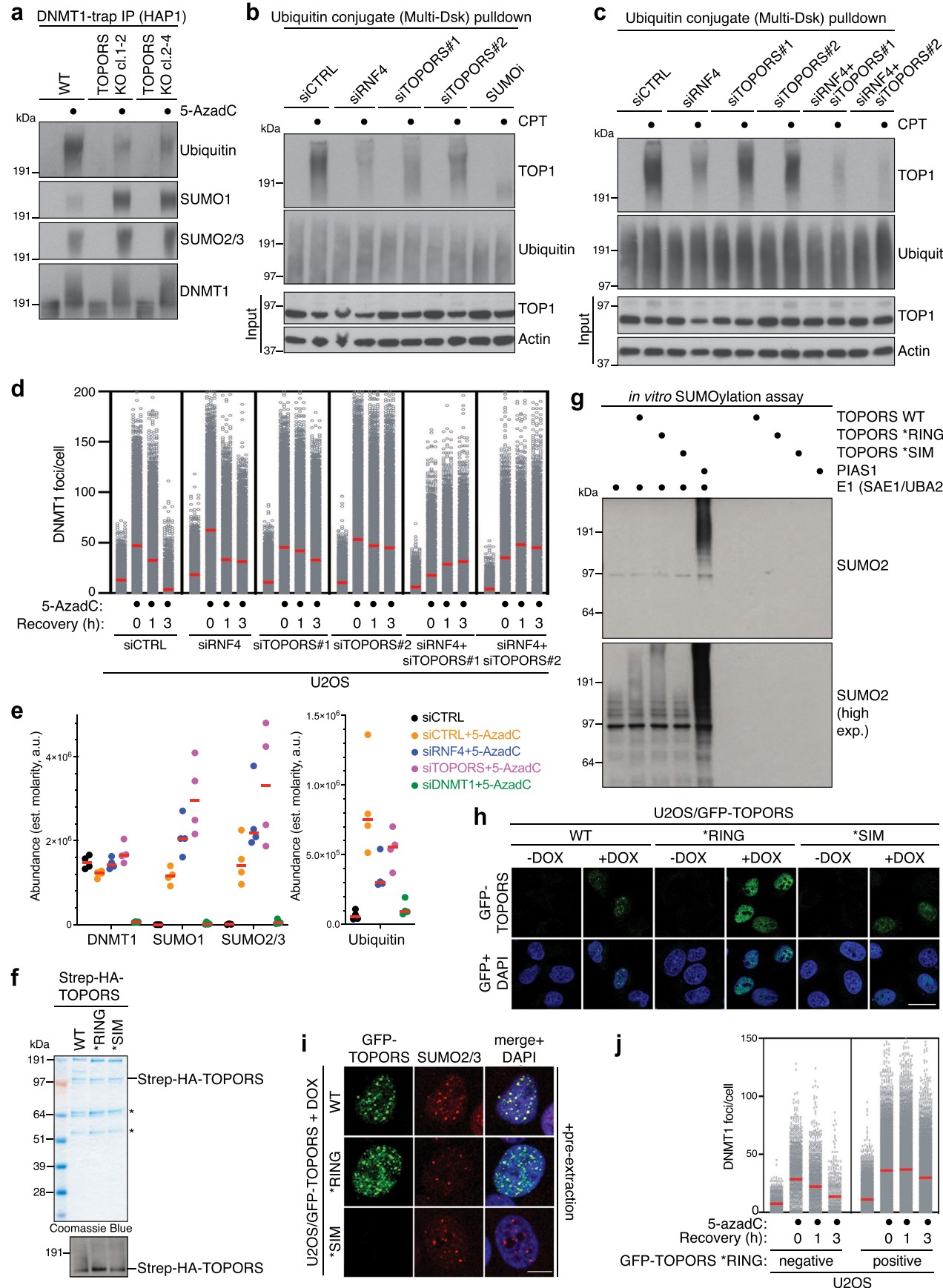

**a** DNMT1-trap IP (HAP1)

**b** Ubiquitin conjugate (Multi-Dsk) pulldown

**c** Ubiquitin conjugate (Multi-Dsk) pulldown

**Extended Data Fig. 2 | See next page for caption.**

**Extended Data Fig. 2 | TOPORS functions as an E3 ubiquitin ligase in DPC repair. a**. Immunoblot analysis of HAP1 WT and TOPORS-KO cells transfected with control (CTRL) or TOPORS siRNAs that were released from synchronization in early S phase, pulse-labeled or not with 5-AzadC for 30 min and subjected to DNMT1 IP under denaturing conditions. **b**. Immunoblot analysis of U2OS cells transfected with siRNAs for 72 h, treated or not with camptothecin (CPT) and SUMO inhibitor (SUMOi) and subjected to Multi-Dsk pulldown under denaturing conditions to isolate total cellular ubiquitylated proteins. **c**. As in (b), except cells were treated with siRNAs for 40 h. **d**. U2OS cells transfected with siRNAs released from synchronization in early S phase were pulse-labeled with 5-AzadC for 30 min, collected at indicated times, pre-extracted and immunostained with DNMT1 antibody. DNMT1 foci formation was analyzed by quantitative image-based cytometry (QIBC) (red bars, mean; >2,400 cells analyzed per condition). **e**. siRNA-transfected U2OS cells were subjected to DNMT1 IP under denaturing conditions to isolate DNMT1 but not associated proteins. Samples were digested with trypsin and DNMT1, SUMO1, SUMO2/3, and ubiquitin peptides were identified by MS ($n$ = 4 independent experiments). Molarity was approximated by dividing each protein's intensity-based abundance by its molecular weight. **f**. Strep-HA-TOPORS proteins purified from HEK293-6E cells were analyzed by Coomassie Blue staining or immunoblotting with HA antibody. Asterisks indicate Strep-HA-TOPORS degradation products. **g**. Immunoblot analysis of *in vitro* SUMOylation reactions containing recombinant Strep-HA-TOPORS proteins, E1 (SAE1-UBA2) and E2 (UBE2I) enzymes, SUMO2, ATP and SUMO2-VS. **h**. Representative images of non-pre-extracted stable U2OS/GFP-TOPORS cell lines induced or not to express GFP-TOPORS proteins with Doxycycline (DOX). Scale bar, 10 μm. **i**. Representative images of DOX-treated U2OS/GFP-TOPORS cell lines immunostained with SUMO2/3 antibody after pre-extraction. Scale bar, 10 μm. **j**. U2OS cells stably expressing GFP-TOPORS *RING were released from synchronization in early S phase, pulse-treated with 5-AzadC for 30 min, pre-extracted and immunostained with DNMT1 antibody. DNMT1 foci formation in GFP-positive and -negative cells was analyzed by QIBC (red bars, mean; >479 cells analyzed per condition). Data information: Data are representative of three (b,c,f-j) and two (a,d) independent experiments with similar outcome.

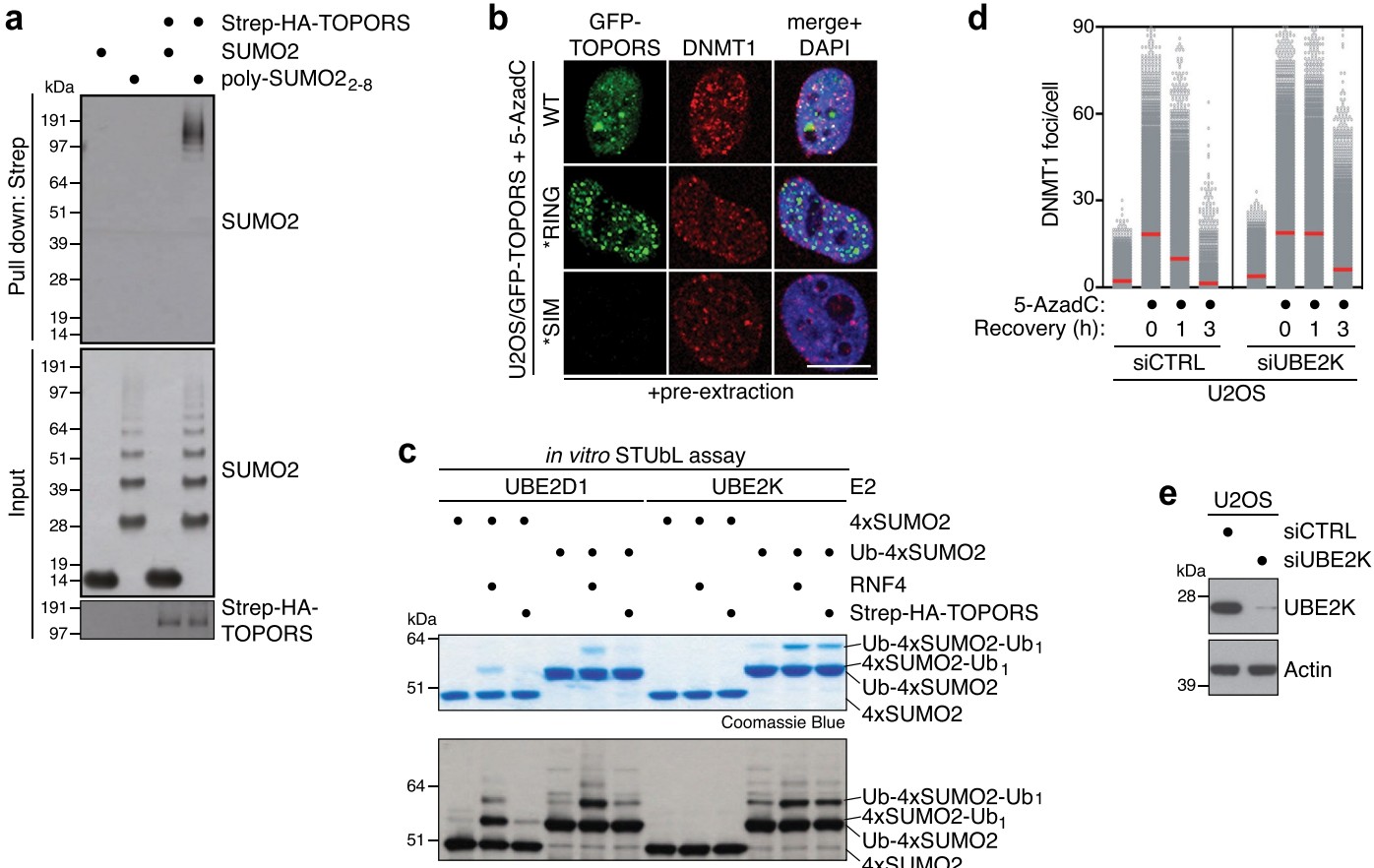

**Extended Data Fig. 3 | TOPORS is a SUMO-targeted ubiquitin ligase.**
**a**. Immunoblot analysis of recombinant Strep-HA-TOPORS proteins incubated with free SUMO2 or poly-SUMO2 chains and subjected to Strep-Tactin pulldown. **b**. Representative images of DOX-treated U2OS/GFP-TOPORS cell lines immunostained with DNMT1 antibody after pre-extraction. Scale bar, 10 μm. **c**. Coomassie staining (top) and immunoblot analysis using SUMO2 antibody (bottom) of *in vitro* STUbL reactions containing recombinant RNF4 or Strep-HA-TOPORS, E1 (UBA1) and E2 enzymes, FLAG-ubiquitin, 4xSUMO2 STUbL substrate,

ATP, Ub-VS and SUMO2-VS. **d**. U2OS cells transfected with indicated siRNAs were released from single thymidine synchronization in early S phase and pulse-treated with 5-AzadC for 30 min. Cells were collected at indicated time points, pre-extracted and immunostained with DNMT1 antibody. DNMT1 foci formation was analyzed by QIBC (red bars, mean; >6,400 cells analyzed per condition). **e**. Immunoblot analysis of U2OS cells transfected with non-targeting control (CTRL) or UBE2K siRNAs. Data information: Data are representative of three (a-d) and two (e) independent experiments with similar outcome.

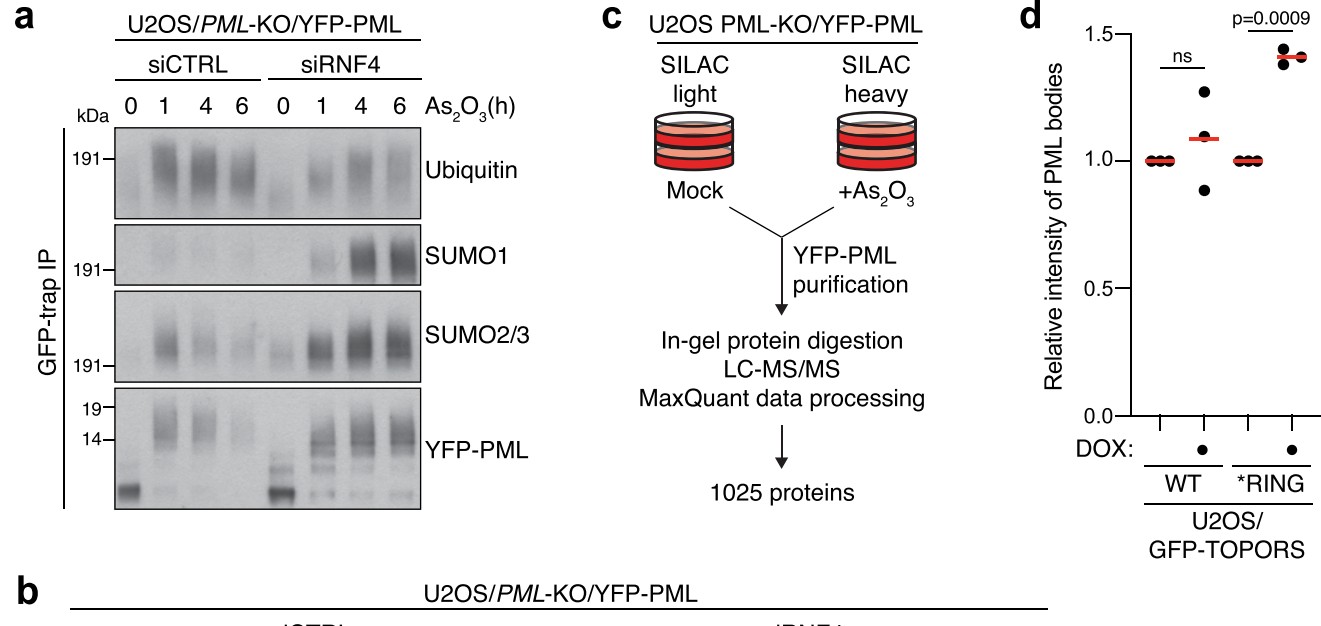

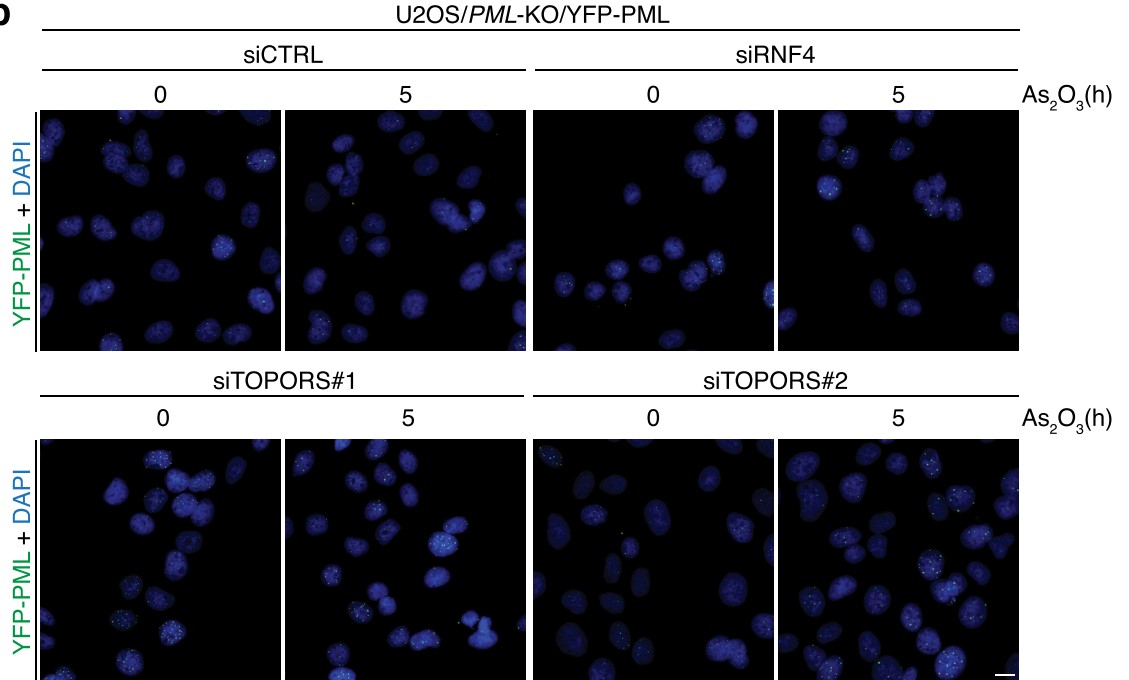

**Extended Data Fig. 4 | STUbL-dependent turnover of PML bodies. a.** U2OS PML-KO/YFP-PML cells transfected with indicated siRNAs were treated with arsenic and collected at indicated times. Cells were subjected to GFP IP under denaturing conditions and immunoblotted with indicated antibodies. **b.** Representative images of cells in Fig. 3b. Scale bar, 10 μM. **c.** Schematic overview of the proteomics experiment in Fig. 3e. **d.** U2OS cells stably expressing GFP-TOPORS WT or *RING were treated or not with Doxycycline (DOX) for 36 h. Cells were pre-extracted and immunostained with PML antibody. Intensity of PML bodies in GFP-positive and -negative cells was analyzed by QIBC and normalized to untreated GFP-negative cells (red bars, mean; *n* = 3; ns: not significant, one-tailed paired t-test). Data information: Data are representative of three (a,b) independent experiments with similar outcome.

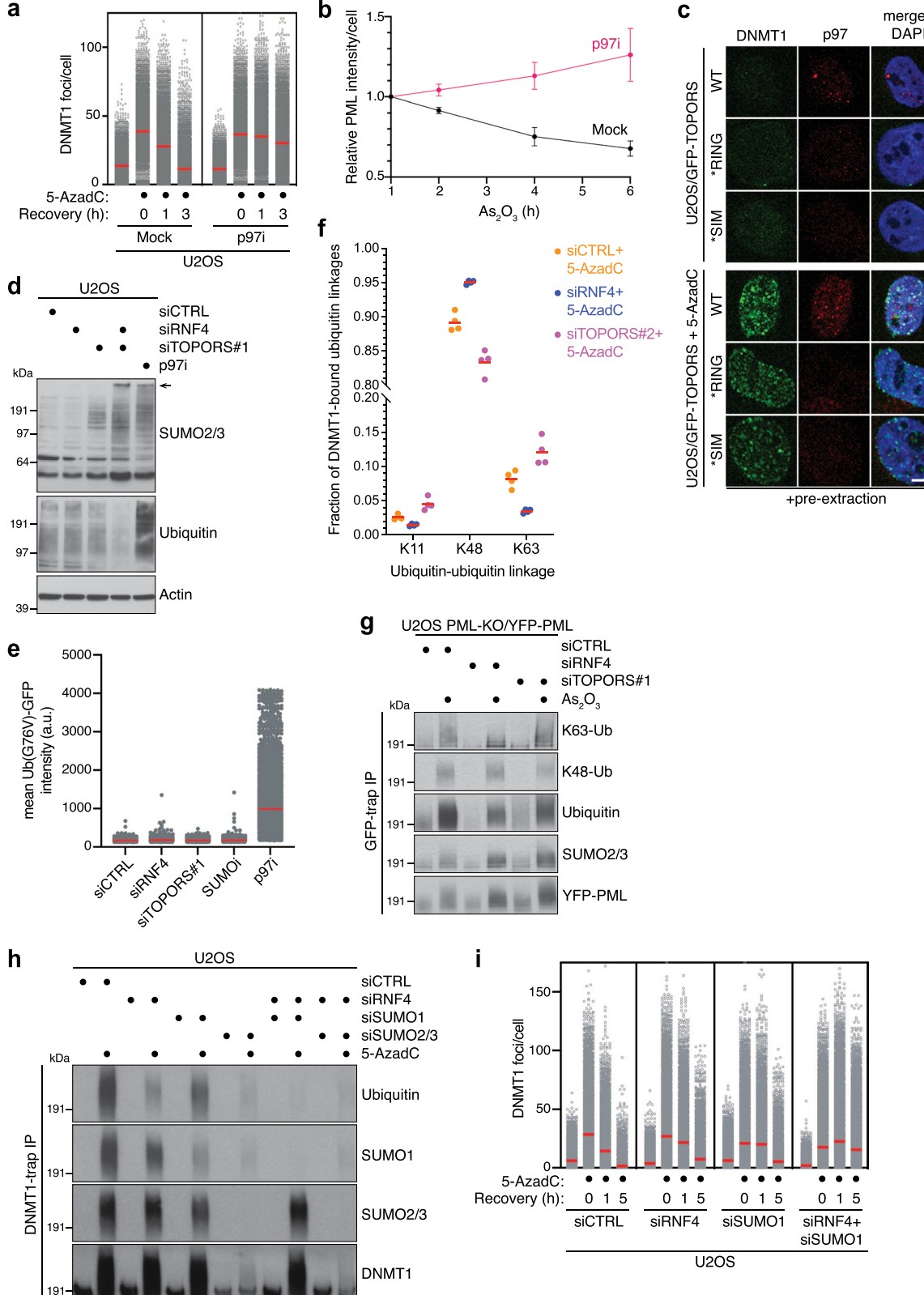

**Extended Data Fig. 5 | See next page for caption.**

**Extended Data Fig. 5 | RNF4 and TOPORS promote p97-dependent processing of SUMOylated proteins. a**. U2OS cells released from single thymidine synchronization in early S phase and pulse-labeled with 5-AzadC for 30 min in presence or absence of p97 inhibitor (p97i) were pre-extracted and immunostained with DNMT1 antibody. DNMT1 foci formation was analyzed by QIBC (red bars, mean; >8,100 cells analyzed per condition). **b**. U2OS PML-KO/YFP-PML cells were exposed to arsenic in presence or absence of USP7i. Cells were collected at indicated times, pre-extracted and YFP-PML intensity was analyzed by QIBC (mean ± SEM; $n$ = 3 independent experiments). **c**. Representative images of DOX-treated U2OS/GFP-TOPORS cell lines treated or not with 5-AzadC and co-immunostained with DNMT1 and p97 antibodies after pre-extraction and fixation. Scale bar, 10 µm. **d**. Immunoblot analysis of U2OS cells transfected with siRNAs for 40 h or treated with p97i for 10 h. **e**. U2OS stably expressing Ub(G76V)-GFP reporter transfected with indicated siRNAs were treated or not with p97i for 4 h. Nuclear Ub(G76V)-GFP signal intensity was analyzed by QIBC (red bars, mean; >12,000 cells analyzed per condition). **f**. U2OS cells transfected with indicated siRNAs were subjected to DNMT1 IP under stringent denaturing conditions. Samples were digested with trypsin, and di-glycine remnants on ubiquitin were identified by MS ($n$ = 4 independent experiments). **g**. Immunoblot analysis of U2OS PML-KO/YFP-PML cells transfected with indicated siRNAs for 72 h, exposed to arsenic for 1 h and subjected to GFP IP under denaturing conditions. **h**. Immunoblot analysis of U2OS cells transfected with control (CTRL), RNF4, SUMO1 and/or SUMO2/3 siRNAs that were released from synchronization in early S phase, treated or not with 5-AzadC for 30 min and subjected to DNMT1 IP under denaturing conditions. **i**. U2OS cells transfected with control (CTRL), RNF4 and/or SUMO1 siRNAs were released from single-round thymidine synchronization in early S phase and pulse-labeled with 5-AzadC for 30 min. Cells were pre-extracted at indicated times, immunostained with DNMT1 antibody, and DNMT1 foci formation was analyzed by quantitative image-based cytometry (QIBC) (red bars, mean; >6,600 cells analyzed per condition). Data information: Data are representative of three (a,c,g-i) or two (d,e) independent experiments with similar outcome.

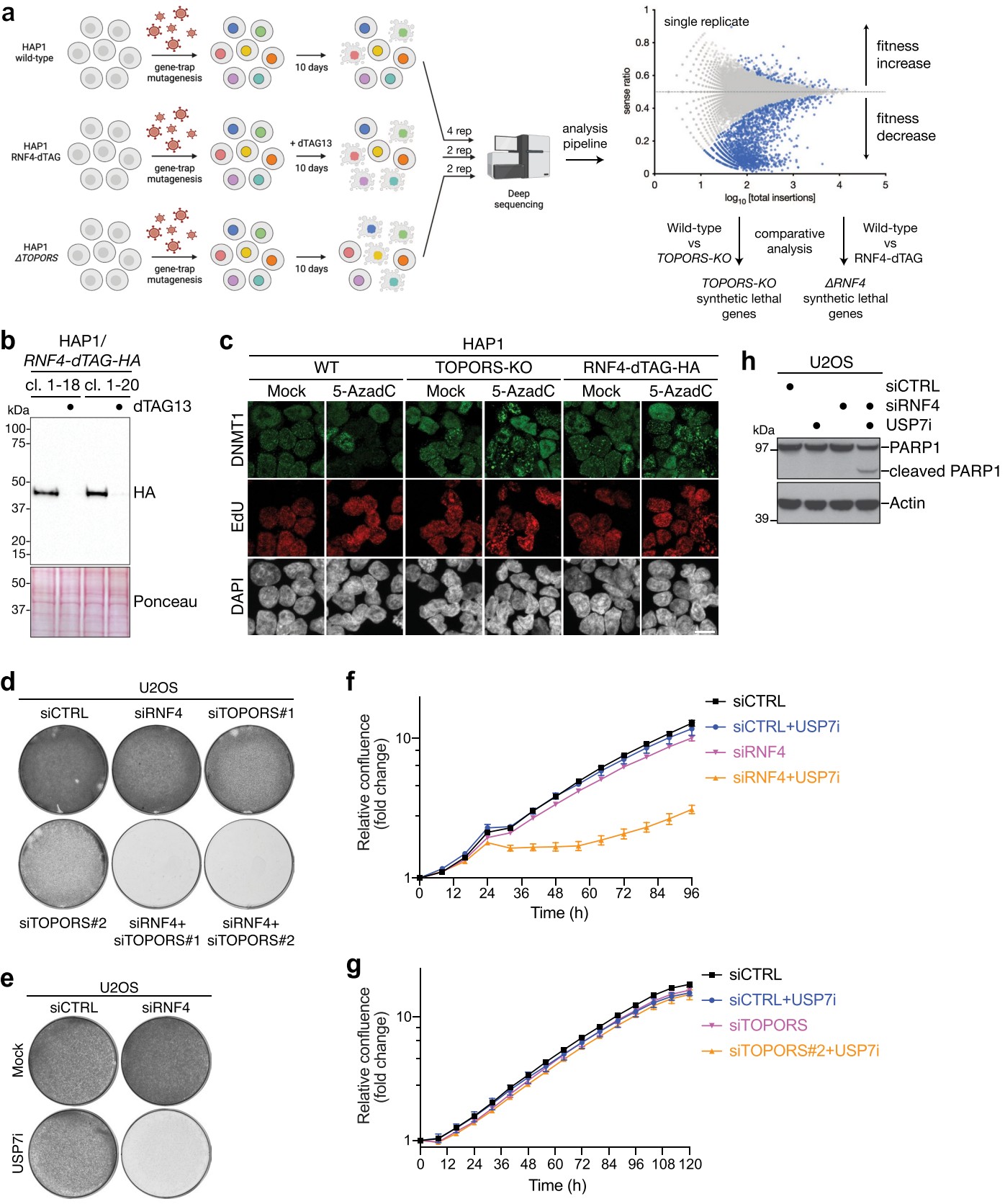

**Extended Data Fig. 6 | See next page for caption.**

**Extended Data Fig. 6 | Synthetic lethality between TOPORS and RNF4.**
**a**. Workflow of fitness-based haploid genetic screens in Fig. 5a–d (created with Biorender). **b**. Immunoblot analysis of HAP1 RNF4-dTAG-HA clones treated or not with dTAG13 for 24 h. The membrane was co-stained with Ponceau S (loading control). **c**. Representative images of HAP1 cell lines treated with EdU in presence or absence of 10 μM 5-AzadC for 60 min. RNF4-dTAG-HA cells were pre-treated with dTAG-13 for 4 h. Cells were fixed and stained using antibodies and reagents for DNMT1, EdU and DAPI. Scale bar, 10 μM. **d**. Equal numbers of cells were seeded in a 6-well plate, transfected with indicated siRNAs for 72 h and stained with crystal violet. **e**. As in (d), except cells were transfected with siRNAs for 48 h and subsequently treated or not with USP7i for an additional 48 h prior to fixation. **f**. Normalized logarithmic proliferation quantification for U2OS cells transfected with control (CTRL) or RNF4 siRNAs for 48 h and then incubated or not with USP7i, as determined by Incucyte image-based confluence analysis (mean ± SD; $n$ = 3 technical replicates). Imaging started at 72 h after siRNA transfection. **g**. As in (f), except cells were transfected with CTRL or TOPORS siRNAs (mean ± SD; $n$ = 3 technical replicates). **h**. Immunoblot analysis of U2OS cells transfected with indicated siRNAs for 48 h and grown in presence or absence of USP7i for an additional 24 h. Data information: Data are representative of three (b,d,e) and two (c,h) independent experiments with similar outcome.

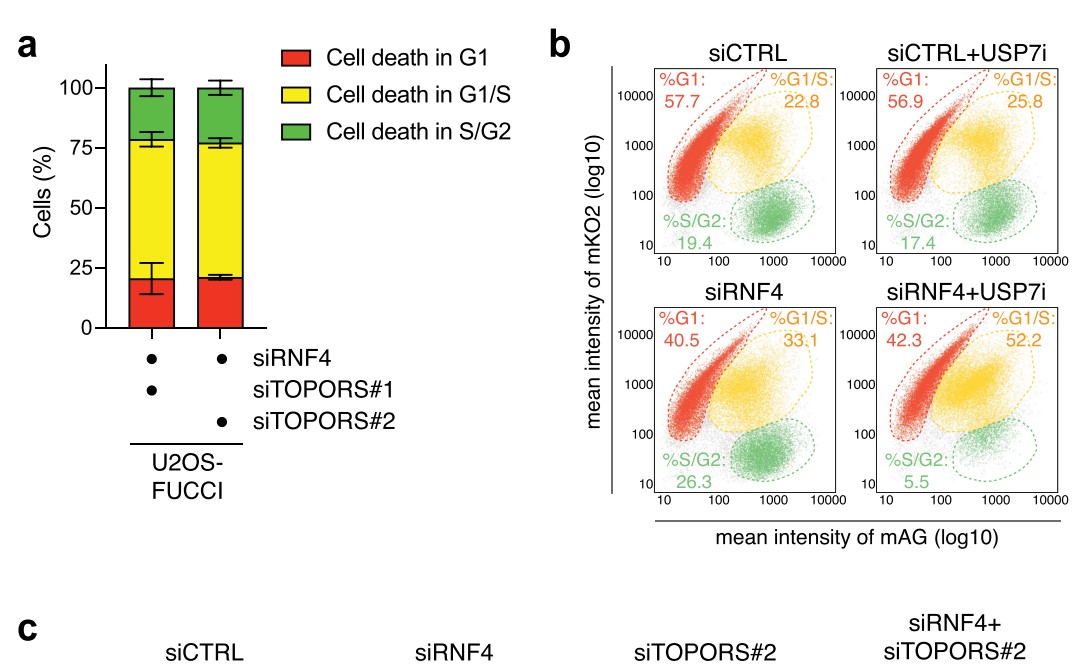

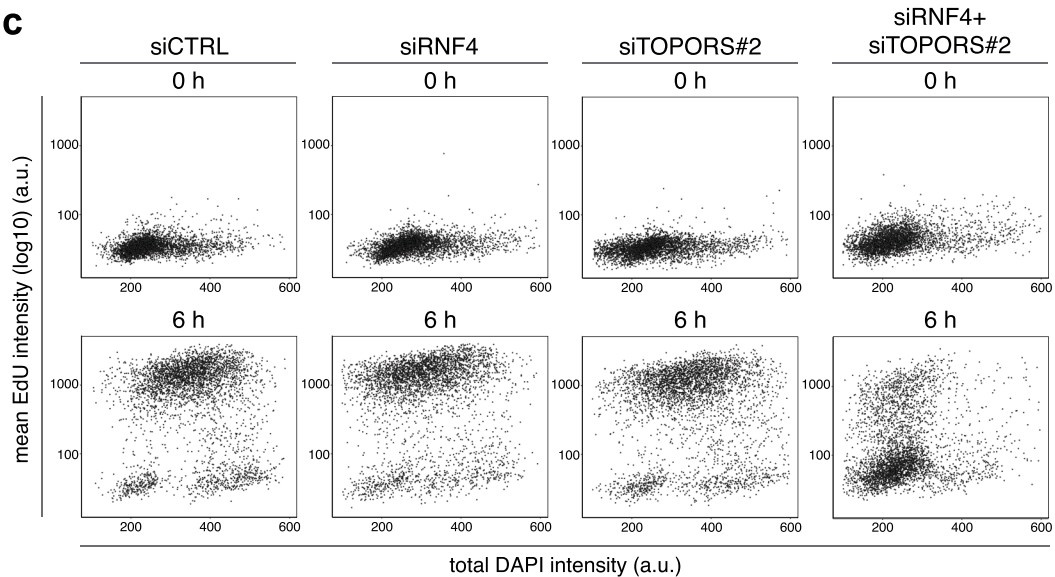

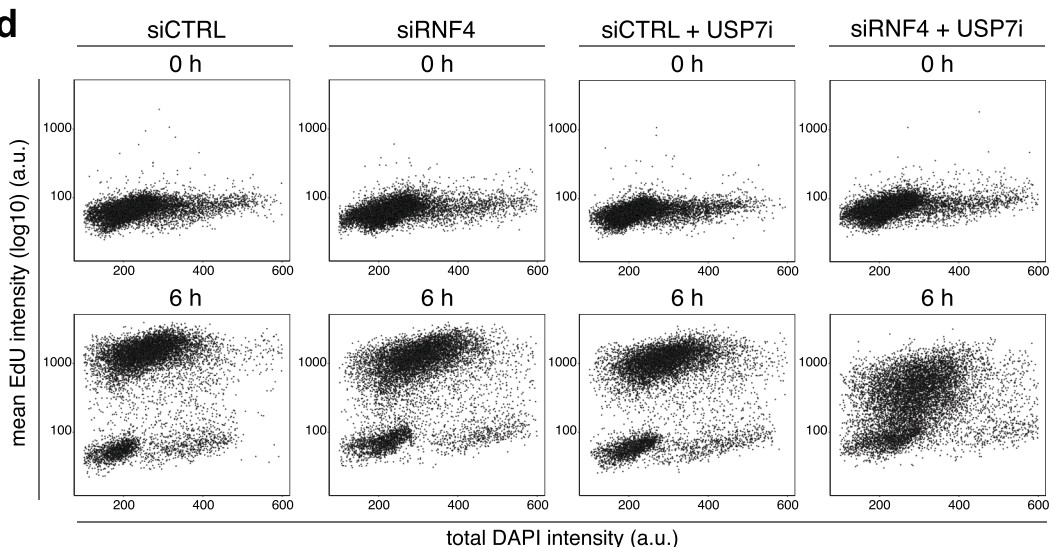

**Extended Data Fig. 7 | See next page for caption.**

**Extended Data Fig. 7 | Combined loss of TOPORS and RNF4 leads to defective DNA replication and cell death in S phase. a**. U2OS Fucci cells transfected with indicated siRNAs were analyzed by live-cell imaging. Expression of CDT1 (mKO2-hCdt1) and Geminin (mAG-hGem) at the point of cell death was assessed (mean ± SEM; $n$ = 2; at least 45 cell death events were analyzed per condition per replicate). **b**. U2OS Fucci cells were transfected with indicated siRNAs for 24 h and grown in the absence or presence of USP7i for an additional 12 h. To determine cell cycle position, CDT1 (mKO2-hCdt1+) and Geminin (mAG-hGem+) intensities were analyzed by QIBC. Quantification of cells in different cell cycle phases was done using indicated gates. **c**. U2OS cells were transfected with indicated siRNAs and synchronized by treatment with thymidine for 18 h. DNA synthesis profiles of cells pulsed with EdU at indicated time points after release from thymidine arrest were determined by QIBC analysis of DAPI and EdU signal intensities (>3,900 cells analyzed per condition). **d**. U2OS cells were transfected with indicated siRNAs and synchronized by treatment with thymidine for 18 h. One h prior to thymidine release, USP7i was added where indicated. DNA synthesis profiles of cells pulsed with EdU at indicated time points after release from thymidine arrest were determined by QIBC analysis of DAPI and EdU signal intensities (>10,000 cells analyzed per condition). Data information: Data are representative of two (b,c,d) independent experiments with similar outcome.

# Reporting Summary

## Statistics

For all statistical analyses, confirm that the following items are present in the figure legend, table legend, main text, or Methods section.

| n/a | Confirmed | |
|---|---|---|
| ☐ | ☒ | The exact sample size (*n*) for each experimental group/condition, given as a discrete number and unit of measurement |
| ☐ | ☒ | A statement on whether measurements were taken from distinct samples or whether the same sample was measured repeatedly |
| ☐ | ☒ | The statistical test(s) used AND whether they are one- or two-sided<br>*Only common tests should be described solely by name; describe more complex techniques in the Methods section.* |
| ☒ | ☐ | A description of all covariates tested |
| ☐ | ☒ | A description of any assumptions or corrections, such as tests of normality and adjustment for multiple comparisons |
| ☐ | ☒ | A full description of the statistical parameters including central tendency (e.g. means) or other basic estimates (e.g. regression coefficient) AND variation (e.g. standard deviation) or associated estimates of uncertainty (e.g. confidence intervals) |
| ☐ | ☒ | For null hypothesis testing, the test statistic (e.g. $F$, $t$, $r$) with confidence intervals, effect sizes, degrees of freedom and $P$ value noted<br>*Give P values as exact values whenever suitable.* |
| ☒ | ☐ | For Bayesian analysis, information on the choice of priors and Markov chain Monte Carlo settings |
| ☒ | ☐ | For hierarchical and complex designs, identification of the appropriate level for tests and full reporting of outcomes |
| ☒ | ☐ | Estimates of effect sizes (e.g. Cohen's *d*, Pearson's *r*), indicating how they were calculated |

*Our web collection on statistics for biologists contains articles on many of the points above.*

## Software and code

Policy information about availability of computer code

| | |
|---|---|
| Data collection | Incucyte S3 Live-Cell Analysis System (version 2021A), SoftWoRx software (version 7.0.0; GE Healthcare), FlowJo (version 10.8.1). |
| Data analysis | ScanR analysis software (version 2.8.1; Olympus), Spotfire (version 10.5.0; Tibco), SoftWoRx software (version 7.0.0; GE Healthcare), FlowJo, GraphPad Prism (version 9.5.1), MaxQuant (versions 1.5.3.30 and 1.6.1.0), Perseus (versions 1.5.5.3 and 1.6.14.0). Aanalysis pipeline for gene-trap insertion orientation bias (sense-ratio) can be found at GitHub (https://github.com/BrummelkampResearch/phenosaurus). |

For manuscripts utilizing custom algorithms or software that are central to the research but not yet described in published literature, software must be made available to editors and reviewers. We strongly encourage code deposition in a community repository (e.g. GitHub). See the Nature Portfolio guidelines for submitting code & software for further information.

## Data

Policy information about availability of data

All manuscripts must include a data availability statement. This statement should provide the following information, where applicable:
- Accession codes, unique identifiers, or web links for publicly available datasets
- A description of any restrictions on data availability
- For clinical datasets or third party data, please ensure that the statement adheres to our policy

The mass spectrometry proteomics data have been deposited to the ProteomeXchange Consortium (ref. 60) via the Proteomics Identification (PRIDE) partner repository (http://www.ebi.ac.uk/pride) under dataset ID PXD041717 (https://proteomecentral.proteomexchange.org/cgi/GetDataset?ID= PXD041717)

# Research involving human participants, their data, or biological material

Policy information about studies with <u>human participants or human data</u>. See also policy information about <u>sex, gender (identity/presentation), and sexual orientation</u> and <u>race, ethnicity and racism</u>.

| | |
|---|---|
| Reporting on sex and gender | N/A |
| Reporting on race, ethnicity, or other socially relevant groupings | N/A |
| Population characteristics | N/A |
| Recruitment | N/A |
| Ethics oversight | N/A |

Note that full information on the approval of the study protocol must also be provided in the manuscript.

# Field-specific reporting

Please select the one below that is the best fit for your research. If you are not sure, read the appropriate sections before making your selection.

☒ Life sciences　　☐ Behavioural & social sciences　　☐ Ecological, evolutionary & environmental sciences

For a reference copy of the document with all sections, see nature.com/documents/nr-reporting-summary-flat.pdf

# Life sciences study design

All studies must disclose on these points even when the disclosure is negative.

| | |
|---|---|
| Sample size | Sample sizes were based on previously reported data. No statistical method was used to predetermine sample size. |
| Data exclusions | No data were excluded from the analysis. |
| Replication | All experimental findings shown in this study were independently replicated at least twice with similar outcome. Information about replication is provided in the figure legends. |
| Randomization | The samples were not randomized. Randomization is generally not relevant for this study since we are working with cell populations and not test subjects. |
| Blinding | The investigators were not blinded to group allocation during data collection and analysis, but great care was taken to avoid bias. |

# Reporting for specific materials, systems and methods

We require information from authors about some types of materials, experimental systems and methods used in many studies. Here, indicate whether each material, system or method listed is relevant to your study. If you are not sure if a list item applies to your research, read the appropriate section before selecting a response.

## Materials & experimental systems

| n/a | Involved in the study |
|---|---|
| ☐ | ☒ Antibodies |
| ☐ | ☒ Eukaryotic cell lines |
| ☒ | ☐ Palaeontology and archaeology |
| ☒ | ☐ Animals and other organisms |
| ☒ | ☐ Clinical data |
| ☒ | ☐ Dual use research of concern |
| ☒ | ☐ Plants |

## Methods

| n/a | Involved in the study |
|---|---|
| ☒ | ☐ ChIP-seq |
| ☐ | ☒ Flow cytometry |
| ☒ | ☐ MRI-based neuroimaging |

# Antibodies

| | |
|---|---|
| Antibodies used | Antibodies to human proteins used in this study included: Actin (1:20,000 dilution (MAB1501 (clone ID: C4), Millipore, RRID:AB_2223041)); DNMT1 (1:1,000, described in ref. 11); FLAG (1:1,000 (A00187 (clone ID: 5A8E5), GenScript, RRID:AB_1720813)); GFP (1:5,000, Abcam (ab6556, RRID:AB_305564)); HA (1:1,000 (11867423001 (clone ID: 3F10), Roche, RRID:AB_390918)); Histone H3 (1:20,000, Abcam (ab1791, RRID:AB_302613)); PARP1 (1:1,000, Santa Cruz Biotechnology (sc-8007 (clone ID: F2), RRID:AB_628105)); PIAS1 (1:1,000 (ab77231, Abcam, RRID:AB_1524188)); RNF4 (1:5,000, described in ref. 54); SUMO1 (1:1,000, Thermo Fisher Scientific (33-2400 (clone ID: 21C7), RRID:AB_2533109)); SUMO2/3 (1:1,000, Abcam (ab3742, RRID:AB_304041); 1:1,000, Abcam (ab81371 (clone 1D: 8A2), RRID:AB_1658424)); TOP1 (1:500, Bethyl (A302-590A, RRID:AB_2034875)); TOPORS (1:250, sheep polyclonal raised against full-length human TOPORS); ubiquitin (1:1,000, Santa Cruz Biotechnology (sc-8017 AC (clone ID: P4D1), RRID:AB_2762364); 1:1,000, Millipore (04-263 (clone ID: FK2), RRID:AB_612093)); ubiquitin (K48-linked) (1:1,000, Millipore (05-1307 (clone ID: Apu2), RRID:AB_1587578)); ubiquitin (K63-linked) (1:1,000, (05-1308 (clone ID: Apu3), RRID:AB_1587580)); UBE2K (1:1,000, Cell Signaling Technology (3847, RRID:AB_2210768)); and USP7 (1:1,000, Bethyl (A300-033A, RRID:AB_203276)). |
| Validation | The specificity of antibodies against TOPORS, RNF4, UBE2K, FLAG, HA, SUMO1 and SUMO2/3 (ab3742) was validated using appropriate knockdown/knockout controls in this study (as shown in the manuscript). The following antibodies were validated in our previous studies: DNMT1 (PMID: 30914427), GFP (PMID: 34346517), SUMO2/3 (ab81371) (PMID: 34346517), ubiquitin (sc-8017 and 04-263) (PMID: 35349166). The following antibodies are commonly used loading controls: Actin, Histone H3. Other antibodies (PARP1, TOP1, ubiquitin (K48-linked), ubiquitin (K63-linked), USP7) were used based on previous validation in the literature and/or manufacturer websites. |

# Eukaryotic cell lines

Policy information about cell lines and Sex and Gender in Research

| | |
|---|---|
| Cell line source(s) | Human HeLa (catalog no. CCL-2) and U2OS (catalog no. HTB-96) cells were obtained from ATCC. HEK293-EBNA1-6E cells were a kind gift from Yves Durocher (National Research Council Canada, Montreal, Canada). The generation of HAP1 cells is described in ref. 53. |
| Authentication | The cell lines were not authenticated. |
| Mycoplasma contamination | All cell lines used in this study were regularly tested negative for mycoplasma infection. |
| Commonly misidentified lines (See ICLAC register) | Cell lines used in this study are not included in the ICLAC register of commonly misidentified cell lines. |

# Flow Cytometry

## Plots

Confirm that:

☒ The axis labels state the marker and fluorochrome used (e.g. CD4-FITC).

☒ The axis scales are clearly visible. Include numbers along axes only for bottom left plot of group (a 'group' is an analysis of identical markers).

☒ All plots are contour plots with outliers or pseudocolor plots.

☒ A numerical value for number of cells or percentage (with statistics) is provided.

## Methodology

| | |
|---|---|
| Sample preparation | Cells were collected by trypsinization, fixed in 4% paraformaldehyde in PBS for 15 min and permeabilized in 0.2% Triton-X, 2 FBS in PBS for 20 min at room temperature. Permeabilized cells were washed in FACS buffer (PBS+10% FBS) and incubated with sheep polyclonal anti-DNMT1 antibody 14 diluted in FACS buffer for 90 min at room temperature. Following two washes in FACS buffer, cells were incubated with anti-sheep AlexaFluor 488 (Invitrogen) diluted in FACS buffer for 1 h at room temperature. For EdU co-staining, washed cells were subsequently stained using the Click-iT Plus EdU Alexa Fluor 647 Kit (Invitrogen) according to the manufacturer's instructions. Following two additional washes in FACS buffer, cells were resuspended in FACS buffer containing 1 μg/ml DAPI (Thermo Fisher) and strained to a single cell solution (40 μm filter). |
| Instrument | Stained cells were analyzed using a BD LSRFortessa flow cytometer (BD Biosciences) or sorted using a BD FACSAria Fusion. |
| Software | Analysis was performed with and plots were generated using FlowJo software. |
| Cell population abundance | Single cell haploid S-phase population based on DAPI/EdU was approximately 35%, of these the top and bottom 5% based on DNMT1 signal were sorted. 13E6 cells were sorted for each channel. |
| Gating strategy | Cells were selected based on FSC/SSC, single cells gated based on DAPI-A vs DAPI-H, haploid S-phase selected based on EdU signal vs DAPI, DNMT1 high and low based on DNMT1 vs DAPI (Extended Data Figure 1A). |

☒ Tick this box to confirm that a figure exemplifying the gating strategy is provided in the Supplementary Information.

