## [Peer Review File · Nature Structural & Molecular Biology]

Peer Review Information

Manuscript Title: Concerted SUMO-targeted ubiquitin ligase activities of TOPORS and RNF4 are essential for stress management and cell proliferation

Corresponding author name(s): Niels Mailand, Peter Haahr

Reviewer Comments & Decisions:

Decision Letter, initial version:
--

Message: 25th Jul 2023

Dear Professor Mailand,

Thank you again for submitting your manuscript "Concerted SUMO-targeted ubiquitin ligase activities of TOPORS and RNF4 are essential for stress management and cell proliferation". I apologize for the delay in responding, which resulted from the difficulty in obtaining suitable referee reports. Nevertheless, we now have comments (below) from the 3 reviewers who evaluated your paper. In light of those reports, we remain interested in your study and would like to see your response to the comments of the referees, in the form of a revised manuscript.

You will see that while reviewers appreciate the results, they raise several concerns which will need to be addressed in a revision. Specifically, we would expect the results to be quantified where indicated by the reviewers, and appropriate controls added. We agree with reviewers #1 and #3 that exploring potential selectivity of TOPORS for SUMO 1 and 3 would be of interest. Moreover, we also find the aspect of delineating the hierarchy of TOPORS and RNF4 interesting, if feasible. We would encourage you to explore the idea of double KO's, suggested by reviewer #2. In line with reviewers' #2 and #3 comments, we think that exploring the question of whether topoisomerase-DPCs are resolved on TOPORS would strengthen the manuscript. While we agree that reconstituting the system in vitro would be informative, we don't consider it essential in the context of the current work. We do think, however, that K68 vs. K48 chain formation data could be strengthened, as suggested by reviewer #3.

Regarding Reviewer #2's point 2 on the western blots, we have assessed the image and we currently do not have editorial concerns regarding the integrity of the data, including western blots. However we do, as always, request that the raw and source data to support figures presented in manuscripts be provided with the revised manuscript. Please make sure to include raw, uncropped images of all western blots in main and extended data -

more detail is provided below

Please be sure to address/respond to all concerns of the referees in full in a point-by-point response and highlight all changes in the revised manuscript text file.

We appreciate the requested revisions are extensive. We thus expect to see your revised manuscript within 6 months. If you cannot send it within this time, please let us know. We will be happy to consider your revision as long as nothing similar has been accepted for publication at NSMB or published elsewhere. Should your manuscript be substantially delayed without notifying us in advance and your article is eventually published, the received date would be that of the revised, not the original, version.

Reporting Summary:

When submitting the revised version of your manuscript, please pay close attention to our [href="https://www.nature.com/nature-portfolio/editorial-policies/image-integrity">Digital Image Integrity Guidelines](https://www.nature.com/nature-portfolio/editorial-policies/image-integrity). and to the following points below:

Please note that all key data shown in the main figures as cropped gels or blots must be presented in uncropped form, with molecular weight markers. These data can be aggregated into a single supplementary figure. While these data can be displayed in a relatively informal style, they must refer back to the relevant figures.

SOURCE DATA: we request that authors provide, in tabular form, the data underlying the graphical representations used in figures. This is to further increase transparency in data

reporting, as detailed in this editorial (<http://www.nature.com/nsmb/journal/v22/n10/full/nsmb.3110.html>). Spreadsheets can be submitted in excel format. Only one (1) file per figure is permitted; thus, for multi-paneled figures, the source data for each panel should be clearly labeled in the Excel file; alternately the data can be provided as multiple, clearly labeled sheets in an Excel file. When submitting files, the title field should indicate which figure the source data pertains to.

We require deposition of coordinates (and, in the case of crystal structures, structure factors) into the Protein Data Bank with the designation of immediate release upon publication (HPUB). Electron microscopy-derived density maps and coordinate data must be deposited in EMDB and released upon publication. Deposition and immediate release of NMR chemical shift assignments are highly encouraged. Deposition of deep sequencing and microarray data is mandatory, and the datasets must be released prior to or upon publication. To avoid delays in publication, dataset accession numbers must be supplied with the final accepted manuscript and appropriate release dates must be indicated at the galley proof stage. Please find the complete NRG policies on data availability at <http://www.nature.com/authors/policies/availability.html>.

Nature Structural & Molecular Biology is committed to improving transparency in authorship. As part of our efforts in this direction, we are now requesting that all authors identified as 'corresponding author' on published papers create and link their Open Researcher and Contributor Identifier (ORCID) with their account on the Manuscript Tracking System (MTS), prior to acceptance. This applies to primary research papers only. ORCID helps the scientific community achieve unambiguous attribution of all scholarly contributions. You can create and link your ORCID from the home page of the MTS by clicking on 'Modify my Springer Nature account'. For more information please visit please visit www.springernature.com/orcid.

[redacted]

Sincerely,

Katarzyna Ciazynska
(she/her)
Associate Editor
Nature Structural & Molecular Biology
<https://orcid.org/0000-0002-9899-2428>

Referee expertise:

Referee #1: Ub signalling, stress response

Referee #2: DPC

Referee #3: SUMO

Reviewers' Comments:

Reviewer #1:

Remarks to the Author:

In this study, Liu and co-workers identify TOPORS as a novel SUMO-targeted ubiquitin ligase (STUbL) that plays an important role in the removal of DNA protein crosslinks and protein quality control. These are roles that have been previously linked to the STUbL RNF4. The authors argue that RNF4 and TOPORS jointly regulate these processes. Instead of functioning as an E3 ubiquitin ligase like RNF4, they provide evidence that TOPORS extends pre-existing ubiquitin modifications on SUMOylated proteins. In contrast to RNF4, TOPORS modifies the substrates primarily with K48-linked ubiquitin chains. This results in the recruitment of p97/VCP and proteasomal degradation of ubiquitylated substrates. The identification of another STUbL involved in controlling genotoxic and proteotoxic stress is important and opens new avenues to study the role of SUMO modifications in cell viability. The observation that TOPORS is not "just" another RNF4 but displays distinct features, targeting different substrates and modifying them with different chains makes this finding even more intriguing. The claims are well documented with high quality and convincing experiments that approach this phenomenon from different angles. Based on the presented data, I suspect that TOPORS will receive a lot of attention from the scientific community and will be of interest to the broad readership of NSMB.

Comments

The authors argue that RNF4 and TOPORS jointly regulate these processes. However, from their description it is not always clear what they mean with "joint". Do they mean that TOPORS acts downstream of RNF4 by modifying substrates that have been primed by RNF4 with ubiquitin modifications that are subsequently being extended by TOPORS and/or do they mean that it acts in a parallel pathway that targets a different (maybe partially overlapping) set of substrates. The fact that TOPORS targets primarily SUMO1 modified substrates whereas RNF4 is specific for SUMO2/3 modified substrates combined with the synthetic lethality seem to argue in favor of the latter. The authors could be more explicit in their description of the proposed model. Including a schematic model of the joint regulation of these processes by RNF4 and TOPORS in the main figures would also be helpful.

In Figure 2K, the authors show that TOPORS preferably modifies 4xSUMO2-ubiquitin fusion on which they base the claim that TOPORS extends ubiquitin chains on substrates that have been already ubiquitylated by RNF4. However, in Figure 4F they present data that TOPORS primarily targets SUMO1 modified proteins which are not substrates for RNF4. They also speculate that the spaced SUMO interacting motifs (SIMs) may be better suited for binding proteins with multiple SUMO1 modifications instead of SUMO2/3 chains.

This suggests that the artificial substrate used in Figure 2K may not be a natural substrate for TOPORS. It would be informative to see if TOPORS can also ubiquitylate a similar 4xSUMO1-ubiquitin fusion protein. The spacing should not be a problem as the SIMs in TOPORS seem to be able to bind linear 4xSUMO2 and should therefore also be able to bind 4xSUMO1.

If TOPORS ubiquitylates SUMO1-modified proteins that already contain a ubiquitin modification, does this suggest that it acts downstream of another STUbL that modifies SUMO1 modified proteins or does it target SUMO1 proteins that have been provided by a ubiquitin modification independent of the SUMO1 modification? Could there be a role for RNF111?

In Fig 3D, there appears to be a discrepancy between the two TOPORS-specific siRNA. Whereas siTOPORS1 + siRNF4 increases SUMO1 conjugates with little or no effect on SUMO2/3 conjugates, siTOPORS2 + siRNF4 increases SUMO2/3 with little or no effect on SUMO1 conjugates. Is this a representative experiment and, if so, please comment on this observation.

Reviewer #2:

Remarks to the Author:

Liu, Mailand and colleagues uncover the implication of TOPORS in DNMT1-associated DNA-protein complexes and confirm the importance of TOPORS for the degradation of PML in response to arsenic trioxide (As₂O₃).

Overall comments:

The novelty of the study lies in the connection of TOPORS with USP7 and in the detailed analyses of Mass Spectrometry/screening analyses and biochemical assays showing how TOPORS and USP7 regulate DNMT1 and PML ubiquitylation and appear coordinated with the other STUbL RNF4. The authors also demonstrate the synthetic lethality of RNF4 with TOPORS and USP7 inhibitors.

1. We view the relevance of the DNMT1 as being limited as DNMT1-DPCs are non-physiological and only induced by 5-azadC derivatives and zebularine. TOPORS was discovered as binding to both TOP1 and TP53 (please cite and credit previous work such as Rasheed et al. 2002 Exp Cell Res). Yet, the paragraph titles and text are misleading as they state that TOPORS is involved in (general) DPC repair (for instance see line 170, and highlight #1), which is questionable. To make that claim, the authors need to show that TOPORS is involved in physiological (and other pharmacological) DPCs such as those induced by topoisomerases.
2. TOPORS was already known to be an U3 ligase and Rudin and coworkers had identified key residues and domain (please see and cite Rasheed's publications).
3. RNF4 was first identified as a general STUbL for the ubiquitous DPCs induced by topoisomerases in ref. 15. This reference should be cited line 77 together with ref. 11, and line 178 as it precedes ref. 14.
4. TOPORS-induced degradation of PML was previously established. Yet, the paper reads as if the authors made the discovery. We suggest that the authors limit the overstatement of their claims. Also, can the authors discuss how As₂O₃ induces? Is it by the induction of DPCs?

Specific comments:

1. Chromatin fractionation enriches both covalent (DPCs) and non-covalent DNA-bound proteins, and the contamination of non-covalent DNA-bound DNMT species is not negligible. It is suggested to perform the RADAR assay or ICE assay to confirm 5-AzadC-induced DNMT-DPCs and their ubiquitylation and SUMOylation.
2. Figure 1C SUMO1 and SUMO2/3 blots look like duplicates. Please provide uncropped blots for all the SUMO blots if possible. Also, Figure 1C should include a ubiquitin blot.
3. Double knock-down of RNF4 and TOPORS on DNMT-DPC removal needs to be looked at.
4. Does 5-AzadC induce TOPORS deubiquitylation as an activation mechanism by promoting the binding of USP7 to TOPORS? This needs to be investigated. TOPORS depletion experiment needs to include a proteasome inhibitor (Figure 2H).
5. Figure 3G shows siRNF4 destabilizes TOPORS (lane 2). Please provide an explanation and better immunoblot for TOPORS. Also, this blot needs biological triplicates and quantitation for SUMO as the differences between samples are quite minor.
6. Figure 3H also requires quantitation. And why does USP7i alone reduce global SUMOylation? Based on the proposed STUbL activity of TOPORS, one would expect increase SUMOylation. Does this relate to the SUMO ligase activity of TOPORS (see Hammer, E. et al. 2007; <https://www.ncbi.nlm.nih.gov/pubmed/17976381>)?
7. Figure 4F SUMO blots require triplicate and quantitation.
8. What is the endpoint of the pathway for the DPC clearance? p97, proteasome or proteases like SPRTN? Please look at DPC levels using their inhibitors or siRNA.

Reviewer #3:

Remarks to the Author:

SUMO-mediated ubiquitylation of proteins represents an important signaling process safeguarding cellular genome and proteome stability in response to genotoxic or proteotoxic stress. This is best exemplified by the SUMO-dependent resolution of DNA-protein crosslinks or the arsenic-induced degradation of PML. In both processes the SUMO-targeted ubiquitin ligase (StUbL) RNF4 has been defined as the key factor mediating ubiquitylation and subsequent proteasomal degradation of polysumoylated targets. In this manuscript Liu et al. identify the RING-type ubiquitin ligase Topors as an additional player in SUMO-targeted ubiquitylation. The authors demonstrate that RNF4 and Topors cooperate in the resolution of DNMT1-DNA crosslinks induced by 5-AzaDC as well as in arsenic induced degradation of PML. They further propose that RNF4 primarily acts on SUMO2 conjugates triggering the formation of K63-linked Ub chains, while Topors preferentially targets SUMO1 conjugates and mediates K48-linked ubiquitylation. The authors conclude that the concerted activities of both ligases the formation of complex ubiquitin-SUMO chains that can be recognized by the p97/VCP machinery. Altogether, the identification of Topors as a novel StUBL is a very important, original and significant finding. The data generated from unbiased screenings are very conclusive and convincing. Furthermore, the phenotypes observed upon co-depletion of RNF4 and Topors strongly support the conclusion of a cooperation between RNF4 and Topors in SUMO-

mediated ubiquitylation. However, experiments exploring the molecular mechanism of this cooperation are not fully conclusive yet. In particular, the conclusion that RNF4 and Topors act on SUMO2 or SUMO1 conjugates, respectively, is based on relatively sparse data. Furthermore, data on Ub-linkage specificity of both ligases should be deepened. To my opinion, these parts of the manuscript need to be strengthened prior to publication.

Major points:

- 1) The authors use ubiquitin-dependent resolution/clearance of DNMT1-DNA crosslinks as one model system for the cooperative action of RNF4 and Topors. These data are very strong and convincing. To expand on this finding, it would be important to investigate whether Topors is generally involved in the RNF4-mediated resolution of DPCs. Sun et al. have shown that RNF4-mediated ubiquitylation degrades topoisomerase DNA-protein crosslinks induced by etoposide (Sci Advances, 2020). Does Topors also affect this process? In contrast to the model proposed here, Sun et al. did not observe impaired ubiquitylation of Top1/Top2 DPC in the absence of Topors (Sun et al., Figure 3). Clarifying this point would be important.
- 2) As mentioned above, the conclusion that RNF4 and Topors act on SUMO2 or SUMO1 conjugates, respectively, is based on relatively sparse data. In Figure 2I the authors clearly show that Topors strongly binds to polySUMO2 chains contradicting the hypothesis that Topors is primarily recruited to SUMO1-conjugates. Furthermore, the in vitro experiment in Figure 2K was performed with a 4xSUMO2 STUbL substrates indicating that Topors can catalyze ubiquitylation of a linear polySUMO2 conjugate. The role of SUMO1 vs. SUMO2 in RNF4/Topors-mediated ubiquitylation needs to be clarified by cellular depletion of SUMO1 or SUMO2/3 followed by analysis of DNMT1 and/or PML ubiquitylation.
- 3) Ideally, the authors should reconstitute Topors/RNF4 mediated ubiquitylation of PML in a purified system. This would clarify the requirement/preference of SUMO1 vs. SUMO2 and potentially reveal a sequential action of both ligases.
- 4) The data on formation of K63 vs. K48 chains are also not fully conclusive at this stage. The conclusion primarily relies on proteomics data obtained on DNMT1. To strengthen this point and to generalize the proposed model it would be important to reveal changes in ubiquitin linkages on PML upon depletion of Topors or RNF4. Given the availability of linkage-specific antibodies this should be doable.
- 5) The data on DRIP clearance are not fully conclusive. The authors claim that RNF4 has been implicated in DRIP clearance. However, at least to my knowledge this has not been shown (at least references 39-41 did not show RNF4-dependent clearance of DRIPs). Therefore, the mechanism of RNF4/Topors mediated DRIP clearance described here remains somewhat unclear. Does this directly rely on ubiquitylation of DRIPs? Although I do acknowledge that the initial observation of RNF4/Topors-mediated DRIP clearance is interesting, I do feel that at this stage the data are too preliminary to be included in the manuscript.

Author Rebuttal to Initial comments

Point-by-point reply to the referees' comments

We would like to thank the reviewers for their constructive and insightful feedback on our study. In the revised version of our manuscript, we have included the results of a range of new experiments performed on the basis of the reviewers' helpful suggestions. Collectively, we believe the new additions to the manuscript address all of the referees' concerns, corroborating and extending the conclusion that TOPORS is a novel STUbL that acts in parallel with RNF4 in driving major SUMO-dependent cellular processes including DPC repair and PML body degradation. In particular, we now clarify and provide improved insights into the nature of the partnership between TOPORS and RNF4 by demonstrating more directly that TOPORS STUbL activity preferentially targets SUMO1-modified proteins. In the following, we provide a detailed point-by-point response to the referee reports. The reviewers' comments (reproduced in their entirety) are shown in bold and italicized text, while our responses are in plain text. With all the new additions, we hope the reviewers find our paper improved and suitable for publication in Nature Structural & Molecular Biology.

Reviewer #1:

In this study, Liu and co-workers identify TOPORS as a novel SUMO-targeted ubiquitin ligase (STUbL) that plays an important role in the removal of DNA protein crosslinks and protein quality control. These are roles that have been previously linked to the STUbL RNF4. The authors argue that RNF4 and TOPORS jointly regulate these processes. Instead of functioning as an E3 ubiquitin ligase like RNF4, they provide evidence that TOPORS extends pre-existing ubiquitin modifications on SUMOylated proteins. In contrast to RNF4, TOPORS modifies the substrates primarily with K48-linked ubiquitin chains. This results in the recruitment of p97/VCP and proteasomal degradation of ubiquitylated substrates. The identification of another STUbL involved in controlling genotoxic and proteotoxic stress is important and opens new avenues to study the role of SUMO modifications in cell viability. The observation that TOPORS is not "just" another RNF4 but displays distinct features, targeting different substrates and modifying them with different chains makes this finding even more intriguing. The claims are well documented with high quality and convincing experiments that approach this phenomenon from different angles. Based on the presented data, I suspect that TOPORS will receive a lot of attention from the scientific community and will be of interest to the broad readership of NSMB.

Comments

The authors argue that RNF4 and TOPORS jointly regulate these processes. However, from their description it is not always clear what they mean with "joint". Do they mean that TOPORS acts downstream of RNF4 by modifying substrates that have been primed by RNF4 with ubiquitin modifications that are subsequently being extended by TOPORS and/or do they mean that it acts in a parallel pathway that targets a different (maybe partially overlapping) set of substrates. The fact that TOPORS targets primarily SUMO1 modifies substrates whereas RNF4 is specific for SUMO2/3 modified substrates combined with the synthetic lethality seem to argue in favor of the latter. The authors could be more explicit in their description of the proposed model. Including a schematic model of the joint regulation of these processes by RNF4 and TOPORS in the main figures would also be helpful.

This is an important point. We have performed several new experiments that now enable us to better define the nature of the partnership between TOPORS and RNF4. In particular, as elaborated in our response to the point below, we added new data to the manuscript strengthening the notion that TOPORS preferentially targets SUMO1-modified proteins and does not act downstream of other STUbLs in this process (new Figure 4F,G). Moreover, while TOPORS binds efficiently to poly-SUMO2 chains *in vitro* (Figure 2I), we observed using agarose-conjugated SUMO1 and SUMO2 that it displays some preference for interacting with SUMO1 (new Figure 4H), whereas RNF4 showed an inverse selectivity in line with previous findings (new Figure 4I) (Aguilar-Martinez *et al.*, 2015; Gonzalez-Prieto *et al.*, 2021). We also show in cell-based experiments that depletion of RNF4 and SUMO1 have additive effects on reducing DNMT1 DPC ubiquitylation and resolution, mimicking the impact of co-depleting RNF4 and TOPORS (new Extended Data Figure 5H,I). Together, these findings add weight to the notion that TOPORS and RNF4 act in parallel in STUbL pathways, with TOPORS being particularly important for ubiquitylating SUMO1-modified proteins. Indeed, in line with the reviewer's comment, such a relationship involving complementary but partially overlapping functions helps to explain the synthetic lethal interaction between TOPORS and RNF4, as in the absence of either E3 STUbL pathways may be compromised while retaining basic functionality, whereas combined loss of TOPORS and RNF4 imposes full deficiency of these processes. We have added a schematic model illustrating this relationship between TOPORS and RNF4 in STUbL pathways (new Figure 6), and we carefully edited the manuscript text to more clearly explain and specify the nature of the partnership between these proteins in driving STUbL pathways.

In Figure 2K, the authors show that TOPORS preferably modifies 4xSUMO2-ubiquitin fusion on which they base the claim that TOPORS extends ubiquitin chains on substrates that have been already ubiquitylated by RNF4. However, in Figure 4F they present data that TOPORS primarily targets SUMO1 modified proteins which are not substrates for RNF4. They also speculate that the spaced SUMO interacting motifs (SIMs) may be better suited for binding proteins with multiple SUMO1 modifications instead of SUMO2/3 chains. This suggests that the artificial substrate used in Figure 2K may not be a natural substrate for TOPORS. It would be informative to see if TOPORS can also ubiquitylate a similar 4xSUMO1-ubiquitin fusion protein. The spacing should not be a problem as the SIMs in TOPORS seem to be able to bind linear 4xSUMO2 and should therefore also be able to bind 4xSUMO1.

We fully agree with the reviewer's point. Indeed, we also surmised that the SUMO2 substrate used in Figure 2K may not be an optimal TOPORS substrate. Prompted by this, we tested the ability of TOPORS to ubiquitylate a poly-SUMO1-modified substrate and how this compares with its STUbL activity towards a comparable SUMO2 substrate. To this end, we decided against using a linear 4xSUMO1 fusion protein, as SUMO1 mostly does not form homotypic chains unlike SUMO2/3. Instead, we took advantage of previously described SUMO substrates in which SUMO1 or SUMO2 is fused to a COMP pentamerization domain (Aguilar-Martinez *et al.*, 2015) (new Figure 4F), giving rise to COMP-SUMO model substrates resembling multi-mono-SUMO1- or -SUMO2/3-modified proteins, which in the case of SUMO1 may more faithfully recapitulate SUMO1-modified proteins in cells than a linear 4xSUMO1 fusion. Importantly, using these SUMO substrates, we found that purified TOPORS displays markedly higher E3 ubiquitin ligase activity towards COMP-SUMO1 than both a corresponding COMP-SUMO2 substrate and monomeric SUMO1 *in vitro* (new Figure 4F,G). These data provide direct evidence that TOPORS preferentially targets SUMO1-modified proteins, in excellent agreement with our finding that TOPORS loss strongly

decreases the level of proteins co-modified by ubiquitin and SUMO1, but not SUMO2/3 (Figure 4E).

If TOPORS ubiquitylates SUMO1-modified proteins that already contain a ubiquitin modification, does this suggest that it acts downstream of another STUbL that modifies SUMO1 modified proteins or does it targets SUMO1 proteins that have been provided by a ubiquitin modification independent of the SUMO1 modification? Could there be a role for RNF111?

As described above, we found that TOPORS displays a clear preference for ubiquitylating SUMO1-modified proteins relative to SUMO2/3-modified proteins *in vitro* (new Figure 4F,G). Importantly, TOPORS-dependent ubiquitylation of a multi-mono-SUMO1 substrate (GST-COMP-SUMO1) occurs efficiently in the absence of its prior modification by ubiquitin (new Figure 4F,G), suggesting that TOPORS does not act downstream of other STUbLs in ubiquitylating SUMO1 substrates. In line with this, we have previously investigated whether RNF111 contributes to SUMO-dependent DNMT1 DPC resolution and residual DNMT1 DPC ubiquitylation in the absence of RNF4 but found no significant role of RNF111 in these processes (see Figure EV1E,F in (Liu *et al*, 2021)). Thus, in keeping with the notion that co-depletion of TOPORS and RNF4 strongly suppresses ubiquitylation of SUMOylated proteins (Figure 2C; Figure 3D), these observations suggest that TOPORS and RNF4 jointly contribute the bulk of STUbL activity impacting SUMO-modified proteins.

In Fig 3D, there appears to be a discrepancy between the two TOPORS-specific siRNA. Whereas siTOPORS1 + siRNF4 increases SUMO1 conjugates with little or no effect on SUMO2/3 conjugates, siTOPORS2 + siRNF4 increases SUMO2/3 with little or no effect on SUMO1 conjugates. Is this a representative experiment and, if so, please comment on this observation.

This apparent discrepancy between the TOPORS siRNAs was unique to this particular experiment and does not represent a consistent effect across a range of replicates. To avoid confusion, we have replaced the experiment originally shown in Figure 3D with a more representative replicate. From this it can be appreciated that some SUMO1 modification of YFP-PML is apparent even in the absence of arsenic upon co-depletion of RNF4 and TOPORS by siRNF4+siTOPORS#1 or siRNF4+siTOPORS#2. This likely reflects impaired ubiquitin-dependent turnover of SUMOylated PML, which occurs at a low level in unperturbed cells, further supporting an important role of TOPORS and RNF4 STUbL activities in jointly promoting PML body degradation.

Reviewer #2:

Liu, Mailand and colleagues uncover the implication of TOPORS in DNMT1-associated DNA-protein complexes and confirm the importance of TOPORS for the degradation of PML in response to arsenic trioxide (As₂O₃).

Overall comments:

The novelty of the study lies in the connection of TOPORS with USP7 and in the detailed analyses of MassSpectrometry/screening analyses and biochemical assays showing how TOPORS and USP7 regulate DNMT1 and PML ubiquitylation and appear coordinated with the other STUbL RNF4. The authors also demonstrate the synthetic lethality of RNF4

with TOPORS and USP7 inhibitors.

1. We view the relevance of the DNMT1 as being limited as DNMT1-DPCs are non-physiological and only induced by 5-azadC derivatives and zebularine. TOPORS was discovered as binding to both TOP1 and TP53 (please cite and credit previous work such as Rasheed et al. 2002 Exp Cell Res). Yet, the paragraph titles and text are misleading as they state that TOPORS is involved in (general) DPC repair (for instance see line 170, and highlight #1), which is questionable. To make that claim, the authors need to show that TOPORS is involved in physiological (and other pharmacological) DPCs such as those induced by topoisomerases.

In the original manuscript, we showed that TOPORS is not only important for resolution of DNMT1 DPCs but also physiological formaldehyde-induced DPCs (Extended Data Figure 1F-H), suggesting that TOPORS has a more general role in SUMO-mediated DPC repair that is not restricted to DNMT1 DPCs. It should also be mentioned in this context that very recent work from Yves Pommier's group showed that DNMT1 is among the endogenous cellular proteins that are most prone to formaldehyde-induced DPC formation (Sun *et al*, 2023). That said, we do agree with the reviewer that it is relevant to address whether TOPORS is involved in the processing of topoisomerase DPCs, which like DNMT1 DPCs are also known to be targeted by SUMOylation and RNF4 (Sun *et al*, 2020). In the revised manuscript, we included new data showing that both TOPORS and RNF4 contribute to SUMO-dependent ubiquitylation of Topoisomerase I (TOP1) DPCs induced by treatment with camptothecin, and that co-depletion of TOPORS and RNF4 strongly reduces TOP1 DPC ubiquitylation (new Extended Data Figure 2B,C), mirroring our findings for DNMT1 DPCs and PML. Thus, TOPORS is important for ubiquitylating several different types of DPCs that are processed in a SUMO-dependent manner. As suggested by the reviewer, we now cite the paper by Rubin and co-workers reporting an interaction between TOPORS and TOP1 (Haluska *et al*, 1999) (line 236, page 8).

2. TOPORS was already known to be an U3 ligase and Rudin and coworkers had identified key residues and domain (please see and cite Rasheed's publications).

Indeed, as we state clearly in the manuscript, TOPORS has already been shown to be an E3 ubiquitin ligase; as suggested by the reviewer, we now cite the study by Rasheed et al. (Rasheed *et al*, 2002) that first identified the RING domain in TOPORS along with other relevant publications (line 223, page 8). However, as TOPORS has also been reported to possess SUMO E3 ligase activity, it has been unclear whether TOPORS mainly acts as an E3 for ubiquitin, SUMO, or both. We therefore believe that our *in vitro* TOPORS E3 ubiquitin and SUMO ligase activity assays revealing that TOPORS mainly displays E3 ligase activity towards ubiquitin are relevant to include. Moreover, one of the salient new findings of our study is that TOPORS defines a novel STUbL, thus offering an important extension and refinement of our understanding of the E3 ligase function of TOPORS, which in our opinion has hitherto been relatively unclear.

3. RNF4 was first identified as a general STUbL for the ubiquitous DPCs induced by topoisomerases in ref. 15. This reference should be cited line 77 together with ref. 11, and line 178 as it precedes ref. 14.

We concur with this notion and have made the suggested changes to the revised manuscript.

4. TOPORS-induced degradation of PML was previously established. Yet, the paper reads as if the authors made the discovery. We suggest that the authors limit the overstatement of their claims. Also, can the authors discuss how As2O3 induces? Is it by the induction of DPCs?

We are not aware of any published work demonstrating that TOPORS is required for arsenic-induced PML body degradation. We surmise that the reviewer might be referring to a recent study reporting that TOPORS-depleted cells display a slight decrease in PML body number (Ji *et al*, 2020). However, this paper did not address whether TOPORS is involved in PML body degradation.

Although arsenic-induced PML body degradation by SUMOylation and STUbL activity has been studied extensively during the past decades, the mechanistic basis of how arsenic triggers PML SUMOylation remains elusive. Recent data suggest that arsenic binding to PML may induce conformational changes in PML body assemblies via direct binding to PML, triggering its SUMOylation (Bercier *et al*, 2023). However, there is no evidence to suggest that this involves arsenic-induced DPC formation.

Specific comments:

1. Chromatin fractionation enriches both covalent (DPCs) and non-covalent DNA-bound proteins, and the contamination of non-covalent DNA-bound DNMT species is not negligible. It is suggested to perform the RADAR assay or ICE assay to confirm 5-AzadC-induced DNMT-DPCs and their ubiquitylation and SUMOylation.

We carefully optimized the chromatin fractionation protocol employed in this study to minimize the presence of non-covalent DNA-bound DNMT1 species, by using a high-salt buffer for stringent washing of chromatin fractions (lanes 3 and 4 in Reviewer Figure 1 below). While we occasionally observe a low level of unmodified DNMT1 in chromatin fractions from unperturbed cells (e.g. Figure 1C), we do not think this has significant bearings on our conclusion that TOPORS is important for DNMT1 DPC ubiquitylation and resolution. Of relevance to this, we replaced the chromatin fractionation experiment shown in Extended Data Figure 1J with a different replicate in which the level of chromatin-associated, unmodified DNMT1 in mock-treated cells was considerably lower.

Reviewer Figure 1. Chromatin fractionation using mild or harsh (this study) buffer conditions
U2OS cells released from single-round thymidine synchronization in early S phase were treated or not with 5-AzadC for 30 min, collected and subjected to subcellular fractionation. Chromatin-enriched fractions were washed extensively using mild (no NaCl) or harsh (500 mM NaCl) buffer conditions and immunoblotted with DNMT1 and histone H3 antibodies. Stringent washing, as done in this study, diminishes the presence of unmodified, non-covalently bound DNMT1 in the chromatin fractions (compare lanes 1 and 3) but has no impact on levels of DNMT1 DPCs (compare lanes 2 and 4).

Unfortunately, we have found that the RADAR assay is not sufficiently sensitive for consistent detection of 5-AzadC-induced DNMT1 DPCs. Importantly, however, a recent study from the Stingele lab described the establishment of an improved method for isolating DPCs, Purification of x-linked Proteins (PxP), and showed that our key findings on DNMT1 DPC SUMOylation/ubiquitylation and resolution obtained using the chromatin fractionation protocol applied in the present study can be fully recapitulated by the orthogonal PxP method (Weickert *et al*, 2023). We believe this validates our chromatin fractionation approach as an appropriate and reliable method for probing DNMT1 DPC processing and repair.

2. Figure 1C SUMO1 and SUMO2/3 blots look like duplicates. Please provide uncropped blots for all the SUMO blots if possible. Also, Figure 1C should include a ubiquitin blot.

The SUMO blots shown in Figure 1C were not duplicates, but when immunoblotting DNMT1 IPs or chromatin fractions from 5-AzadC-treated cells, where DNMT1 is by far the most prominently SUMOylated protein, the SUMO1 and SUMO2/3 antibodies we used (which are highly specific for recognizing their respective SUMO isoforms (e.g. Figure 4H,I)) tend to produce a very similar band pattern (see Figure 2C for another example) given that DNMT1 is abundantly modified by both SUMO1 and SUMO2/3 upon 5-AzadC treatment (Extended Data Figure 2E). We have exchanged the results previously shown in Figure 1C with data from another replicate where the difference between the SUMO1 and SUMO2/3 blots may be easier to appreciate (new Figure 1C). We also supply uncropped blots for this experiment and all other immunoblots shown in our study (see Source Data files).

Reviewer Figure 2. Ubiquitin blot for Figure 1C

HAP1 WT or TOPORS-KO cell lines released from single-round thymidine synchronization in early S phase were treated with 5-AzadC for 30 min, then washed and collected at the indicated times. Chromatin-enriched fractions were immunoblotted with ubiquitin and histone H3 antibodies.

We also immunoblotted the chromatin fractions in Figure 1C with a ubiquitin antibody (Reviewer Figure 2). As ubiquitylation of chromatin-associated proteins is very substantial even in unperturbed cells, no pronounced increase in chromatin ubiquitylation upon 5-AzadC treatment is apparent (Reviewer Figure 2), unlike the prominent SUMOylation changes. Therefore, we believe including a ubiquitin blot in Figure 1C is of limited value, but we can add it if deemed necessary.

3. Double knock-down of RNF4 and TOPORS on DNMT-DPC removal needs to be looked at.

We added new data showing that co-depleting TOPORS and RNF4 fully blocks DNMT1 DPC resolution (new Figure S2D), consistent with the abrogation of DNMT1 DPC ubiquitylation under these conditions (Figure 2C).

4. Does 5-AzadC induce TOPORS deubiquitylation as an activation mechanism by promoting the binding of USP7 to TOPORS? This needs to be investigated. TOPORS depletion experiment needs to include a proteasome inhibitor (Figure 2H).

This is an interesting suggestion, which we tested by analyzing whether TOPORS ubiquitylation status is impacted by 5-AzadC treatment. As shown in Reviewer Figure 3 below, we observed no 5-AzadC-induced change in TOPORS ubiquitylation, suggesting that USP7 does not regulate TOPORS ubiquitylation and stability upon 5-AzadC treatment.

We exchanged the data previously shown in Figure 1H with a new experiment showing that the rapid loss of TOPORS induced by USP7i treatment is fully rescued by the proteasome inhibitor MG132 (new Figure 1H), providing further evidence that USP7 underpins TOPORS stability by antagonizing its auto-ubiquitylation and subsequent proteasomal degradation.

Reviewer Figure 3. TOPORS ubiquitylation status is not altered by 5-AzadC treatment

U2OS cells transfected with GFP-TOPORS WT expression construct were treated or not with 5-AzadC for 30 min, subjected to GFP IP under denaturing conditions and immunoblotted with ubiquitin and GFP antibodies.

5. Figure 3G shows siRNF4 destabilizes TOPORS (lane 2). Please provide an explanation and better immunoblot for TOPORS. Also, this blot needs biological triplicates and quantitation for SUMO as the differences between samples are quite minor.

We apologize for the inferior quality of the TOPORS blot shown in this figure. We have replaced this with a new and improved blot, from which it can be seen that RNF4 depletion has little impact on TOPORS abundance (new Figure 3F; see also new Supplementary Figure

1A). The key conclusion of this experiment is that co-depletion of TOPORS and RNF4, but not knockdown of each individual STUbL, leads to a strong increase in hyper-SUMOylated proteins whose migration in SDS-PAGE are highly retarded (marked by arrows in Figure 3F). We quantified the abundance of these hyper-SUMOylated species in three independent replicates, which consistently show that co-depletion of TOPORS and RNF4, but not individual knockdown of either ligase, leads to a considerable accumulation of hyper-SUMOylated cellular proteins (new Supplementary Figure 1A).

6. Figure 3H also requires quantitation. And why does USP7i alone reduce global SUMOylation? Based on the proposed STUbL activity of TOPORS, one would expect increase SUMOylation. Does this relate to the SUMO ligase activity of TOPORS (see Hammer, E. et al. 2007; <https://www.ncbi.nlm.nih.gov/pubmed/17976381>)?

As done for Figure 3F above, we have now quantified the levels of slow-migrating hyper-SUMOylated proteins (marked by arrows) in previous Figure 3H (Figure 3G in the revised manuscript) in three independent experiments (new Supplementary Figure 1B). The USP7i-dependent decrease in global SUMOylation that was apparent in the experiment shown was unique to this particular repeat and does not represent a consistent effect across a range of replicates. We have therefore replaced this experiment with a more representative replicate (new Figure 3G; see also new Supplementary Figure 1B).

7. Figure 4F SUMO blots require triplicate and quantitation.

We quantified the abundance of SUMO1 and SUMO2/3 conjugates in three independent repeats of the experiment in previous Figure 4F (now Figure 4E), supporting the notion that TOPORS depletion markedly decreases the abundance of proteins co-modified by SUMO1 and ubiquitin (new Supplementary Figure 2). In agreement with this finding, we have added new data providing more direct evidence that TOPORS STUbL activity is mainly directed towards SUMO1-modified proteins (new Figure 4F-H).

8. What is the endpoint of the pathway for the DPC clearance? p97, proteasome or proteases like SPRTN? Please look at DPC levels using their inhibitors or siRNA.

Our previous work, along with a more recent study (Weickert *et al.*, 2023), showed that the clearance of SUMO-modified DPCs is predominantly mediated by the proteasome. Specifically, we showed that the proteasome inhibitor MG132 quantitatively blocks DNMT1 DPC resolution (please see Figure 1C in (Liu *et al.*, 2021)), while SPRTN has a more limited role in processing these lesions (Weickert *et al.*, 2023). We also show in the current study, in full agreement with recently published work (Weickert *et al.*, 2023), that p97 activity is essential for processing SUMOylated DPCs (Extended Data Figure 5A). This likely reflects the recruitment of p97 to, and ensuing unfolding of, SUMOylated DPCs ubiquitylated by RNF4 and TOPORS, as depletion of TOPORS and RNF4 both reduce p97 enrichment at DNMT1 DPC sites (Figure 4B).

Reviewer #3:

SUMO-mediated ubiquitylation of proteins represents an important signaling process safeguarding cellular genome and proteome stability in response to genotoxic or proteotoxic stress. This is best exemplified by the SUMO-dependent resolution of DNA-

protein crosslinks or the arsenic-induced degradation of PML. In both processes the SUMO-targeted ubiquitin ligase (StUbl) RNF4 has been defined as the key factor mediating ubiquitylation and subsequent proteasomal degradation of polysumoylated targets. In this manuscript Liu et al. identify the RING-type ubiquitin ligase Topors as an additional player in SUMO-targeted ubiquitylation. The authors demonstrate that RNF4 and Topors cooperate in the resolution of DNMT1-DNA crosslinks induced by 5-AzaDC as well as in arsenic induced degradation of PML. They further propose that RNF4 primarily acts on SUMO2 conjugates triggering the formation of K63-linked Ub chains, while Topors preferentially targets SUMO1 conjugates and mediates K48-linked ubiquitylation. The authors conclude that the concerted activities of both ligases the formation of complex ubiquitin-SUMO chains that can be recognized by the p97/VCP machinery. Altogether, the identification of Topors as a novel StUBL is a very important, original and significant finding. The data generated from unbiased screenings are very conclusive and convincing. Furthermore, the phenotypes observed upon co-depletion of RNF4 and Topors strongly support the conclusion of a cooperation between RNF4 and Topors in SUMO-mediated ubiquitylation. However, experiments exploring the molecular mechanism of this cooperation are not fully conclusive yet. In particular, the conclusion that RNF4 and Topors act on SUMO2 or SUMO1 conjugates, respectively, is based on relatively sparse data. Furthermore, data on Ub-linkage specificity of both ligases should be deepened. To my opinion, these parts of the manuscript need to be strengthened prior to publication.

Major points:

1) The authors use ubiquitin-dependent resolution/clearance of DNMT1-DNA crosslinks as one model system for the cooperative action of RNF4 and Topors. These data are very strong and convincing. To expand on this finding, it would be important to investigate whether Topors is generally involved in the RNF4-mediated resolution of DPCs. Sun et al. have shown that RNF4-mediated ubiquitylation degrades topoisomerase DNA-protein crosslinks induced by etoposide (Sci Advances, 2020). Does Topors also affect this process? In contrast to the model proposed here, Sun et al. did not observe impaired ubiquitylation of Top1/Top2 DPC in the absence of Topors (Sun et al., Figure 3). Clarifying this point would be important.

We agree with the reviewer that it is relevant to address whether TOPORS is involved in the processing of topoisomerase DPCs, which like DNMT1 DPCs are also known to be targeted by SUMOylation and RNF4-dependent ubiquitylation (Sun *et al.*, 2020). In the revised manuscript, we included new data showing that both TOPORS and RNF4 contribute to SUMO-dependent ubiquitylation of Topoisomerase I (TOP1) DPCs induced by treatment with camptothecin, and that co-depletion of TOPORS and RNF4 strongly reduces TOP1 DPC ubiquitylation (new Extended Data Figure 2B,C), mirroring our observations for DNMT1 DPCs and PML. Together with our findings on DNMT1 and formaldehyde-induced DPCs, these data show that TOPORS is important for ubiquitylating several different types of DPCs that are processed in a SUMO-dependent manner, acting in concert with RNF4.

In the study by Sun *et al.* mentioned by the reviewer, a moderate decrease in the ubiquitylation of TOP1 DPCs upon TOPORS knockdown was in fact apparent (Figure 3C in (Sun *et al.*, 2020)), although the effect may be somewhat less pronounced than what we observe (Extended Data Figure 2B). However, it should be pointed out in this regard that Sun *et al.* did not validate the knockdown efficiency of the single TOPORS siRNA they used, possibly due to a lack of a working TOPORS antibody. It is therefore possible that the

relatively mild impact of TOPORS depletion on TOP1 DPC ubiquitylation observed by Sun et al. could be due to incomplete knockdown of TOPORS.

2) As mentioned above, the conclusion that RNF4 and Topors act on SUMO2 or SUMO1 conjugates, respectively, is based on relatively sparse data. In Figure 2I the authors clearly show that Topors strongly binds to polySUMO2 chains contradicting the hypothesis that Topors is primarily recruited to SUMO1-conjugates. Furthermore, the *in vitro* experiment in Figure 2K was performed with a 4xSUMO2 STUbL substrates indicating that Topors can catalyze ubiquitylation of a linear polySUMO2 conjugate. The role of SUMO1 vs. SUMO2 in RNF4/Topors-mediated ubiquitylation needs to be clarified by cellular depletion of SUMO1 or SUMO2/3 followed by analysis of DNMT1 and/or PML ubiquitylation.

We agree that our study would be strengthened by additional evidence supporting differential contributions of TOPORS and RNF4 in ubiquitylating SUMO1- and SUMO2/3-modified proteins. While we found that TOPORS has STUbL activity towards a linear 4xSUMO2 substrate *in vitro* (Figure 2K), this reaction was relatively inefficient, suggesting that 4xSUMO2 may not be an optimal substrate for TOPORS STUbL activity. We therefore set out to directly compare the E3 ubiquitin ligase activity of TOPORS towards SUMO1 and SUMO2 substrates. To this end, we took advantage of previously described SUMO substrates in which SUMO is fused to a COMP pentamerization domain (Aguilar-Martinez *et al.*, 2015) (new Figure 4F), giving rise to COMP-SUMO model substrates resembling multi-mono-SUMO1- or -SUMO2/3-modified proteins. In the case of SUMO1, this may more faithfully mimic SUMO1-modified proteins in cells than a linear poly-SUMO1 fusion, considering that SUMO1 does not readily engage in chain formation unlike SUMO2/3. Importantly, using these SUMO substrates, we found that purified TOPORS displays markedly higher E3 ubiquitin ligase activity towards COMP-SUMO1 than a corresponding COMP-SUMO2 substrate and monomeric SUMO1 *in vitro* (new Figure 4F,G). These data provide direct evidence that TOPORS preferentially targets SUMO1-modified proteins, in full agreement with our finding that TOPORS loss strongly decreases the level of proteins co-modified by ubiquitin and SUMO1, but not ubiquitin and SUMO2/3 (Figure 4E).

Adding to this, we followed the reviewer's excellent suggestion to test the impact of depleting SUMO1 and SUMO2/3 on STUbL-mediated ubiquitylation of DNMT1 DPCs. Interestingly, we found that RNF4 and SUMO1 knockdown has additive impacts on reducing DNMT1 DPC ubiquitylation, mimicking the effect of RNF4 and TOPORS co-depletion (new Extended Data Figure 5H). By contrast, such a relationship was not apparent for RNF4 and SUMO2/3 depletion (new Extended Data Figure 5H). Moreover, RNF4 and SUMO1 depletion delayed DNMT1 DPC clearance in an additive manner (new Extended Data Figure 5I). These cell-based findings further support the notion that the critical contribution of TOPORS STUbL activity mainly entails its preference for targeting SUMO1-modified proteins.

Finally, while TOPORS does indeed bind efficiently to poly-SUMO2 chains *in vitro* (Figure 2I), we observed using agarose-conjugated SUMO1 and SUMO2 that it displays some preference for interacting with SUMO1 (new Figure 4H), whereas RNF4 showed clear selectivity for binding SUMO2 in line with previous findings (new Figure 4I) (Aguilar-Martinez *et al.*, 2015; Gonzalez-Prieto *et al.*, 2021). Altogether, these findings suggests a division of labor between TOPORS and RNF4 in STUbL pathways where TOPORS mainly targets SUMO1-modified proteins, whereas RNF4 may have a preference for SUMO2/3-

modified proteins, although this selectivity is clearly not absolute. We have added a schematic model illustrating this partnership between TOPORS and RNF4 in STUbL pathways (new Figure 6).

3) Ideally, the authors should reconstitute Topors/RNF4 mediated ubiquitylation of PML in a purified system. This would clarify the requirement/preference of SUMO1 vs. SUMO2 and potentially reveal a sequential action of both ligases.

Reconstitution of arsenic-induced PML SUMOylation and subsequent STUbL-dependent ubiquitylation in a purified system is a long-standing major goal for the PML field. Unfortunately, however, this represents a formidable challenge, being principally due to the superior difficulty of producing functional purified full-length PML protein. Indeed, although arsenic-induced PML degradation has been studied for decades, the mechanistic basis of how arsenic exposure triggers PML SUMOylation remains fundamentally unclear, in large part reflecting the obstacles in reconstituting this process with purified components. Accordingly, reconstituting TOPORS- and RNF4-dependent ubiquitylation of SUMOylated PML *in vitro* is unfortunately not feasible within the scope of the current study.

4) The data on formation of K63 vs. K48 chains are also not fully conclusive at this stage. The conclusion primarily relies on proteomics data obtained on DNMT1. To strengthen this point and to generalize the proposed model it would be important to reveal changes in ubiquitin linkages on PML upon depletion of Topors or RNF4. Given the availability of linkage-specific antibodies this should be doable.

Using K48- and K63-linkage-specific ubiquitin antibodies, we show in Extended Data Figure 5G that depletion of RNF4, but not TOPORS, reduces the extent of arsenic-induced K63-linked ubiquitylation of PML, while knockdown of TOPORS diminishes PML modification by K48-linked ubiquitin conjugates in response to arsenic treatment. These data are consistent with our findings for DNMT1 DPCs, further suggesting that TOPORS mainly promotes K48-linked, but not K63-linked, ubiquitylation of SUMOylated target proteins.

5) The data on DRIP clearance are not fully conclusive. The authors claim that RNF4 has been implicated in DRIP clearance. However, at least to my knowledge this has not been shown (at least references 39-41 did not show RNF4-dependent clearance of DRIPs). Therefore, the mechanism of RNF4/Topors mediated DRIP clearance described here remains somewhat unclear. Does this directly rely on ubiquitylation of DRIPs? Although I do acknowledge that the initial observation of RNF4/Topors-mediated DRIP clearance is interesting, I do feel that at this stage the data are too preliminary to be included in the manuscript.

We concur with the reviewer's opinion. Indeed, unlike SUMO, RNF4 has not been directly implicated in DRiP clearance, although our data clearly support such an involvement. We have attempted to directly detect ubiquitin and SUMO modifications of OP-puro-labeled DRiPs using a click chemistry-based pulldown approach, with the aim to directly monitor the impact of RNF4 and TOPORS depletion on these modifications. However, while we did obtain supportive evidence for RNF4- and TOPORS-dependent DRiP ubiquitylation using this setup, these effects were somewhat inconsistent and may be better explored in a dedicated study on SUMO/ubiquitin-dependent DRiP processing. Therefore, in accordance with the reviewer's suggestion, we decided to remove the data on DRiPs from the

manuscript. We believe that the omission of these data does not compromise any conclusions made in this study.

References

- Aguilar-Martinez E, Chen X, Webber A, Mould AP, Seifert A, Hay RT, Sharrocks AD (2015) Screen for multi-SUMO-binding proteins reveals a multi-SIM-binding mechanism for recruitment of the transcriptional regulator ZMYM2 to chromatin. *Proc Natl Acad Sci U S A* 112: E4854-4863
- Bercier P, Wang QQ, Zang N, Zhang J, Yang C, Maimaitiyiming Y, Abou-Ghali M, Berthier C, Wu C, Niwa-Kawakita M *et al* (2023) Structural basis of PML/RARA oncoprotein targeting by arsenic unravels a cysteine rheostat controlling PML body assembly and function. *Cancer Discov*
- Gonzalez-Prieto R, Eifler-Olivi K, Claessens LA, Willemstein E, Xiao Z, Talavera Ormeno CMP, Ovaa H, Ulrich HD, Vertegaal ACO (2021) Global non-covalent SUMO interaction networks reveal SUMO-dependent stabilization of the non-homologous end joining complex. *Cell reports* 34: 108691
- Haluska P, Jr., Saleem A, Rasheed Z, Ahmed F, Su EW, Liu LF, Rubin EH (1999) Interaction between human topoisomerase I and a novel RING finger/arginine-serine protein. *Nucleic Acids Res* 27: 2538-2544
- Ji L, Huo X, Zhang Y, Yan Z, Wang Q, Wen B (2020) TOPORS, a tumor suppressor protein, contributes to the maintenance of higher-order chromatin architecture. *Biochim Biophys Acta Gene Regul Mech* 1863: 194518
- Liu JCY, Kuhbacher U, Larsen NB, Borgermann N, Garvanska DH, Hendriks IA, Ackermann L, Haahr P, Gallina I, Guerillon C *et al* (2021) Mechanism and function of DNA replication-independent DNA-protein crosslink repair via the SUMO-RNF4 pathway. *EMBO J* 40: e107413
- Rasheed ZA, Saleem A, Ravee Y, Pandolfi PP, Rubin EH (2002) The topoisomerase I-binding RING protein, topors, is associated with promyelocytic leukemia nuclear bodies. *Exp Cell Res* 277: 152-160
- Sun Y, Jenkins LM, Touny LHE, Jo U, Yang X, Maity TK, Saha LK, Uribe I, Saha S, Takeda S *et al* (2023) Flap endonuclease 1 repairs DNA-protein crosslinks via ADP-ribosylation. *bioRxiv*: 2023.2010.2019.563118
- Sun Y, Miller Jenkins LM, Su YP, Nitiss KC, Nitiss JL, Pommier Y (2020) A conserved SUMO pathway repairs topoisomerase DNA-protein cross-links by engaging ubiquitin-mediated proteasomal degradation. *Sci Adv* 6
- Weickert P, Li HY, Gotz MJ, Durauer S, Yaneva D, Zhao S, Cordes J, Acampora AC, Forne I, Imhof A, Stingle J (2023) SPRTN patient variants cause global-genome DNA-protein crosslink repair defects. *Nat Commun* 14: 352

Decision Letter, first revision:

Message: Our ref: NSMB-A47670A

2nd Jan 2024

Dear Dr. Mailand,

Thank you for submitting your revised manuscript "Concerted SUMO-targeted ubiquitin ligase activities of TOPORS and RNF4 are essential for stress management and cell proliferation" (NSMB-A47670A). It has now been seen by the original referees and their comments are below. The reviewers find that the paper has improved in revision, and therefore we'll be happy in principle to publish it in Nature Structural & Molecular Biology, pending minor revisions to satisfy the referees' final requests and to comply with our editorial and formatting guidelines.

We are now performing detailed checks on your paper and will send you a checklist detailing our editorial and formatting requirements in about 2 weeks. Please do not upload the final materials and make any revisions until you receive this additional information from us.

To facilitate our work at this stage, it is important that we have a copy of the main text as a word file. If you could please send along a word version of this file as soon as possible, we would greatly appreciate it; please make sure to copy the NSMB account (cc'ed above).

Sincerely,

Katarzyna Ciazynska, PhD
(she/her)
Associate Editor
Nature Structural & Molecular Biology
<https://orcid.org/0000-0002-9899-2428>

Reviewer #1 (Remarks to the Author):

The authors have addressed my concerns. I have no further comments.

Reviewer #2 (Remarks to the Author):

The authors have addressed my comments. The revision is satisfactory. The only suggestion is to cite the paper by Sun, Y. et al. "Parylation prevents the proteasomal degradation of topoisomerase 1 DNA-protein crosslinks and induce their debilitation" as they also found USP7 as a DUB for TOP1-DPCs, which echoes the findings in

the current manuscript that USP7 acts as a DUB for DNMT1-DPCs.

Reviewer #3 (Remarks to the Author):

The authors have addressed my criticisms. The additional data clarify the issues I had raised on the initial version. I fully support publication of this important and significant work.

Author Rebuttal, first revision

Point-by-point reply to the referees' comments

Reviewer #1:

The authors have addressed my concerns. I have no further comments.

Reviewer #2:

The authors have addressed my comments. The revision is satisfactory. The only suggestion is to cite the paper by Sun, Y. et al. "Parylation prevents the proteasomal degradation of topoisomerase I DNA-protein crosslinks and induce their debilitation" as they also found USP7 as a DUB for TOP1-DPCs, which echos the findings in the current manuscript that USP7 acts as a DUB for DNMT1-DPCs.

We thank the Reviewer for the suggestion but would like to point out that we do not show that USP7 acts as a DUB for DNMT1 DPCs, although we do not rule out this possibility. Rather, our collective findings clearly favor the notion that the principal involvement of USP7 in DNMT1 DPC resolution pertains to its role in deubiquitylating TOPORS, which is instrumental for preserving TOPORS expression in cells (e.g. Figure 1h,i; Extended Data Figure 1i-m). To avoid confusion, we therefore believe it is prudent not to cite the paper suggested by the Reviewer.

Reviewer #3:

The authors have addressed my criticisms. The additional data clarify the issues I had raised on the initial version. I fully support publication of this important and significant work.

Final Decision Letter:**Message:** 26th Mar 2024

Dear Dr. Mailand,

We are now happy to accept your revised paper "Concerted SUMO-targeted ubiquitin ligase activities of TOPORS and RNF4 are essential for stress management and cell proliferation" for publication as an Article in Nature Structural & Molecular Biology.

Your paper will be published online soon after we receive proof corrections and will appear

in print in the next available issue. You can find out your date of online publication by contacting the production team shortly after sending your proof corrections.

Please note that *Nature Structural & Molecular Biology* is a Transformative Journal (TJ). Authors may publish their research with us through the traditional subscription access route or make their paper immediately open access through payment of an article-processing charge (APC). Authors will not be required to make a final decision about access to their article until it has been accepted. Find out more about Transformative Journals

Authors may need to take specific actions to achieve compliance with funder and institutional open access mandates. If your research is supported by a funder that requires immediate open access (e.g. according to Plan S principles) then you should select the gold OA route, and we will direct you to the compliant route where possible. For authors selecting the subscription publication route, the journal's standard licensing terms will need to be accepted, including self-archiving policies. Those licensing terms will supersede any other terms that the author or any third party may assert apply to any

version of the manuscript.

Sincerely,

Katarzyna Ciazynska, PhD
(she/her)
Associate Editor
Nature Structural & Molecular Biology
<https://orcid.org/0000-0002-9899-2428>